# Fast Spectrally Sparse Signal Reconstruction via Jacobi-Preconditioned Gradient Descent

Jian-Feng Cai [1 2]   Xueyang Quan [1]   Yang Wang [1]   Jiaxi Ying [1]

## Abstract

Spectrally sparse signal reconstruction arises in a wide range of applications and can be formulated as a low-rank Hankel matrix completion problem. We develop a Jacobi-preconditioned gradient descent method that preserves the low per-iteration complexity of first-order algorithms while achieving linear convergence at a rate independent of the condition number. By introducing a generator that maps factor-based iterates to matrix space, we establish equivalence with manifold-based methods, enabling direct convergence analysis while avoiding the need to define distances under complex-symmetric factorization ambiguity. Extensive experiments demonstrate that the proposed algorithm outperforms state-of-the-art methods in both iteration count and computational time across a broad range of problem settings.

## 1. Introduction

Spectrally sparse signals—superpositions of a few complex exponentials—arise in diverse scientific and engineering applications, including inverse scattering in seismic imaging (Borcea et al., 2002), nuclear magnetic resonance spectroscopy (Qu et al., 2015; Ying et al., 2017; Lu et al., 2017), direction-of-arrival estimation (Wu et al., 2023), and analog-to-digital conversion (Cohen et al., 2018; Tropp et al., 2009). This paper studies one-dimensional spectrally $r$-sparse signals of the form

$$x(t) = \sum_{k=1}^{r} d_k e^{(2\pi \imath f_k - \tau_k)t}, \qquad (1)$$

[1]Department of Mathematics, The Hong Kong University of Science and Technology, Hong Kong [2]IAS Center for AI for Scientific Discoveries, The Hong Kong University of Science and Technology, Hong Kong. Correspondence to: Xueyang Quan <xquan@connect.ust.hk>, Jiaxi Ying <jx.ying@connect.ust.hk>.

*Proceedings of the 43rd International Conference on Machine Learning*, Seoul, South Korea. PMLR 306, 2026. Copyright 2026 by the author(s).

where $\imath = \sqrt{-1}$, $d_k \in \mathbb{C}$ are complex amplitudes, $f_k \in [0, 1)$ are normalized frequencies, and $\tau_k \geq 0$ are damping factors. Sampling $x(t)$ at $t \in \{0, \ldots, n_s - 1\}$ yields the discrete vector $\boldsymbol{x} = [x_0, \ldots, x_{n_s-1}]^\top \in \mathbb{C}^{n_s}$. In practice, collecting all samples is often prohibitive due to experimental cost or hardware limitations, so only a subset of entries can be observed. Let $\Omega \subset \{0, \ldots, n_s - 1\}$ with $|\Omega| = m$ denote the observed index set, and let $\mathcal{P}_\Omega$ be the coordinate projection onto $\Omega$: $\mathcal{P}_\Omega(\boldsymbol{x}) = \sum_{a \in \Omega} \langle \boldsymbol{x}, \boldsymbol{e}_a \rangle \boldsymbol{e}_a$, where $\{\boldsymbol{e}_a\}_{a=0}^{n_s-1}$ is the canonical basis of $\mathbb{C}^{n_s}$. The goal is to recover $\boldsymbol{x}$ from its partial observations $\mathcal{P}_\Omega(\boldsymbol{x})$:

$$\text{find } \boldsymbol{z} \qquad \text{subject to} \quad \mathcal{P}_\Omega(\boldsymbol{z}) = \mathcal{P}_\Omega(\boldsymbol{x}). \qquad (2)$$

This problem is ill-posed without additional structure: the unobserved entries can be arbitrary yet still satisfy (2).

Nevertheless, compressed sensing (Candès et al., 2006; Donoho, 2006) and matrix completion (Candès & Recht, 2009; Recht et al., 2010) show that structured signals can be recovered from far fewer measurements than their ambient dimension. The challenge is that $\boldsymbol{x}$ has only $O(r)$ degrees of freedom, yet its spectral support is typically *off the grid*. When $\tau_k = 0$, one can discretize the frequency interval and apply standard compressed sensing methods (Candès et al., 2006; Chen et al., 2001; Dai & Milenkovic, 2009; Needell & Tropp, 2009), but discretization induces *basis mismatch* (Chi et al., 2011), which degrades sparsity and recovery. Atomic norm minimization (ANM) avoids discretization by working with continuous frequencies and enjoys strong guarantees under frequency separation (Yang et al., 2016; Tang et al., 2013). However, ANM struggles with damped exponentials and closely spaced components (Yi et al., 2023). These limitations motivate our approach: we lift the signal to a structured matrix whose low rank encodes its spectral sparsity.

The lifting is defined by the Hankel operator $\mathcal{H} : \mathbb{C}^{n_s} \to \mathbb{C}^{n_1 \times n_2}$, with $n_1 + n_2 = n_s + 1$, which maps $\boldsymbol{x}$ to a matrix with constant anti-diagonals, namely $[\mathcal{H}\boldsymbol{x}]_{i,j} = x_{i+j}$. Throughout, we assume $n_s$ is odd and set $n_1 = n_2 = n$, so that $\mathcal{H}\boldsymbol{x} \in \mathbb{C}^{n \times n}$ with $2n = n_s + 1$; when $n_s$ is even, the last sample is simply treated as unobserved. For a spectrally sparse signal $\boldsymbol{x}$ of the form (1), the lifted Hankel matrix $\mathcal{H}\boldsymbol{x}$ admits the following Vandermonde

decomposition (Ying et al., 2018; Cai et al., 2016)

$$\mathcal{H}\boldsymbol{x} = \boldsymbol{E}_L \boldsymbol{D} \boldsymbol{E}_R^\top, \tag{3}$$

where $\boldsymbol{D} = \mathrm{diag}(d_1, \ldots, d_r)$, and $\boldsymbol{E}_L$ and $\boldsymbol{E}_R$ are Vandermonde matrices with nodes $y_k = e^{(2\pi i f_k - \tau_k)}$. Equivalently, the $k$th columns of $\boldsymbol{E}_L$ and $\boldsymbol{E}_R$ are geometric sequences in $y_k$ of length $n$, so each rank-one term in (3) represents a (possibly damped) complex exponential, and hence $\mathrm{rank}(\mathcal{H}\boldsymbol{x}) = r$. Thus, spectral sparsity translates into low rank after the Hankel lifting, and recovery from partial samples can be formulated as a low-rank Hankel matrix completion problem:

$$\min_{\boldsymbol{z} \in \mathbb{C}^{n_s}} \quad \langle \mathcal{P}_\Omega(\mathcal{H}\boldsymbol{z} - \mathcal{H}\boldsymbol{x}), \ \mathcal{H}\boldsymbol{z} - \mathcal{H}\boldsymbol{x} \rangle$$
$$\text{subject to} \quad \mathrm{rank}(\mathcal{H}\boldsymbol{z}) = r. \tag{4}$$

With a slight abuse of notation, we use $\Omega$ to denote both the set of observed time indices and the corresponding set of revealed anti-diagonals of $\mathcal{H}\boldsymbol{x}$. Although our analysis focuses on the undamped case ($\tau_k = 0$), the method itself also applies to damped signals.

Existing approaches to low-rank Hankel matrix completion fall into two broad categories: convex relaxations and nonconvex optimization methods. Convex formulations replace the rank constraint with a tractable surrogatemost often the nuclear normand admit rigorous recovery guarantees; a representative example is enhanced matrix completion (EMaC) (Chen & Chi, 2013). Their primary limitation is that solving the resulting semidefinite or nuclear-norm programs is expensive at scale. Nonconvex methods, by contrast, enforce low rank directly and therefore tend to scale more favorably.

Within the nonconvex family, early methods take a feasible-point viewpoint, alternating between fitting the observed entries under Hankel structure and projecting onto the low-rank set via projected-gradient steps and iterative hard thresholding (Cai et al., 2015). More recent work reduces per-iteration cost by exploiting problem structure: manifold-based methods perform Riemannian gradient updates along tangent spaces of the fixed-rank manifold, thereby avoiding repeated large SVDs (Cai et al., 2019); factor-based methods enforce low rank via matrix factorization with efficient factor-space gradients (Cai et al., 2018); symmetric variants such as Li et al. (2024) leverage the complex symmetry of the lifted Hankel matrix when $n_s$ is odd; and preconditioned schemes such as scaled gradient descent (SGD) (Tong et al., 2021) accelerate convergence by improving conditioning. Despite these advances, the relationship between factor-space and manifold-space updates, and its implications for preconditioning, is not fully clarified. We establish an equivalence between these viewpoints, which clarifies how to strengthen the SGD preconditioner from a matrix-space perspective and motivates a new preconditioning strategy.

**Our contributions** We propose a new preconditioned gradient descent method for spectrally sparse signal reconstruction. Our main contributions are as follows.

- We propose a Jacobi-preconditioned gradient descent method for low-rank Hankel matrix completion, with a preconditioner derived from the Jacobi operator of a generator that maps low-dimensional factors to symmetric matrices. The algorithm maintains the low per-iteration cost of first-order methods while achieving linear convergence with a rate independent of the Hankel matrix condition number. Experiments demonstrate improvements in both iteration count and runtime over state-of-the-art methods. Moreover, the preconditioning framework also extends to other structured and general low-rank matrix recovery problems.

- We unify factorization and manifold perspectives by introducing a *generator* that maps factor-space iterates to their matrix-space counterparts. Under suitable retractions, the generated iterates coincide with those of manifold-based methods. This equivalence enables convergence analysis in matrix space, avoiding the need to define distances in factor space under the complex-symmetric factorization ambiguity $\boldsymbol{X} = \boldsymbol{Z}\boldsymbol{Z}^\top = (\boldsymbol{Z}\boldsymbol{Q})(\boldsymbol{Z}\boldsymbol{Q})^\top$, where $\boldsymbol{Q}$ is complex orthogonal (not unitary), i.e., $\boldsymbol{Q}\boldsymbol{Q}^\top = \boldsymbol{I}$. Moreover, this unification provides a principled route to preconditioning. Viewed in matrix space, it reveals how the SGD preconditioner (Tong et al., 2021) can be strengthened: our method corresponds to an orthogonal projection of the gradient onto tangent space, whereas SGD does not, yielding a new preconditioning strategy.

**Notation:** Bold lowercase, bold uppercase, and calligraphic letters denote vectors, matrices, and operators. For $n_s \in \mathbb{N}$, $[n_s] := \{0, \ldots, n_s - 1\}$. For matrix $\boldsymbol{Z}$, $\boldsymbol{Z}^\top$, $\boldsymbol{Z}^\mathsf{H}$, and $\bar{\boldsymbol{Z}}$ denote transpose, conjugate transpose, and entrywise conjugate; its entries/rows/columns are $\boldsymbol{Z}_{i,j}$, $\boldsymbol{Z}^{(i,:)}$, and $\boldsymbol{Z}^{(:,j)}$. We use $\|\cdot\|$, $\|\cdot\|_F$, and $\|\cdot\|_\infty$ for spectral, Frobenius, and infinity norms, and $\|\boldsymbol{Z}\|_{2,\infty} := \max_i \|\boldsymbol{Z}^{(i,:)}\|$. Inner products $\langle \boldsymbol{x}_1, \boldsymbol{x}_2 \rangle := \boldsymbol{x}_2^\mathsf{H} \boldsymbol{x}_1$ and $\langle \boldsymbol{Z}_1, \boldsymbol{Z}_2 \rangle := \mathrm{trace}(\boldsymbol{Z}_2^\mathsf{H} \boldsymbol{Z}_1)$.

## 2. Proposed Algorithm

This section establishes the equivalence between factor-space and matrix-space iterations and then presents our Jacobi-preconditioned gradient descent method.

Let $\mathcal{H}^*$ denote the adjoint of $\mathcal{H}$. Define the diagonal operator $\mathcal{D}^2 = \mathcal{H}^*\mathcal{H}$ and set $\mathcal{G} := \mathcal{H}\mathcal{D}^{-1}$. By construction, $\mathcal{G}$ is an orthogonalized version of $\mathcal{H}$ and satisfies $\mathcal{G}^*\mathcal{G} = \mathcal{I}$. Consequently, $\mathcal{G}\mathcal{G}^*$ is the orthogonal projector onto the Hankel subspace. We consider the following reconstruction

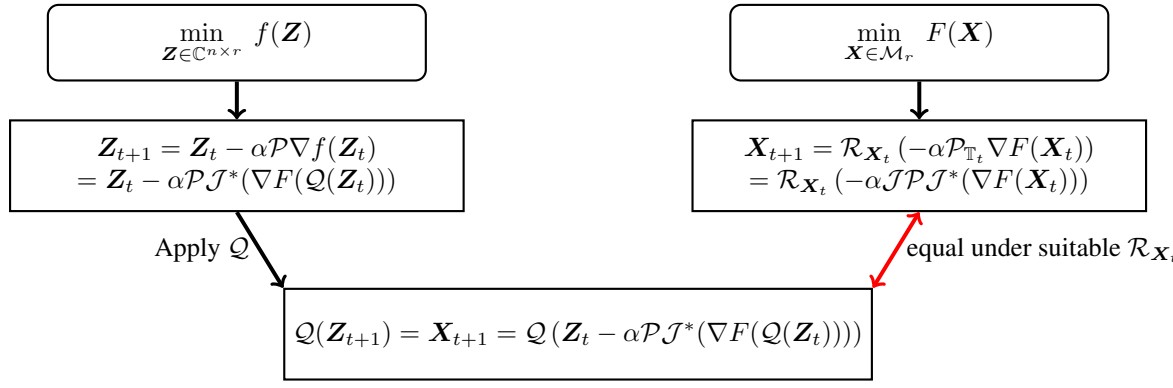

**Figure 1.** Equivalence of factor-space and matrix-space iterates. Applying the generator $\mathcal{Q}$ to the preconditioned gradient descent iterate in factor space (left) yields the Riemannian gradient descent iterate in matrix space (right), under the retraction $\mathcal{R}_{\boldsymbol{X}_t}$ defined in (16).

problem in matrix space:

$$\min_{\boldsymbol{X} \in \mathcal{M}_r} F(\boldsymbol{X}) = \frac{1}{p} g(\boldsymbol{X}) + h(\boldsymbol{X}). \quad (5)$$

Here, $\mathcal{M}_r$ denotes the rank-$r$ Riemannian manifold embedded in the complex symmetric space $\mathbb{C}_S^{n \times n}$, and $F : \mathbb{C}_S^{n \times n} \to \mathbb{R}$ is the objective function. The term $g(\boldsymbol{X}) := \frac{1}{4} \langle \mathcal{P}_\Omega (\mathcal{G}^* \boldsymbol{X} - \boldsymbol{y}), \mathcal{G}^* \boldsymbol{X} - \boldsymbol{y} \rangle$ enforces a least-squares fit to the observed samples, while $h(\boldsymbol{X}) := \frac{1}{4} \| (\mathcal{I} - \mathcal{G}\mathcal{G}^*) \boldsymbol{X} \|_F^2$ penalizes deviations from the Hankel subspace since $\boldsymbol{X}$ is a Hankel matrix if and only if $(\mathcal{I} - \mathcal{G}\mathcal{G}^*) \boldsymbol{X} = \boldsymbol{0}$.

The corresponding factor-space formulation is

$$\min_{\boldsymbol{Z} \in \mathbb{C}^{n \times r}} f(\boldsymbol{Z}) = \frac{1}{p} g(\boldsymbol{Z}\boldsymbol{Z}^\top) + h(\boldsymbol{Z}\boldsymbol{Z}^\top), \quad (6)$$

where $f : \mathbb{C}^{n \times r} \to \mathbb{R}$ is the induced objective in the factor variables. This formulation avoids explicit rank constraints by enforcing low rank through the factorization itself. Having introduced factor- and matrix-space approaches to Hankel matrix completion, we show that the resulting algorithm families are equivalent.

### 2.1. Equivalence of Factor- and Matrix-Space Algorithms via the Generator Framework

We introduce the Jacobi generator framework to establish an equivalence between factor-space and matrix-space algorithms: it characterizes how updates in the factor variable $\boldsymbol{Z}$ induce corresponding updates in the lifted matrix variable $\boldsymbol{X} = \boldsymbol{Z}\boldsymbol{Z}^\top$, and conversely, how matrix-space updates can be realized through factor-space dynamics, while clarifying the role of Jacobi preconditioning in aligning the two perspectives.

**Definition 2.1** (Generator). For $\boldsymbol{Z} \in \mathbb{C}^{n \times r}$, define the generator $\mathcal{Q} : \mathbb{C}^{n \times r} \to \mathbb{C}_S^{n \times n}$ by

$$\mathcal{Q}(\boldsymbol{Z}) := \boldsymbol{Z}\boldsymbol{Z}^\top \in \mathbb{C}_S^{n \times n},$$

where $\mathbb{C}_S^{n \times n}$ is the space of complex symmetric matrices.

The Jacobi of $\mathcal{Q}$ at $\boldsymbol{Z}$, denoted by $\mathcal{J} : \mathbb{C}^{n \times r} \to \mathbb{C}_S^{n \times n}$, is the linear operator

$$\mathcal{J}(\boldsymbol{M}) = \boldsymbol{Z}\boldsymbol{M}^\top + \boldsymbol{M}\boldsymbol{Z}^\top \quad \text{for any } \boldsymbol{M} \in \mathbb{C}^{n \times r}. \quad (7)$$

Its adjoint $\mathcal{J}^* : \mathbb{C}_S^{n \times n} \to \mathbb{C}^{n \times r}$ is given by

$$\mathcal{J}^*(\boldsymbol{Y}) = 2\boldsymbol{Y}\bar{\boldsymbol{Z}} \quad \text{for any } \boldsymbol{Y} \in \mathbb{C}_S^{n \times n}. \quad (8)$$

**Corollary 2.1.** The pseudoinverse of the Jacobian $\mathcal{J}$ satisfies

$$\mathcal{J}^\dagger = (\mathcal{J}^*\mathcal{J})^\dagger \mathcal{J}^*. \quad (9)$$

The proof is provided in Appendix A.4. Using Wirtinger calculus, the gradients with respect to $\boldsymbol{X}$ and $\boldsymbol{Z}$ are

$$\nabla F(\boldsymbol{X}) = \frac{1}{2p} \mathcal{G} \mathcal{P}_\Omega (\mathcal{G}^* \boldsymbol{X} - \boldsymbol{y}) + \frac{1}{2} (\mathcal{I} - \mathcal{G}\mathcal{G}^*) \boldsymbol{X},$$

$$\nabla f(\boldsymbol{Z}) = 2\nabla F(\boldsymbol{Z}\boldsymbol{Z}^\top)\bar{\boldsymbol{Z}}.$$

The generator $\mathcal{Q}$ and its Jacobian $\mathcal{J}$ make the factor- and matrix-space quantities align in a direct way:

1. Variables $\boldsymbol{Z}$ and $\boldsymbol{X}$: $\boldsymbol{X} = \mathcal{Q}(\boldsymbol{Z})$.

2. Objective functions: $f(\boldsymbol{Z}) = F(\mathcal{Q}(\boldsymbol{Z}))$.

3. Gradients: $\nabla_{\boldsymbol{Z}} f(\boldsymbol{Z}) = \mathcal{J}^*(\nabla_{\mathcal{Q}(\boldsymbol{Z})} F(\mathcal{Q}(\boldsymbol{Z})))$.

These identities underpin the iterate-level equivalence between factor and matrix spaces summarized in Figure 1.

### 2.2. Exact SVD-based Jacobi-preconditioned Hankel Gradient Descent

To solve problem (6), we begin with the standard gradient descent method, which updates the iterate by moving in the negative gradient direction:

$$\boldsymbol{Z}_{t+1} = \boldsymbol{Z}_t - \alpha_t \nabla f(\boldsymbol{Z}_t),$$

where $\nabla f(\boldsymbol{Z}_t)$ is the Euclidean gradient at iteration $t$ and $\alpha_t > 0$ is a fixed or adaptive step size.

To accelerate convergence, one can instead use preconditioned gradient descent:

$$\boldsymbol{Z}_{t+1} = \boldsymbol{Z}_t - \alpha_t \mathcal{P} \nabla f(\boldsymbol{Z}_t),$$

where $\mathcal{P}$ denotes a preconditioner, typically chosen to be positive definite. The choice of $\mathcal{P}$ is crucial, as it modifies the metric in which the gradient step is taken and thereby directly affects the convergence behavior of the method. In the present setting, this effect can be characterized precisely: since $\nabla F$ is bi-Lipschitz continuous, the method is optimally conditioned under the standard metric when $\mathcal{J}\mathcal{P}\mathcal{J}^*$ is an orthogonal projector. This observation motivates the following definition and our subsequent choice of preconditioner.

**Definition 2.2** (Orthogonal Projection). *An orthogonal projection on a vector space $V$ is a linear operator $\mathcal{P} : V \to V$ satisfying $\mathcal{P}^2 = \mathcal{P}$ and $\mathcal{P}^* = \mathcal{P}$.*

Guided by this criterion, we introduce the factor-space preconditioner

$$\mathcal{P}_E := (\mathcal{J}^*\mathcal{J})^{\dagger}. \tag{10}$$

By Corollary 2.1, this choice yields

$$\mathcal{J}\mathcal{P}_E\mathcal{J}^* = \mathcal{J}(\mathcal{J}^*\mathcal{J})^{\dagger}\mathcal{J}^* = \mathcal{J}\mathcal{J}^{\dagger}, \tag{11}$$

which is an orthogonal projection in the sense of Definition 2.2. The remainder of this subsection is devoted to deriving an explicit expression for $(\mathcal{J}^*\mathcal{J})^{\dagger}$. To this end, we first characterize the singular pairs of the Jacobi operator.

**Theorem 2.1** (SVD of Jacobi). *Suppose that the SVD of $\boldsymbol{Z} \in \mathbb{C}^{n \times r}$ is $\boldsymbol{Z} = \boldsymbol{U}\boldsymbol{\Sigma}\boldsymbol{V}^{\mathsf{H}} = \sum_{k=1}^{r} \sigma_k \boldsymbol{u}_k \boldsymbol{v}_k^{\mathsf{H}}$, where $\boldsymbol{U} = [\boldsymbol{u}_1, \cdots, \boldsymbol{u}_n] \in \mathbb{C}^{n \times n}$, $\boldsymbol{\Sigma} \in \mathbb{R}^{n \times r}$ with $\mathrm{diag}(\boldsymbol{\Sigma}) = [\sigma_1, \cdots, \sigma_r]$ and other entries are 0, $\boldsymbol{V} = [\boldsymbol{v}_1, \cdots, \boldsymbol{v}_r] \in \mathbb{C}^{r \times r}$. Then the singular pairs for $\mathcal{J}$ can be expressed as:*

- $i = j = 1, \cdots, r.$

$$(\sigma_{ii}, \boldsymbol{U}_{ii}, \boldsymbol{V}_{ii}) = \left(2\sigma_i,\ \boldsymbol{u}_i \boldsymbol{u}_i^{\top},\ \boldsymbol{u}_i \boldsymbol{v}_i^{\top}\right).$$

- $r \geq i > j$ or $i > r \geq j$, and $i, j = 1, \cdots, n.$

$$(\sigma_{ij}, \boldsymbol{U}_{ij}, \boldsymbol{V}_{ij}) = \left(\sqrt{2\sigma_i^2 + 2\sigma_j^2},\ \frac{1}{\sqrt{2}}\boldsymbol{u}_i \boldsymbol{u}_j^{\top} + \frac{1}{\sqrt{2}}\boldsymbol{u}_j \boldsymbol{u}_i^{\top}, \right.$$
$$\left. \sqrt{\frac{\sigma_j^2}{\sigma_i^2 + \sigma_j^2}}\boldsymbol{u}_i \boldsymbol{v}_j^{\top} + \sqrt{\frac{\sigma_i^2}{\sigma_i^2 + \sigma_j^2}}\boldsymbol{u}_j \boldsymbol{v}_i^{\top}\right).$$

The proof is deferred to Appendix A.2. The next result expands the Jacobi operator $\mathcal{J}$ in its singular-vector basis.

**Theorem 2.2.** *Let $\mathcal{J}$ be the Jacobi operator in (7) with adjoint $\mathcal{J}^*$ in (8). Under the singular-pair characterization in Theorem 2.1, $\mathcal{J}$, $\mathcal{J}^*$, and $\mathcal{J}^{\dagger}$ admit the expansions*

$$\mathcal{J}(\boldsymbol{M}) = \sum_{\substack{(i,j):\min\{i,j\}\leq r \\ i \geq j}} \sigma_{ij} \langle \boldsymbol{M}, \boldsymbol{V}_{ij} \rangle \boldsymbol{U}_{ij},$$

$$\mathcal{J}^*(\boldsymbol{Y}) = \sum_{\substack{(i,j):\min\{i,j\}\leq r \\ i \geq j}} \sigma_{ij} \langle \boldsymbol{Y}, \boldsymbol{U}_{ij} \rangle \boldsymbol{V}_{ij},$$

$$\mathcal{J}^{\dagger}(\boldsymbol{Y}) = \sum_{\substack{(i,j):\min\{i,j\}\leq r \\ i \geq j}} \sigma_{ij}^{-1} \langle \boldsymbol{Y}, \boldsymbol{U}_{ij} \rangle \boldsymbol{V}_{ij}.$$

A detailed proof of Theorem 2.2 is given in Appendix A.3. Combining Theorem 2.2 with Corollary 2.1 yields a basis representation of the exact preconditioner $(\mathcal{J}^*\mathcal{J})^{\dagger}$:

$$\mathcal{P}_E(\boldsymbol{M}) = \sum_{\substack{(i,j):\min\{i,j\}\leq r \\ i \geq j}} \sigma_{ij}^{-2} \langle \boldsymbol{M}, \boldsymbol{V}_{ij} \rangle \boldsymbol{V}_{ij},$$

which can be evaluated by summing three groups of terms:

(A) $i = j = 1, \cdots, r,\ \sum_{i=1}^{r} \frac{1}{4\sigma_i^2}(\boldsymbol{u}_i^{\mathsf{H}} \boldsymbol{M} \bar{\boldsymbol{v}}_i)\boldsymbol{u}_i \boldsymbol{v}_i^{\top},$

(B) $r \geq i > j$,

$$\sum_{i,j=1}^{r} \frac{1}{2(\sigma_i^2 + \sigma_j^2)^2} \left(\sigma_j \boldsymbol{u}_i^{\mathsf{H}} \boldsymbol{M} \bar{\boldsymbol{v}}_j + \sigma_i \boldsymbol{u}_j^{\mathsf{H}} \boldsymbol{M} \bar{\boldsymbol{v}}_i\right)$$
$$\cdot \left(\sigma_j \boldsymbol{u}_i \boldsymbol{v}_j^{\top} + \sigma_i \boldsymbol{u}_j \boldsymbol{v}_i^{\top}\right)$$

(C) $i > r \geq j, i = r+1, \cdots, n, j = 1, \cdots, r,$

$$\sum_{i=r+1}^{n} \frac{1}{2}\boldsymbol{u}_i \boldsymbol{u}_i^{\mathsf{H}} \boldsymbol{M} \sum_{j=1}^{r} \frac{1}{\sigma_j^2} \bar{\boldsymbol{v}}_j \boldsymbol{v}_j^{\top}$$
$$= \frac{1}{2}\left(\boldsymbol{I}_n - \boldsymbol{Z}(\boldsymbol{Z}^{\mathsf{H}}\boldsymbol{Z})^{-1}\boldsymbol{Z}^{\mathsf{H}}\right)\boldsymbol{M}(\overline{\boldsymbol{Z}^{\mathsf{H}}\boldsymbol{Z}})^{-1}$$

With $\mathcal{P}_E$ in hand, we update $\boldsymbol{Z}_{t+1}$ via preconditioned gradient descent, yielding our *Exact Jacobi-preconditioned Hankel Gradient Descent* (exact JHGD) algorithm. Although exact JHGD converges rapidly, evaluating the preconditioned gradient $\mathcal{P}_E \nabla f(\boldsymbol{Z})$ requires costly nested double loops at each iteration. We therefore develop an approximate JHGD variant retaining the key structure of $\mathcal{P}_A$ while being substantially more efficient in practice.

### 2.3. Jacobi-Preconditioned Hankel Gradient Descent

The computational bottleneck of the exact preconditioner $\mathcal{P}_E$ comes from group (B), which involves nested sums over all pairs $(i,j)$. We reduce this cost by approximating the combined groups (A)–(B) contribution without enumerating all pairs. The following theorem shows that this term is well approximated by a factored expression computable with a single loop.

**Theorem 2.3.** *Assume $\boldsymbol{M}$ is uniformly distributed in $\boldsymbol{u}_i$ and $\boldsymbol{v}_j$. Then we have*

$$
\mathbb{E}_{\boldsymbol{M}}\left[\left\|\frac{(\sigma_j \boldsymbol{u}_i^{\mathsf{H}} \boldsymbol{M} \bar{\boldsymbol{v}}_j + \sigma_i \boldsymbol{u}_j^{\mathsf{H}} \boldsymbol{M} \bar{\boldsymbol{v}}_i)}{2(\sigma_i^2 + \sigma_j^2)^2}\left(\sigma_j \boldsymbol{u}_i \boldsymbol{v}_j^{\top} + \sigma_i \boldsymbol{u}_j \boldsymbol{v}_i^{\top}\right)\right.\right.
$$

$$
\left.\left. - \frac{1}{4\sigma_j^2} \boldsymbol{u}_i \boldsymbol{u}_i^{\mathsf{H}} \boldsymbol{M} \bar{\boldsymbol{v}}_j \boldsymbol{v}_j^{\top} - \frac{1}{4\sigma_i^2} \boldsymbol{u}_j \boldsymbol{u}_j^{\mathsf{H}} \boldsymbol{M} \bar{\boldsymbol{v}}_i \boldsymbol{v}_i^{\top}\right\|_F^2\right]
$$

$$
\leq \frac{1}{2\sigma_{\min}^4(\boldsymbol{Z})} \mathbb{E}_{\boldsymbol{M}}\left[\|\boldsymbol{M}\|_2^2\right],
$$

*where $\sigma_{\min}(\boldsymbol{Z})$ is the minimum singular value of $\boldsymbol{Z}$; $\boldsymbol{u}_i$ and $\boldsymbol{v}_j$ are singular vectors of $\boldsymbol{Z}$.*

See Appendix A.5 for the proof of Theorem 2.3. We retain the exact contribution from group (C) (i.e., $i > r \geq j$) and substitute the approximation from Theorem 2.3 for groups (A)(B). Combining these two components yields the following approximate preconditioner:

$$
\mathcal{P}_A(\boldsymbol{M}) := \left(\frac{1}{2}\boldsymbol{I}_n - \frac{1}{4}\boldsymbol{Z}(\boldsymbol{Z}^{\mathsf{H}}\boldsymbol{Z})^{-1}\boldsymbol{Z}^{\mathsf{H}}\right)\boldsymbol{M}(\overline{\boldsymbol{Z}^{\mathsf{H}}\boldsymbol{Z}})^{-1}, \quad (12)
$$

which eliminates the nested double loops. The resulting Jacobi-preconditioned Hankel gradient descent method is summarized below.

---

**Algorithm 1** Jacobi-Preconditioned Hankel Gradient Descent (JHGD) in Factor Space

---

**Input:** $\boldsymbol{Z}_0, \mathcal{P}_{\Omega}(\boldsymbol{x}), \Omega, \alpha$

   **for** $t = 0, 1, \cdots, K$ **do**

   Calculate the preconditioned gradient $\boldsymbol{W}_t$

$$
\boldsymbol{W}_t = \left(\frac{1}{2}\boldsymbol{I}_n - \frac{1}{4}\boldsymbol{Z}_t(\boldsymbol{Z}_t^{\mathsf{H}}\boldsymbol{Z}_t)^{-1}\boldsymbol{Z}_t^{\mathsf{H}}\right)\nabla f(\boldsymbol{Z}_t)(\overline{\boldsymbol{Z}_t^{\mathsf{H}}\boldsymbol{Z}_t})^{-1}
$$

   $\boldsymbol{Z}_{t+1} = \boldsymbol{Z}_t - \alpha\boldsymbol{W}_t$

   **end for**

**Output:** $\boldsymbol{x}_K = \mathcal{D}^{-1}\boldsymbol{y}_K$ with $\boldsymbol{y}_K = \mathcal{G}^*(\boldsymbol{Z}_K(\boldsymbol{Z}_K)^{\top})$.

---

**Theorem 2.4.** *Let $\mathcal{J}$ be the Jacobi of the generator $\mathcal{Q}$. Then the exact preconditioner $\mathcal{P}_E$ in (10) and the approximate preconditioner $\mathcal{P}_A$ in (12) induce the same orthogonal projection onto the tangent space:*

$$
\mathcal{J}\mathcal{P}_A\mathcal{J}^* = \mathcal{J}\mathcal{P}_E\mathcal{J}^* = \mathcal{J}\mathcal{J}^{\dagger}, \quad (13)
$$

*where $\mathcal{J}^{\dagger}$ denotes the pseudoinverse of $\mathcal{J}$.*

Hence, $\mathcal{P}_A$ preserves the key geometric property of $\mathcal{P}_E$: both induce the same orthogonal projection onto the tangent space and therefore the same Riemannian gradient. Thus, $\mathcal{P}_A$ is a computationally cheaper surrogate for $\mathcal{P}_E$, differing only in higher-order terms. The proof of Theorem 2.4 is deferred to Appendix A.6.

For efficient implementation, we rewrite the gradient as

$$
\nabla f(\boldsymbol{Z}) = \mathcal{G}\left(p^{-1}\mathcal{P}_{\Omega}\left(\mathcal{G}^*\left(\boldsymbol{Z}\boldsymbol{Z}^{\top}\right) - \boldsymbol{y}\right) - \mathcal{G}^*\left(\boldsymbol{Z}\boldsymbol{Z}^{\top}\right)\right)
$$

$$
\cdot \overline{\boldsymbol{Z}} + \boldsymbol{Z}\left(\boldsymbol{Z}^{\top}\overline{\boldsymbol{Z}}\right).
$$

Note that $\mathcal{G}^*\left(\boldsymbol{Z}\boldsymbol{Z}^{\top}\right)$ can be computed busing $r$ fast convolutions. Computing $\boldsymbol{Z}(\boldsymbol{Z}^{\mathsf{H}}\boldsymbol{Z})^{-1}$ costs $2nr^2$ flops. Let $\boldsymbol{z} = p^{-1}\mathcal{P}_{\Omega}\left(\mathcal{G}^*\left(\boldsymbol{Z}\boldsymbol{Z}^{\top}\right) - \boldsymbol{y}\right) - \mathcal{G}^*\left(\boldsymbol{Z}\boldsymbol{Z}^{\top}\right)$. Then $(\mathcal{G}\boldsymbol{z})\overline{\boldsymbol{Z}(\boldsymbol{Z}^{\mathsf{H}}\boldsymbol{Z})^{-1}}$ can be formed via $r$ fast Hankel matrix-vector multiplications, costing $Cnr\log(n)$ flops for a constant $C > 0$. The remaining projection term $\boldsymbol{Z}(\boldsymbol{Z}^{\mathsf{H}}\boldsymbol{Z})^{-1}\boldsymbol{Z}^{\mathsf{H}}(\mathcal{G}\boldsymbol{z})\overline{\boldsymbol{Z}(\boldsymbol{Z}^{\mathsf{H}}\boldsymbol{Z})^{-1}}$ costs an additional $2nr^2$ flops. Overall, the per-iteration complexity of Algorithm 1 is $\mathcal{O}(n\log(n)r + nr^2)$.

### 2.4. JHGD Expressed in Matrix Space

Problem (5) can be addressed using Riemannian gradient descent. At each iteration, one takes a step along the negative *Riemannian* gradient in the tangent space and then maps the result back to the manifold via a retraction.

**Definition 2.3** (Retraction (Absil et al., 2009))**.** A retraction on a manifold $\mathcal{M}$ is a smooth mapping $\mathcal{R}$ from the tangent bundle $T\mathcal{M}$ onto $\mathcal{M}$ with the following properties. Let $\mathcal{R}_x$ denote the restriction of $\mathcal{R}$ to $T_x\mathcal{M}$.

1. $\mathcal{R}_x\left(0_x\right) = x$, where $0_x$ denotes the zero element of $T_x\mathcal{M}$.

2. With the canonical identification $T_{0_x}T_x\mathcal{M} \simeq T_x\mathcal{M}$, $\mathcal{R}_x$ satisfies $D\mathcal{R}_x\left(0_x\right) = id_{T_x\mathcal{M}}$, where $id_{T_x\mathcal{M}}$ denotes the identity mapping on $T_x\mathcal{M}$.

With $\mathcal{R}_{\boldsymbol{X}_t}$, Riemannian gradient descent updates as

$$
\boldsymbol{X}_{t+1} = \mathcal{R}_{\boldsymbol{X}_t}\left(-\alpha_t\nabla_{\mathcal{M}_r}F(\boldsymbol{X}_t)\right).
$$

Let the rank-$r$ iterate $\boldsymbol{X}_t$ have Takagi decomposition $\boldsymbol{X}_t = \boldsymbol{U}_t\boldsymbol{\Sigma}_t\boldsymbol{U}_t^{\top}$, where $\boldsymbol{U}_t \in \mathbb{C}^{n \times r}$ is orthonormal and $\boldsymbol{\Sigma}_t \in \mathbb{R}^{r \times r}$ is diagonal nonnegative. Define

$$
\mathbb{T}_t = \left\{\boldsymbol{U}_t\boldsymbol{M}^{\top} + \boldsymbol{M}\boldsymbol{U}_t^{\top} \mid \boldsymbol{M} \in \mathbb{C}^{n \times r}\right\}. \quad (14)
$$

This is the tangent space of $\mathcal{M}_r$ at $\boldsymbol{X}_t$. The orthogonal projection (with respect to the Frobenius inner product) of an arbitrary $\boldsymbol{Y} \in \mathbb{C}^{n \times n}$ onto $\mathbb{T}_t$ is

$$
\mathcal{P}_{\mathbb{T}_t}(\boldsymbol{Y}) = \boldsymbol{U}_t\boldsymbol{U}_t^{\mathsf{H}}\boldsymbol{Y} + \boldsymbol{Y}\bar{\boldsymbol{U}}_t\bar{\boldsymbol{U}}_t^{\mathsf{H}} - \boldsymbol{U}_t\boldsymbol{U}_t^{\mathsf{H}}\boldsymbol{Y}\bar{\boldsymbol{U}}_t\bar{\boldsymbol{U}}_t^{\mathsf{H}}. \quad (15)
$$

Under the canonical (embedded) metric on $\mathcal{M}_r$, the Riemannian gradient is the projected Euclidean gradient:

$$
\nabla_{\mathcal{M}_r}F(\boldsymbol{X}_t) = \mathcal{P}_{\mathbb{T}_t}\left(\nabla F(\boldsymbol{X}_t)\right).
$$

Therefore, the update can be written as

$$
\boldsymbol{X}_{t+1} = \mathcal{R}_{\boldsymbol{X}_t}\left(-\alpha_t\mathcal{P}_{\mathbb{T}_t}(\nabla F(\boldsymbol{X}_t))\right).
$$

A common choice of retraction on $\mathcal{M}_r$ is the truncated SVD, which is defined as

$$\mathcal{R}_{\boldsymbol{X}}(\boldsymbol{Z}) := \sum_{i=1}^{r} \sigma_i(\boldsymbol{X} + \boldsymbol{Z}) \boldsymbol{u}_i \boldsymbol{v}_i^{\mathsf{H}},$$

where $\sigma_i(\boldsymbol{X} + \boldsymbol{Z})$ are the singular values in decreasing order and $\boldsymbol{u}_i$, $\boldsymbol{v}_i$ are the corresponding left and right singular vectors, respectively.

In this paper, we propose a retraction induced by the generator $\mathcal{Q}$. Suppose $\boldsymbol{X} = \mathcal{Q}(\boldsymbol{Z}) = \boldsymbol{Z}\boldsymbol{Z}^{\top}$, and $\nabla F(\boldsymbol{X})$ denotes the gradient of the objective function in the matrix space. For any $\boldsymbol{A} \in \boldsymbol{X} + \mathbb{T}_{\boldsymbol{X}}$ parametrized as $\boldsymbol{A} = \boldsymbol{X} + \mathcal{J}\mathcal{J}^{\dagger}\boldsymbol{\Xi}$ for some $\boldsymbol{\Xi} \in \mathbb{C}_S^{n \times n}$ satisfying $\left(\mathcal{I} - \mathcal{J}\mathcal{J}^{\dagger}\right)\boldsymbol{\Xi} = \left(\mathcal{I} - \mathcal{J}\mathcal{J}^{\dagger}\right)\nabla F(\boldsymbol{X})$, we define $\mathcal{R} : \boldsymbol{X} + \mathbb{T}_{\boldsymbol{X}} \to \mathcal{M}_r$ by

$$\mathcal{R}(\boldsymbol{A}) = \mathcal{Q}\left(\boldsymbol{Z} + \mathcal{J}^{\dagger}\boldsymbol{\Xi}\right) = \left(\boldsymbol{Z} + \mathcal{J}^{\dagger}\boldsymbol{\Xi}\right)\left(\boldsymbol{Z} + \mathcal{J}^{\dagger}\boldsymbol{\Xi}\right)^{\top}$$

which is well-defined.

Well-definedness follows from existence and uniqueness. Existence holds since the parametrization spans the tangent space, so any admissible $\boldsymbol{A}$ can be represented by a tangent perturbation of $\boldsymbol{X}$, with the orthogonal condition determining the remaining component. Uniqueness holds because identical $\boldsymbol{A}$ implies identical tangent components, and the orthogonal condition forces the complementary components to coincide. Hence, $\mathcal{R}(\boldsymbol{A})$ is independent of the chosen parametrization.

Using $\mathcal{R}$, the iteration can be rewritten as

$$\boldsymbol{X}_{t+1} = \mathcal{R}\left(\boldsymbol{X}_t - \alpha_t \mathcal{P}_{\mathbb{T}_t}\left(\nabla F(\boldsymbol{X}_t)\right)\right).$$

Finally, letting $\mathcal{M} = \mathcal{M}_r$, we define the associated retraction $\mathcal{R}_{\boldsymbol{X}} : T\mathcal{M} \to \mathcal{M}$ by

$$\mathcal{R}_{\boldsymbol{X}}\left(\mathcal{J}(\boldsymbol{\Delta})\right) := \mathcal{R}\left(\boldsymbol{X} + \mathcal{J}(\boldsymbol{\Delta})\right). \tag{16}$$

We now relate the factor-space algorithms to their matrix-space counterparts using the generator map in Figure 1.

Consider Algorithm 1, which updates $\boldsymbol{Z}_t$ in factor space using $\mathcal{P}_A$. Applying the generator $\mathcal{Q}$ to $\boldsymbol{Z}_{t+1}$ and expanding the resulting expression, we have

$$\boldsymbol{X}_{t+1} = \boldsymbol{X}_t - \alpha \mathcal{J}\mathcal{P}_A\mathcal{J}^*(\nabla F(\boldsymbol{X}_t)) + \mathcal{O}(\|\boldsymbol{W}_t'\|^2),$$

where $\mathcal{J}\mathcal{P}_A\mathcal{J}^* = \mathcal{J}\mathcal{J}^{\dagger}$ is an orthogonal projection onto the tangent space (by Theorem 2.4), and $\boldsymbol{W}_t'$ is the projection of $\boldsymbol{W}_t = \mathcal{Q}(\alpha\mathcal{P}_A\mathcal{J}^*(\nabla F(\boldsymbol{X}_t)))$ onto the tangent space. Under the suitable retraction $\mathcal{R}_{\boldsymbol{X}_t}$ in (16) and the canonical metric on $\mathcal{M}_r$, the Riemannian gradient is the projected Euclidean gradient, yielding the Riemannian gradient descent iteration:

$$\boldsymbol{X}_{t+1} = \mathcal{R}_{\boldsymbol{X}_t}\left(-\alpha\mathcal{P}_{\mathbb{T}_t}(\nabla F(\boldsymbol{X}_t))\right), \tag{17}$$

where $\mathcal{P}_{\mathbb{T}_t}$ denotes the orthogonal projection onto the tangent space $\mathbb{T}_t$ at $\boldsymbol{X}_t$. This update is precisely the result of $\mathcal{Q}(\boldsymbol{Z}_{t+1})$, thereby confirming the equivalence between the two methods. The matrix-form version of JHGD is summarized in Algorithm 2.

**Remark 2.1.** A prominent preconditioned algorithm in factor space is SGD (Tong et al., 2021), which utilizes the preconditioner $\mathcal{P}_S(\boldsymbol{M}) = \boldsymbol{M}(\overline{\boldsymbol{Z}^{\mathsf{H}}\boldsymbol{Z}})^{-1}$, resulting in the iteration update $\boldsymbol{Z}_{t+1} = \boldsymbol{Z}_t - \alpha\mathcal{P}_S(\nabla f(\boldsymbol{Z}_t))$. The corresponding matrix-space iteration for $\boldsymbol{X}_{t+1}$ is:

$$\mathcal{R}_{\boldsymbol{X}_t}\left(-\alpha(\boldsymbol{U}_t\boldsymbol{U}_t^{\mathsf{H}}\nabla F(\boldsymbol{X}_t) + \nabla F(\boldsymbol{X}_t)\bar{\boldsymbol{U}}_t\bar{\boldsymbol{U}}_t^{\mathsf{H}})\right).$$

Evidently, when calculating the Riemannian gradient, SGD employs an oblique projection onto the tangent space, whereas our (exact) JHGD utilizes an orthogonal projection. This distinction makes our proposed method superior, a claim substantiated by numerical experiments where JHGD outperforms SGD.

---

**Algorithm 2** JHGD in Matrix Space

**Input:** $\boldsymbol{X}_0, \mathcal{P}_{\Omega}(\boldsymbol{x}), \Omega, \alpha$
  for $t = 0, 1, \cdots, K$ do
  Update $\boldsymbol{X}_t$ via Riemannian gradient:

$$\boldsymbol{G}_t = \mathcal{P}_{\mathbb{T}_t}\left[\frac{1}{2p}\mathcal{G}\mathcal{P}_{\Omega}\left(\mathcal{G}^*(\boldsymbol{X}_t) - \boldsymbol{y}\right) + \frac{1}{2}\left(\mathcal{I} - \mathcal{G}\mathcal{G}^*\right)(\boldsymbol{X}_t)\right]$$

  Retraction: $\boldsymbol{X}_{t+1} = \mathcal{R}_{\boldsymbol{X}_t}\left(-\alpha\boldsymbol{G}_t\right).$
  **end for**
**Output:** $\boldsymbol{x}_K = \mathcal{D}^{-1}\boldsymbol{y}_K$ with $\boldsymbol{y}_K = \mathcal{G}^*(\boldsymbol{X}_K)$.

---

## 3. Theoretical Results

This section presents recovery guarantees for JHGD. Our approach is nonconvex due to the low-rank constraint (or factorization) imposed on the lifted matrix. As is common for nonconvex methods, the algorithm naturally follows a two-stage strategy: (i) an initialization stage that computes a coarse estimate, and (ii) a refinement stage that applies JHGD iterations to achieve geometric error contraction.

Although JHGD is implemented in factor space (e.g., $\mathbb{C}^{n \times r}$), our analysis is conducted in matrix space (e.g., $\mathbb{C}^{n \times n}$) using the equivalence established in Section 2.4. The reason is that comparing factors directly is complicated by non-uniqueness: a common approach is to quotient out the ambiguity induced by right transformations, namely

$$\text{dist}(\boldsymbol{Z}, \boldsymbol{Z}_{\star}) := \inf_{\boldsymbol{Q} \in \mathcal{U}} \|\boldsymbol{Z} - \boldsymbol{Z}_{\star}\boldsymbol{Q}\|_F, \tag{18}$$

where $\mathcal{U} := \{\boldsymbol{Q} \in \mathbb{C}^{r \times r} \mid \boldsymbol{Q}\boldsymbol{Q}^{\top} = \boldsymbol{Q}^{\top}\boldsymbol{Q} = \boldsymbol{I}\}$. However, unlike the real orthogonal group, the complex orthogonal group $\mathcal{U}$ is non-compact, which complicates

the existence and attainment of the infimum and obscures first-order optimality conditions for (18). To avoid these technical issues, we establish an equivalent matrix-space view of the iterations and conduct the convergence analysis directly on the lifted iterates $\{\boldsymbol{X}_t\}$.

We adopt a sampling-with-replacement model for $\Omega$: each index is drawn independently and uniformly from $[n_s]$. To ensure recovery from partial observations, we assume the lifted matrix $\mathcal{H}\boldsymbol{x}$ satisfies the following condition.

**Definition 3.1** ($\mu_0$-incoherence). Let $\mathcal{H}\boldsymbol{x} \in \mathbb{C}_S^{n \times n}$ be a rank-$r$ Hankel matrix with Vandermonde decomposition $\mathcal{H}\boldsymbol{x} = \boldsymbol{E}\boldsymbol{D}\boldsymbol{E}^\top$. We say that $\mathcal{H}\boldsymbol{x}$ is $\mu_0$-incoherent if, for some numerical constant $\mu_0 > 0$,

$$\sigma_{\min}(\boldsymbol{E}^\mathsf{H}\boldsymbol{E}) \geq \frac{n}{\mu_0}.$$

This incoherence condition is standard in Hankel matrix completion (Chen & Chi, 2014); in the undamped case, it holds when the minimum wrap-around frequency separation exceeds $2/n_s$ (Liao & Fannjiang, 2016).

### 3.1. Initialization via Takagi Factorization

We use a spectral initializer tailored to the complex symmetric structure. Let $\mathcal{T}_r(\cdot)$ denote the best rank-$r$ approximation (truncated SVD). For a matrix $\boldsymbol{A} = \sum_i \sigma_i \boldsymbol{u}_i \boldsymbol{v}_i^\mathsf{H}$ with singular values $\sigma_1 \geq \sigma_2 \geq \cdots$, define

$$\mathcal{T}_r(\boldsymbol{A}) := \sum_{i=1}^r \sigma_i \boldsymbol{u}_i \boldsymbol{v}_i^\mathsf{H}.$$

Given partial observations $\mathcal{P}_\Omega(\boldsymbol{y})$ with sampling probability $p$, we truncate $p^{-1}\mathcal{H}\mathcal{P}_\Omega(\boldsymbol{y})$ to rank $r$:

$$\boldsymbol{X}_0 := \mathcal{T}_r\Big(p^{-1}\mathcal{H}\mathcal{P}_\Omega(\boldsymbol{y})\Big).$$

By construction, $\boldsymbol{A}_0$ is rank-$r$ and complex symmetric (i.e., $\boldsymbol{X}_0^\top = \boldsymbol{X}_0$), and hence admits a Takagi factorization $\boldsymbol{X}_0 = \boldsymbol{U}_0\boldsymbol{\Sigma}_0\boldsymbol{U}_0^\top$, where $\boldsymbol{U}_0 \in \mathbb{C}^{n \times r}$ has orthonormal columns and $\boldsymbol{\Sigma}_0 \in \mathbb{R}^{r \times r}$ is diagonal. We initialize

$$\boldsymbol{Z}_0 = \boldsymbol{U}_0\,\boldsymbol{\Sigma}_0^{1/2}. \tag{19}$$

Thus, $\boldsymbol{X}_0 = \boldsymbol{Z}_0\boldsymbol{Z}_0^\top$. In practice, $\boldsymbol{Z}_0$ can be computed from a single rank-$r$ truncated SVD. Let $p^{-1}\mathcal{H}\mathcal{P}_\Omega(\boldsymbol{y}) = \boldsymbol{U}\boldsymbol{\Sigma}\boldsymbol{V}^\mathsf{H}$, and denote by $\boldsymbol{U}_r, \boldsymbol{V}_r$ the leading $r$ singular vectors and by $\boldsymbol{\Sigma}_r$ the leading $r \times r$ block of $\boldsymbol{\Sigma}$. As shown in Chebotarev & Teretenkov (2014), one may take

$$\boldsymbol{U}_0 = \boldsymbol{U}_r\big(\boldsymbol{U}_r^\mathsf{H}\overline{\boldsymbol{V}_r}\big)^{1/2}, \quad \text{and} \quad \boldsymbol{\Sigma}_0 = \boldsymbol{\Sigma}_r,$$

where $(\cdot)^{1/2}$ denotes the principal matrix square root.

Starting from $\boldsymbol{Z}_0$, we then run JHGD as a refinement stage and establish recovery guarantees.

**Theorem 3.1.** *Assume $\mathcal{H}\boldsymbol{x} = \mathcal{G}\boldsymbol{y}$ is $\mu_0$-incoherent. Let $0 < \epsilon_0 < \frac{1}{28}$, $1 < \alpha < 3$ be some numerical constants. Then with probability at least $1 - 3n_s^{-2}$, the iterations generated by Algorithm 1, initialized by (19), satisfy*

$$\|\boldsymbol{x}_t - \boldsymbol{x}\| \leq \nu^t\|\boldsymbol{X}_0 - \mathcal{H}\boldsymbol{x}\|_F,$$

*for some contraction factor $\nu < 1$, provided that*

$$m \geq C \max\left\{\epsilon_0^{-2}\mu_0, \mu_0, (1+\epsilon_0)^{\frac{1}{2}}\epsilon_0^{-1}\mu_0^{\frac{1}{2}}\right\} \kappa^2 r n_s^{\frac{1}{2}} \log(n_s)$$

*for some numerical universal constant $C > 0$, where $\kappa = \frac{\sigma_{\max}(\mathcal{H}\boldsymbol{x})}{\sigma_{\min}(\mathcal{H}\boldsymbol{x})}$ denotes the condition number of $\mathcal{H}\boldsymbol{x}$.*

The proof is provided in Appendix B.2. Theorem 3.1 guarantees that, under the stated incoherence and sampling assumptions, the JHGD iterates converge to the ground truth at a linear rate, with the reconstruction error decreasing by a fixed constant factor at each iteration. Importantly, this contraction factor is strictly less than 1 and is independent of the condition number of $\mathcal{H}\boldsymbol{x}$.

### 3.2. Initialization via Resampled JHGD

The sample complexity in Theorem 3.1 has a suboptimal $\sqrt{n_s}$ factor, though a spectrally sparse signal has only $O(r)$ degrees of freedom. To remove it, we use resampled gradient descent with iterative trimming, following Cai et al. (2019). Resampling uses fresh samples at each update to avoid statistical coupling and sharpen concentration, while trimming enforces $\mu_0$-incoherence by projection. The full resampling/trimming procedure is given in Appendix B.3.1; see also Algorithms 3 and 4.

With this initializer, we run the standard JHGD refinement stage and obtain recovery guarantees under improved sampling requirements.

**Theorem 3.2.** *Assume $\mathcal{H}\boldsymbol{x}$ is $\mu_0$-incoherent. Let $0 < \epsilon_0 < \frac{1}{28}$ and $1 < \alpha < 3$, and set $K = \left\lceil 13\log\left(\frac{\sqrt{n_s\log(n_s)}}{91\epsilon_0}\right)\right\rceil$. Then, with probability at least $1-(2K+3)n_s^{-2}$, the iterates generated by Algorithm 1, initialized at the output $\widetilde{\boldsymbol{Z}}_K$ of Algorithm 3, satisfy*

$$\|\boldsymbol{x}_t - \boldsymbol{x}\| \leq \nu^t\|\boldsymbol{X}_0 - \mathcal{H}\boldsymbol{x}\|_F$$

*with $\nu < 1$, provided that*

$$m \geq C'\mu_0\kappa^6 r^2 \log(n_s) \log\left(\frac{\sqrt{n_s\log(n_s)}}{91\epsilon_0}\right)$$

*for some universal constant $C'$.*

We defer the proof of Theorem 3.2 to Appendix B.3. Our JHGD framework extends to multidimensional spectrally sparse signals via the multidimensional Hankel lift: the resulting multi-level block Hankel matrix retains a rank-$r$ Vandermonde decomposition.

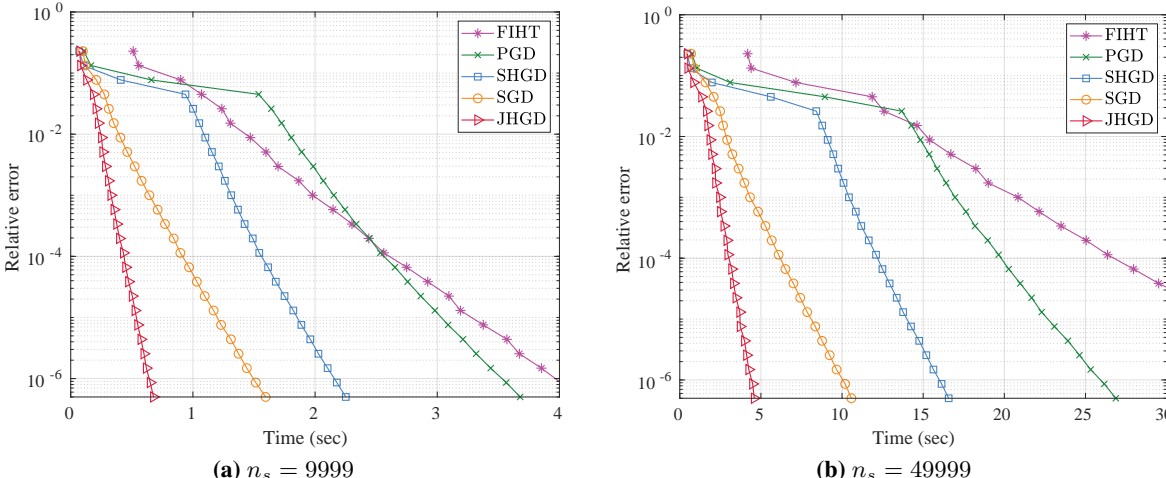

**Figure 2.** Average runtime (sec) versus relative reconstruction error for PGD, SHGD, FIHT, SGD, and JHGD on undamped signals without frequency separation. Parameters: $r = 50$ and $m = 2000$. Left: $n_s = 9999$. Right: $n_s = 49999$.

## 4. Numerical Experiments

We compare JHGD against four state-of-the-art baselines: projected gradient descent (PGD) (Cai et al., 2018), symmetric Hankel gradient descent (SHGD) (Li et al., 2024), fast iterative hard thresholding (FIHT) (Cai et al., 2019), and scaled gradient descent (SGD) (Tong et al., 2021). All experiments are implemented in MATLAB R2021b and run on a desktop computer.

We generate spectrally sparse signals $x \in \mathbb{C}^{n_s}$ according to (1), following the experimental settings in Cai et al. (2018; 2019); Li et al. (2024). Each coefficient $d_k$ has a phase drawn uniformly from $[0, 2\pi]$ and an amplitude $1 + 10^{0.5c_k}$, where $c_k \sim \text{Unif}[0, 1]$. Frequencies are drawn from two models: (i) i.i.d. uniform on $[0, 1)$ (no separation), or (ii) a separated model enforcing a minimum wrap-around separation of at least $1.5/n_s$. For undamped signals, we set $\tau_k = 0$.

### 4.1. Computational Efficiency

We assess computational efficiency by measuring runtime while varying (i) spectral sparsity $r$, (ii) the number of observations $m$, and (iii) the signal length $n_s$. For each $(n_s, m, r)$, we run 10 independent trials. In all runs, each method uses a fixed step size: for PGD and SHGD, we follow (Li et al., 2024) and set $\alpha = 0.75/\sigma_1(X_0)$, where $X_0 = \mathcal{T}_r(p^{-1}\mathcal{HP}_\Omega(x))$; for SGD, we set $\alpha = 0.5$ as recommended in (Tong et al., 2021); and for JHGD, we use $\alpha = 1.8$.

All methods terminate once the relative change of iterates $\|x_{t+1} - x_t\|/\|x_t\| \leq \text{tol}$. We set $\text{tol} = 5 \times 10^{-6}$ for PGD, SHGD, SGD, and JHGD. For FIHT, we use a slightly looser threshold, $\text{tol} = 5 \times 10^{-5}$, consistent with prior comparisons (Cai et al., 2019; 2018). This is because,

under the same $\text{tol}$, FIHT typically attains much smaller errors, making runtime comparisons less meaningful. After termination, we report the relative reconstruction error

$$\text{RelErr} = \frac{\|x_{\text{rec}} - x\|_2}{\|x\|_2}, \qquad (20)$$

where $x$ and $x_{\text{rec}}$ denote ground-truth and reconstructed signals, respectively. All methods use fixed step sizes: PGD and SHGD use $\alpha = 0.75/\sigma_1(X_0)$ as in Li et al. (2024), where $X_0 = \mathcal{T}_r(p^{-1}\mathcal{HP}_\Omega(x))$; SGD uses $\alpha = 0.5$ as in Tong et al. (2021); and JHGD uses $\alpha = 1.8$.

Figure 2 plots average runtime versus relative reconstruction error across different signal sizes (10 trials). In the large-scale case ($n_s = 49999$, yielding a $25000 \times 25000$ Hankel matrix), JHGD is fastest across all accuracy levels; for instance, it reaches $\text{RelErr} = 10^{-4}$ in about 3 s, which is about half the time of SGD and substantially faster than SHGD, PGD, and FIHT. Similar gains hold for $n_s = 9999$, indicating that the speedup persists across problem sizes.

*Table 1.* Average reconstruction error (RelErr), iteration count (Iter), and runtime (Time, in seconds) for five methods with $n_s = 9999$, $m = 5000$, and $r \in \{100, 300\}$.

| $r$ | 100 | | | 300 | | |
|------|--------|------|------|--------|------|-------|
| | RelErr | Iter | Time | RelErr | Iter | Time |
| PGD | 1.93e-5 | 34.7 | 2.48 | 4.06e-5 | 175.9 | 30.60 |
| SHGD | 1.93e-5 | 34.7 | 1.92 | 4.05e-5 | 175.9 | 15.99 |
| FIHT | 1.92e-5 | 8.1 | 3.25 | 6.42e-5 | 15.6 | 16.35 |
| SGD | 1.04e-5 | 19 | 1.64 | 1.54e-5 | 28.6 | 4.38 |
| JHGD | 5.32e-6 | 10.8 | **0.60** | 8.77e-6 | 19.1 | **2.13** |

*Table 2.* Average reconstruction error (RelErr), iteration count (Iter), and runtime (Time, in seconds) for five methods with $n_s = 9999$, $r = 100$, and $m \in \{3000, 4000\}$.

| $m$ | 3000 | | | 4000 | | |
|---|---|---|---|---|---|---|
| | RelErr | Iter | Time | RelErr | Iter | Time |
| PGD | 2.43e-5 | 78.4 | 5.64 | 1.96e-5 | 44.6 | 3.10 |
| SHGD | 2.43e-5 | 78.4 | 4.31 | 1.91e-5 | 44.7 | 2.50 |
| FIHT | 3.38e-5 | 13.3 | 5.32 | 2.93e-5 | 9.7 | 3.82 |
| SGD | 1.26e-5 | 24.1 | 1.99 | 1.11e-5 | 20.8 | 1.71 |
| JHGD | 5.68e-6 | 15.8 | **0.95** | 4.41e-6 | 12.6 | **0.76** |

*Table 3.* Average reconstruction error (RelErr), iteration count (Iter), and runtime (Time, in seconds) for five methods with $m = 5000$, $r = 200$, and $n_s \in \{9999, 49999\}$.

| $n_s$ | 9999 | | | 49999 | | |
|---|---|---|---|---|---|---|
| | RelErr | Iter | Time | RelErr | Iter | Time |
| PGD | 2.31e-5 | 81.2 | 10.95 | 2.53e-5 | 112.3 | 69.96 |
| SHGD | 2.31e-5 | 81.2 | 5.48 | 2.53e-5 | 112.3 | 37.60 |
| FIHT | 5.68e-5 | 11.6 | 8.25 | 2.54e-5 | 21.4 | 116.66 |
| SGD | 1.22e-5 | 23.7 | 2.78 | 1.50e-5 | 30.5 | 16.36 |
| JHGD | 6.53e-6 | 15.2 | **1.29** | 7.50e-6 | 20.6 | **8.18** |

Table 1 compares the performance of five algorithms across different sparsity levels ($n_s = 9999$, $m = 5000$), corresponding to a Hankel matrix of size $5000 \times 5000$. As sparsity increases from $r = 100$ to $r = 300$, all methods require more iterations and runtime. JHGD is the fastest and most accurate in both cases, running more than twice as fast as SGD while achieving substantially lower reconstruction error overall.

Table 2 summarizes performance as the number of observations increases. When $m$ grows from 3000 to 4000, all methods require fewer iterations. JHGD performs best in both settings, achieving reconstruction error on the order of $10^{-6}$ with runtime under 1 s. Relative to SGD, the strongest baseline, JHGD cuts runtime by over $50\%$ while maintaining higher accuracy.

Table 3 reports performance as the signal dimension increases. As $n_s$ grows from 9999 to 49999, runtime rises for all methods: PGD increases from about 11 s to about 70 s, while SHGD takes roughly half as long despite requiring the same number of iterations. FIHT uses the fewest iterations but becomes the slowest at large $n_s$ due to the cost of per-iteration SVDs. Among the baselines, SGD is the most efficient; nevertheless, JHGD performs best overall, running about $50\%$ faster than SGD while achieving the lowest reconstruction error in both settings.

## 4.2. Robustness to Additive Noise

We evaluate the robustness of JHGD to additive noise. We corrupt the observed signal entries with noise of the form

$$\boldsymbol{e} = \sigma \|\mathcal{P}_\Omega(\boldsymbol{x})\|_2 \cdot \frac{\boldsymbol{w}}{\|\boldsymbol{w}\|_2}, \tag{21}$$

where $\boldsymbol{x}$ is the target signal, $\sigma$ is the noise level, and $\boldsymbol{w}$ has i.i.d. standard Gaussian entries. We focus on undamped, non-separated signals with $n_s = 127$, $r = 12$, and $m = 120$. We test seven noise levels $\sigma \in [10^{-3}, 1]$, corresponding to signal-to-noise ratios (SNR) evenly spaced from $60\,\mathrm{dB}$ to $0\,\mathrm{dB}$.

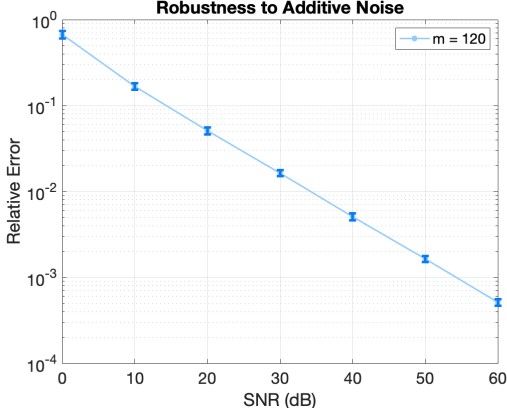

**Figure 3.** Robustness of JHGD to additive noise: mean reconstruction error (light blue) with standard deviation (dark blue bars) over 50 trials versus SNR.

Figure 3 shows that the relative reconstruction error decreases roughly linearly as the noise level decreases (i.e., as SNR). The light blue curve reports the mean relative error over 50 Monte Carlo trials, and the dark blue bars indicate the corresponding standard deviation, which remains small across all SNRs, demonstrating stable and robust performance of JHGD all noise levels considered.

## 5. Conclusions

We studied spectrally sparse signal recovery from partial samples via low-rank Hankel matrix completion. We proposed a Jacobi-preconditioned gradient descent algorithm that preserves low per-iteration complexity while achieving condition-numberindependent linear convergence. Using a generator that maps factor iterates to matrix space, we established equivalence to manifold-based updates, enabling matrix-space convergence analysis and avoiding distance ambiguities from complex-symmetric factorizations. Empirically, the method consistently improves both iteration count and runtime over state-of-the-art algorithms across a range of settings. These results indicate that well-founded preconditioning and factormanifold unification offer a promising route for efficient recovery in general low-rank matrix problems.

## Acknowledgement

This work was supported by the Hong Kong Research Grant Council (RGC) GRFs 16307325, 16306124, and 16307023, and the Hong Kong RGC Postdoctoral Fellowship Scheme of Project No. PDFS2425-6S05. We would also like to thank the anonymous reviewers for their valuable feedback on the manuscript.

## Impact Statement

This paper develops optimization methods for spectral signal reconstruction, with the goal of advancing the field of machine learning and signal processing. Improved reconstruction from fewer samples may benefit applications such as magnetic resonance spectroscopy and seismic imaging by reducing acquisition time and cost. We are not aware of specific ethical concerns that require discussion beyond standard considerations for general-purpose optimization methods.

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

## Appendix Contents

# A. Proofs of Results in Section 2

## A.1. Jacobi Operator and Its Derivatives

A key contribution of our work is a detailed analysis of the Jacobi operator $\mathcal{J}$ associated with the generator $\boldsymbol{X} = \mathcal{Q}(\boldsymbol{Z}) = \boldsymbol{Z}\boldsymbol{Z}^\top$, which serves as a principled link between the factor-space and matrix-space formulations. Building on this analysis, we derive explicit formulas for the adjoint $\mathcal{J}^*$ and the Moore–Penrose pseudoinverse $\mathcal{J}^\dagger$, and use them to design computationally efficient preconditioners that preserve the geometry of the Riemannian gradient while acting in the lower-dimensional factor space.

Recall the Jacobi operator (7) for the generator $\boldsymbol{X} = \mathcal{Q}(\boldsymbol{Z})$, the tangent space at $\boldsymbol{X} \in \mathbb{C}_S^{n \times n}$ is

$$\mathbb{T}_{\boldsymbol{X}} = \left\{ \boldsymbol{Z}\boldsymbol{M}^\top + \boldsymbol{M}\boldsymbol{Z}^\top \,|\, \forall \boldsymbol{M} \in \mathbb{C}^{n \times r} \right\}, \tag{22}$$

which is the range of the Jacobi operator.

We now can calculate the adjoint of $\mathcal{J}$ in (8),

$$\begin{aligned}
\langle \mathcal{J}(\boldsymbol{M}), \boldsymbol{Y} \rangle = \langle \boldsymbol{Z}\boldsymbol{M}^\top + \boldsymbol{M}\boldsymbol{Z}^\top, \boldsymbol{Y} \rangle &= \operatorname{tr}(\boldsymbol{Y}^{\mathsf{H}}\boldsymbol{Z}\boldsymbol{M}^\top) + \operatorname{tr}(\boldsymbol{Y}^{\mathsf{H}}\boldsymbol{M}\boldsymbol{Z}^\top) \\
&= \operatorname{tr}(\boldsymbol{M}\boldsymbol{Z}^\top \bar{\boldsymbol{Y}}) + \operatorname{tr}(\boldsymbol{Y}^{\mathsf{H}}\boldsymbol{M}\boldsymbol{Z}^\top) \\
&= \operatorname{tr}(\boldsymbol{Z}^\top \bar{\boldsymbol{Y}}\boldsymbol{M} + \boldsymbol{Z}^\top \boldsymbol{Y}^{\mathsf{H}}\boldsymbol{M}) \\
&= \langle \boldsymbol{M}, 2\boldsymbol{Y}\bar{\boldsymbol{Z}} \rangle \\
&= \langle \boldsymbol{M}, \mathcal{J}^*(\boldsymbol{Y}) \rangle.
\end{aligned}$$

So we have $\mathcal{J}^*(\boldsymbol{Y}) = 2\boldsymbol{Y}\bar{\boldsymbol{Z}}$ for any $\boldsymbol{Y} \in \mathbb{C}_S^{n \times n}$.

The Pseudoinverse $\mathcal{J}^\dagger$ of the Jacobi operator is defined as,

**Definition A.1** (Pseudoinverse, (Golub & Van Loan, 2013; Penrose, 1955)). For Jacobi operator $\mathcal{J} : \mathbb{C}^{n \times r} \to \mathbb{C}_S^{n \times n}$, the pseudoinverse of $\mathcal{J}$ is defined as $\mathcal{J}^\dagger : \mathbb{C}_S^{n \times n} \to \mathbb{C}^{n \times r}$ satisfying all of the following four criteria, known as the Moore-Penrose conditions:

1. $\mathcal{J}\mathcal{J}^\dagger$ need not be the general identity operator, but it maps $\mathcal{J}$ to itself:

$$\mathcal{J}\mathcal{J}^\dagger\mathcal{J} = \mathcal{J}.$$

2. $\mathcal{J}^\dagger$ acts like a weak inverse:

$$\mathcal{J}^\dagger\mathcal{J}\mathcal{J}^\dagger = \mathcal{J}^\dagger.$$

3. $\mathcal{J}\mathcal{J}^\dagger$ is idempotent:

$$\left(\mathcal{J}\mathcal{J}^\dagger\right)^* = \mathcal{J}\mathcal{J}^\dagger.$$

4. $\mathcal{J}^\dagger\mathcal{J}$ is also idempotent:

$$\left(\mathcal{J}^\dagger\mathcal{J}\right)^* = \mathcal{J}^\dagger\mathcal{J}.$$

## A.2. Proof of Theorem 2.1

*Proof.* To prove this theorem, we need to prove the following

$$\begin{cases} \mathcal{J}(\boldsymbol{V}_{ij}) = \sigma_{ij}\boldsymbol{U}_{ij} \\ \mathcal{J}^*(\boldsymbol{U}_{ij}) = \sigma_{ij}\boldsymbol{V}_{ij} \end{cases}$$

for three cases: $r \geq i > j$, $i > r \geq j$, $i = j$. Moreover, in the case $i > r \geq j$, $\boldsymbol{v}_i = \boldsymbol{0}$ and $\sigma_i = 0$, so the singular pairs will be degraded to

$$(\sigma_{ij}, \boldsymbol{U}_{ij}, \boldsymbol{V}_{ij}) = \left( \sqrt{2}\sigma_j, \frac{1}{\sqrt{2}}\boldsymbol{u}_i\boldsymbol{u}_j^\top + \frac{1}{\sqrt{2}}\boldsymbol{u}_j\boldsymbol{u}_i^\top, \boldsymbol{u}_i\boldsymbol{v}_j^\top \right). \tag{23}$$

Recall $\mathcal{J}(M) = ZM^\top + MZ^\top$, and $Z = \sum_{k=1}^{r} \sigma_k u_k v_k^{\mathsf{H}}$.

For $r \geq i > j$,

$$\mathcal{J}(V_{ij}) = \mathcal{J}\left(\sqrt{\frac{\sigma_j^2}{\sigma_i^2 + \sigma_j^2}} u_i v_j^\top + \sqrt{\frac{\sigma_i^2}{\sigma_i^2 + \sigma_j^2}} u_j v_i^\top\right)$$

$$= \left(\sum_{k=1}^{r} \sigma_k u_k v_k^{\mathsf{H}}\right)\left(\sqrt{\frac{\sigma_j^2}{\sigma_i^2 + \sigma_j^2}} u_i v_j^\top + \sqrt{\frac{\sigma_i^2}{\sigma_i^2 + \sigma_j^2}} u_j v_i^\top\right)^\top + \left(\sqrt{\frac{\sigma_j^2}{\sigma_i^2 + \sigma_j^2}} u_i v_j^\top + \sqrt{\frac{\sigma_i^2}{\sigma_i^2 + \sigma_j^2}} u_j v_i^\top\right)\left(\sum_{k=1}^{r} \sigma_k u_k v_k^{\mathsf{H}}\right)^\top$$

$$= \frac{\sigma_j^2}{\sqrt{\sigma_i^2 + \sigma_j^2}} u_j u_i^\top + \frac{\sigma_i^2}{\sqrt{\sigma_i^2 + \sigma_j^2}} u_i u_j^\top + \frac{\sigma_j^2}{\sqrt{\sigma_i^2 + \sigma_j^2}} u_i u_j^\top + \frac{\sigma_i^2}{\sqrt{\sigma_i^2 + \sigma_j^2}} u_j u_i^\top$$

$$= \sqrt{2\sigma_i^2 + 2\sigma_j^2}\left(\frac{1}{\sqrt{2}} u_i u_j^\top + \frac{1}{\sqrt{2}} u_j u_i^\top\right) = \sigma_{ij} U_{ij}.$$

For $i > r \geq j$,

$$\mathcal{J}(V_{ij}) = \mathcal{J}\left(u_i v_j^\top\right) = \left(\sum_{k=1}^{r} \sigma_k u_k v_k^{\mathsf{H}}\right)(u_i v_j^\top)^\top + (u_i v_j^\top)\left(\sum_{k=1}^{r} \sigma_k u_k v_k^{\mathsf{H}}\right)^\top = \sqrt{2}\sigma_j\left(\frac{1}{\sqrt{2}} u_j u_i^\top + \frac{1}{\sqrt{2}} u_i u_j^\top\right) = \sigma_{ij} U_{ij}.$$

For $i = j$,

$$\mathcal{J}(V_{ii}) = \mathcal{J}\left(u_i v_i^\top\right) = \left(\sum_{k=1}^{r} \sigma_k u_k v_k^{\mathsf{H}}\right)(u_i v_i^\top)^\top + u_i v_i^\top\left(\sum_{k=1}^{r} \sigma_k u_k v_k^{\mathsf{H}}\right)^\top = 2\sigma_i u_i u_i^\top = \sigma_{ii} U_{ii}.$$

Recall $\mathcal{J}^*(Y) = 2Y\bar{Z}$, and $Z = \sum_{k=1}^{r} \sigma_k u_k v_k^{\mathsf{H}}$.

For $r \geq i > j$,

$$\mathcal{J}^*(U_{ij}) = \mathcal{J}^*\left(\frac{1}{\sqrt{2}} u_i u_j^\top + \frac{1}{\sqrt{2}} u_j u_i^\top\right) = 2\left(\frac{1}{\sqrt{2}} u_i u_j^\top + \frac{1}{\sqrt{2}} u_j u_i^\top\right)\left(\sum_{k=1}^{r} \sigma_k \bar{u}_k v_k^\top\right)$$

$$= \sqrt{2\sigma_i^2 + 2\sigma_j^2}\left(\sqrt{\frac{\sigma_j^2}{\sigma_i^2 + \sigma_j^2}} u_i v_j^\top + \sqrt{\frac{\sigma_i^2}{\sigma_i^2 + \sigma_j^2}} u_j v_i^\top\right) = \sigma_{ij} V_{ij}.$$

For $i > r \geq j$,

$$\mathcal{J}^*(U_{ij}) = \mathcal{J}^*\left(\frac{1}{\sqrt{2}} u_i u_j^\top + \frac{1}{\sqrt{2}} u_j u_i^\top\right) = 2\left(\frac{1}{\sqrt{2}} u_i u_j^\top + \frac{1}{\sqrt{2}} u_j u_i^\top\right)\left(\sum_{k=1}^{r} \sigma_k \bar{u}_k v_k^\top\right) = \sqrt{2}\sigma_j u_i v_j^\top = \sigma_{ij} V_{ij}.$$

For $i = j$,

$$\mathcal{J}^*(U_{ii}) = \mathcal{J}^*\left(u_i u_i^\top\right) = 2\left(u_i u_i^\top\right)\left(\sum_{k=1}^{r} \sigma_k \bar{u}_k v_k^\top\right) = 2\sigma_i u_i v_i^\top = \sigma_{ii} V_{ii}.$$

Furthermore, it's easy to check $\langle U_{ii}, U_{kk}\rangle = \delta_{ik}$, $\langle V_{ii}, V_{kk}\rangle = \delta_{ik}$, $\langle U_{ij}, U_{kl}\rangle = \delta_{ij,kl}$, and $\langle V_{ij}, V_{kl}\rangle = \delta_{ij,kl}$. $\qquad\square$

### A.3. Proof of Theorem 2.2

For clarity and completeness, we first recall the full statement of the theorem.

**Theorem A.1.** *Recall the Jacobi operator $\mathcal{J}$ in (7) and its adjoint $\mathcal{J}^*$ in (8), apply Theorem 2.1, we have the following,*

$$\mathcal{J}(\boldsymbol{M}) = \sum_{\substack{(i,j):\min\{i,j\}\leq r \\ i\geq j}} \sigma_{ij} \langle \boldsymbol{M}, \boldsymbol{V}_{ij}\rangle \boldsymbol{U}_{ij} = \boldsymbol{Z}\boldsymbol{M}^\top + \boldsymbol{M}\boldsymbol{Z}^\top, \tag{24}$$

$$\mathcal{J}^*(\boldsymbol{Y}) = \sum_{\substack{(i,j):\min\{i,j\}\leq r \\ i\geq j}} \sigma_{ij} \langle \boldsymbol{Y}, \boldsymbol{U}_{ij}\rangle \boldsymbol{V}_{ij} = 2\boldsymbol{Y}\bar{\boldsymbol{Z}}, \tag{25}$$

$$\mathcal{J}^\dagger(\boldsymbol{Y}) = \sum_{\substack{(i,j):\min\{i,j\}\leq r \\ i\geq j}} \sigma_{ij}^{-1} \langle \boldsymbol{Y}, \boldsymbol{U}_{ij}\rangle \boldsymbol{V}_{ij}, \tag{26}$$

$$\mathcal{J}^*\mathcal{J}(\boldsymbol{M}) = \sum_{\substack{(i,j):\min\{i,j\}\leq r \\ i\geq j}} \sigma_{ij}^2 \langle \boldsymbol{M}, \boldsymbol{V}_{ij}\rangle \boldsymbol{V}_{ij}, \tag{27}$$

$$\mathcal{J}\mathcal{J}^*(\boldsymbol{Y}) = \sum_{\substack{(i,j):\min\{i,j\}\leq r \\ i\geq j}} \sigma_{ij}^2 \langle \boldsymbol{Y}, \boldsymbol{U}_{ij}\rangle \boldsymbol{U}_{ij}. \tag{28}$$

*Proof.* To establish these identities, we work with the basis introduced in Theorem 2.1. In particular, Theorem A.2 provides a complete and explicit description of the corresponding orthonormal bases for $\mathbb{C}^{n\times r}$ and $\mathbb{C}_S^{n\times n}$.

To prove (24), we expand an arbitrary matrix $\boldsymbol{M} \in \mathbb{C}^{n\times r}$ in the full basis

$$\{\boldsymbol{V}_{ij}\}_{\substack{i,j=1 \\ i>j}}^n \;\oplus\; \{\boldsymbol{V}_{ii}\} \;\oplus\; \{\widetilde{\boldsymbol{V}}_{ij}\}_{\substack{i,j=1 \\ i<j}}^n,$$

as introduced in Theorem A.2. Since $\widetilde{\boldsymbol{V}}_{ij} \in \ker(\mathcal{J})$, these components are annihilated by $\mathcal{J}$ and therefore do not contribute to the singular-pair characterization given in Theorem 2.1.

$$\mathcal{J}(\boldsymbol{M}) = \mathcal{J}\left(\sum_{(i,j):\min\{i,j\}\leq r} C_{ij}\boldsymbol{V}_{ij}\right) = \sum_{(i,j):\min\{i,j\}\leq r} C_{ij}\mathcal{J}\left(\boldsymbol{V}_{ij}\right) = \sum_{\substack{(i,j):\min\{i,j\}\leq r \\ i\geq j}} C_{ij}\sigma_{ij}\boldsymbol{U}_{ij} = \sum_{\substack{(i,j):\min\{i,j\}\leq r \\ i\geq j}} \sigma_{ij}\langle \boldsymbol{M}, \boldsymbol{V}_{ij}\rangle\boldsymbol{U}_{ij}$$

$$= \sum_{\substack{(i,j):\min\{i,j\}\leq r \\ i>j}} \langle \boldsymbol{M}, \sigma_j\boldsymbol{u}_i\boldsymbol{v}_j^\top + \sigma_i\boldsymbol{u}_j\boldsymbol{v}_i^\top\rangle \left(\boldsymbol{u}_i\boldsymbol{u}_j^\top + \boldsymbol{u}_j\boldsymbol{u}_i^\top\right) + \sum_{i=j\leq r} 2\sigma_i\langle \boldsymbol{M}, \boldsymbol{u}_i\boldsymbol{v}_i^\top\rangle\boldsymbol{u}_i\boldsymbol{u}_i^\top$$

$$= \sum_{\substack{(i,j):\min\{i,j\}\leq r \\ i>j}} \langle \boldsymbol{M}, \sigma_j\boldsymbol{u}_i\boldsymbol{v}_j^\top\rangle\boldsymbol{u}_i\boldsymbol{u}_j^\top + \sum_{\substack{(i,j):\min\{i,j\}\leq r \\ i>j}} \langle \boldsymbol{M}, \sigma_i\boldsymbol{u}_j\boldsymbol{v}_i^\top\rangle\boldsymbol{u}_j\boldsymbol{u}_i^\top$$

$$+ \sum_{\substack{(i,j):\min\{i,j\}\leq r \\ i>j}} \langle \boldsymbol{M}, \sigma_j\boldsymbol{u}_i\boldsymbol{v}_j^\top\rangle\boldsymbol{u}_j\boldsymbol{u}_i^\top + \sum_{\substack{(i,j):\min\{i,j\}\leq r \\ i>j}} \langle \boldsymbol{M}, \sigma_i\boldsymbol{u}_j\boldsymbol{v}_i^\top\rangle\boldsymbol{u}_i\boldsymbol{u}_j^\top + \sum_{i=j\leq r} 2\sigma_i\langle \boldsymbol{M}, \boldsymbol{u}_i\boldsymbol{v}_i^\top\rangle\boldsymbol{u}_i\boldsymbol{u}_i^\top$$

$$= \sum_{\substack{(i,j):\min\{i,j\}\leq r \\ i>j}} \langle \boldsymbol{M}, \sigma_j\boldsymbol{u}_i\boldsymbol{v}_j^\top\rangle\boldsymbol{u}_i\boldsymbol{u}_j^\top + \sum_{\substack{(i,j):\min\{i,j\}\leq r \\ i>j}} \langle \boldsymbol{M}, \sigma_i\boldsymbol{u}_j\boldsymbol{v}_i^\top\rangle\boldsymbol{u}_j\boldsymbol{u}_i^\top$$

$$+ \sum_{\substack{(i,j):\min\{i,j\}\leq r \\ i>j}} \langle \boldsymbol{M}^\top, \sigma_j\boldsymbol{v}_j\boldsymbol{u}_i^\top\rangle\boldsymbol{u}_j\boldsymbol{u}_i^\top + \sum_{\substack{(i,j):\min\{i,j\}\leq r \\ i>j}} \langle \boldsymbol{M}^\top, \sigma_i\boldsymbol{v}_i\boldsymbol{u}_j^\top\rangle\boldsymbol{u}_i\boldsymbol{u}_j^\top + \sum_{i=j\leq r} 2\sigma_i\langle \boldsymbol{M}, \boldsymbol{u}_i\boldsymbol{v}_i^\top\rangle\boldsymbol{u}_i\boldsymbol{u}_i^\top$$

$$= \sum_{(i,j):\min\{i,j\}\leq r} \sigma_j\langle \boldsymbol{M}, \boldsymbol{u}_i\boldsymbol{v}_j^\top\rangle\boldsymbol{u}_i\boldsymbol{u}_j^\top + \sum_{(i,j):\min\{i,j\}\leq r} \sigma_i\langle \boldsymbol{M}^\top, \boldsymbol{v}_i\boldsymbol{u}_j^\top\rangle\boldsymbol{u}_i\boldsymbol{u}_j^\top$$

$$= \left(\sum_{i=1}^n \boldsymbol{u}_i\boldsymbol{u}_i^\mathsf{H}\right)\boldsymbol{M}\left(\sum_{j=1}^r \sigma_j\bar{\boldsymbol{v}}_j\boldsymbol{u}_j^\top\right) + \left(\sum_{i=1}^r \sigma_i\boldsymbol{u}_i\boldsymbol{v}_i^\mathsf{H}\right)\boldsymbol{M}^\top\left(\sum_{j=1}^n \bar{\boldsymbol{u}}_j\boldsymbol{u}_j^\top\right) = \boldsymbol{M}\left(\sum_{j=1}^r \sigma_j\bar{\boldsymbol{v}}_j\boldsymbol{u}_j^\top\right) + \left(\sum_{i=1}^r \sigma_i\boldsymbol{u}_i\boldsymbol{v}_i^\mathsf{H}\right)\boldsymbol{M}^\top.$$

The second equality applies the linearity of Jacobi operator. The third equality only considers $i \geq j$ since $\mathcal{J}(\widetilde{\boldsymbol{V}}_{ij}) = \boldsymbol{0}$. The seventh equality uses the property of trace, $\langle \boldsymbol{M}, \boldsymbol{u}_j\boldsymbol{v}_i^\top\rangle = \langle \boldsymbol{M}^\top, \boldsymbol{v}_i\boldsymbol{u}_j^\top\rangle$.

Besides, $\mathcal{J}(\boldsymbol{M}) = \boldsymbol{Z}\boldsymbol{M}^\top + \boldsymbol{M}\boldsymbol{Z}^\top = \left(\sum_{k=1}^r \sigma_k\boldsymbol{u}_k\boldsymbol{v}_k^\mathsf{H}\right)\boldsymbol{M}^\top + \boldsymbol{M}\left(\sum_{l=1}^r \sigma_l\bar{\boldsymbol{v}}_l\boldsymbol{u}_l^\top\right)$. Thus, we have proved equation (24).

The next is to prove equation (25), we expand an arbitrary matrix $\boldsymbol{Y} \in \mathbb{C}_S^{n \times n}$ in the full basis

$$\{\boldsymbol{U}_{ij}\}_{\substack{i,j=1 \\ i>j}}^{n} \oplus \{\boldsymbol{U}_{ii}\}$$

as introduced in Theorem A.2.

$$\mathcal{J}^*(\boldsymbol{Y}) = \mathcal{J}^* \left( \sum_{(i,j):i \geq j} D_{ij} \boldsymbol{U}_{ij} \right) = \sum_{(i,j):i \geq j} D_{ij} \mathcal{J}^*(\boldsymbol{U}_{ij}) = \sum_{\substack{(i,j):\min\{i,j\} \leq r \\ i \geq j}} D_{ij} \sigma_{ij} \boldsymbol{V}_{ij} = \sum_{\substack{(i,j):\min\{i,j\} \leq r \\ i \geq j}} \sigma_{ij} \langle \boldsymbol{Y}, \boldsymbol{U}_{ij} \rangle \boldsymbol{V}_{ij}$$

$$= \sum_{\substack{(i,j):\min\{i,j\} \leq r \\ i>j}} \sqrt{2\sigma_i^2 + 2\sigma_j^2} \left\langle \boldsymbol{Y}, \frac{1}{\sqrt{2}} \boldsymbol{u}_i \boldsymbol{u}_j^\top + \frac{1}{\sqrt{2}} \boldsymbol{u}_j \boldsymbol{u}_i^\top \right\rangle \cdot \left( \sqrt{\frac{\sigma_j^2}{\sigma_i^2 + \sigma_j^2}} \boldsymbol{u}_i \boldsymbol{v}_j^\top + \sqrt{\frac{\sigma_i^2}{\sigma_i^2 + \sigma_j^2}} \boldsymbol{u}_j \boldsymbol{v}_i^\top \right) + \sum_{i=j \leq r} 2\sigma_i \langle \boldsymbol{Y}, \boldsymbol{u}_i \boldsymbol{u}_i^\top \rangle \boldsymbol{u}_i \boldsymbol{v}_i^\top$$

$$= \sum_{\substack{(i,j):\min\{i,j\} \leq r \\ i>j}} \sigma_j \langle \boldsymbol{Y}, \boldsymbol{u}_i \boldsymbol{u}_j^\top \rangle \boldsymbol{u}_i \boldsymbol{v}_j^\top + \sum_{\substack{(i,j):\min\{i,j\} \leq r \\ i>j}} \sigma_i \langle \boldsymbol{Y}, \boldsymbol{u}_j \boldsymbol{u}_i^\top \rangle \boldsymbol{u}_j \boldsymbol{v}_i^\top$$

$$+ \sum_{\substack{(i,j):\min\{i,j\} \leq r \\ i>j}} \sigma_j \langle \boldsymbol{Y}, \boldsymbol{u}_j \boldsymbol{u}_i^\top \rangle \boldsymbol{u}_i \boldsymbol{v}_j^\top + \sum_{\substack{(i,j):\min\{i,j\} \leq r \\ i>j}} \sigma_i \langle \boldsymbol{Y}, \boldsymbol{u}_i \boldsymbol{u}_j^\top \rangle \boldsymbol{u}_j \boldsymbol{v}_i^\top + \sum_{i=j \leq r} 2\sigma_i \langle \boldsymbol{Y}, \boldsymbol{u}_i \boldsymbol{u}_i^\top \rangle \boldsymbol{u}_i \boldsymbol{v}_i^\top$$

$$= \sum_{\substack{(i,j):\min\{i,j\} \leq r \\ i>j}} \sigma_j \langle \boldsymbol{Y}, \boldsymbol{u}_i \boldsymbol{u}_j^\top \rangle \boldsymbol{u}_i \boldsymbol{v}_j^\top + \sum_{\substack{(i,j):\min\{i,j\} \leq r \\ i>j}} \sigma_i \langle \boldsymbol{Y}, \boldsymbol{u}_j \boldsymbol{u}_i^\top \rangle \boldsymbol{u}_j \boldsymbol{v}_i^\top$$

$$+ \sum_{\substack{(i,j):\min\{i,j\} \leq r \\ i>j}} \sigma_j \langle \boldsymbol{Y}, \boldsymbol{u}_i \boldsymbol{u}_j^\top \rangle \boldsymbol{u}_i \boldsymbol{v}_j^\top + \sum_{\substack{(i,j):\min\{i,j\} \leq r \\ i>j}} \sigma_i \langle \boldsymbol{Y}, \boldsymbol{u}_j \boldsymbol{u}_i^\top \rangle \boldsymbol{u}_j \boldsymbol{v}_i^\top + \sum_{i=j \leq r} 2\sigma_i \langle \boldsymbol{Y}, \boldsymbol{u}_i \boldsymbol{u}_i^\top \rangle \boldsymbol{u}_i \boldsymbol{v}_i^\top$$

$$= \sum_{(i,j):\min\{i,j\} \leq r} \sigma_j \langle \boldsymbol{Y}, \boldsymbol{u}_i \boldsymbol{u}_j^\top \rangle \boldsymbol{u}_i \boldsymbol{v}_j^\top + \sum_{(i,j):\min\{i,j\} \leq r} \sigma_i \langle \boldsymbol{Y}, \boldsymbol{u}_j \boldsymbol{u}_i^\top \rangle \boldsymbol{u}_j \boldsymbol{v}_i^\top$$

$$= \sum_{i=1}^{n} \boldsymbol{u}_i \boldsymbol{u}_i^\mathsf{H} \boldsymbol{Y} \sum_{j=1}^{r} \sigma_j \bar{\boldsymbol{u}}_j \boldsymbol{v}_j^\top + \sum_{j=1}^{n} \boldsymbol{u}_j \boldsymbol{u}_j^\mathsf{H} \boldsymbol{Y} \sum_{i=1}^{r} \sigma_i \bar{\boldsymbol{u}}_i \boldsymbol{v}_i^\top = 2\boldsymbol{Y} \sum_{j=1}^{r} \sigma_j \bar{\boldsymbol{u}}_j \boldsymbol{v}_j^\top.$$

The second equality applies the linearity of the Jacobi operator. The third equality makes use of singular pairs of Jacobi. The seventh equality uses the symmetry of $\boldsymbol{Y}$ and, the property of trace $\langle \boldsymbol{Y}, \boldsymbol{u}_j \boldsymbol{u}_i^\top \rangle = \langle \boldsymbol{Y}, \boldsymbol{u}_i \boldsymbol{u}_j^\top \rangle$.

Besides, $\mathcal{J}^*(\boldsymbol{Y}) = 2\boldsymbol{Y}\bar{\boldsymbol{Z}} = 2\boldsymbol{Y} \left( \sum_{k=1}^{r} \sigma_k \bar{\boldsymbol{u}}_k \boldsymbol{v}_k^\top \right)$. Thus, we prove equation (25).

Next, we prove that (26) is a valid pseudoinverse by checking 4 conditions in Definition A.1.
1. Prove $\mathcal{J} \mathcal{J}^\dagger \mathcal{J}(\boldsymbol{M}) = \mathcal{J}(\boldsymbol{M}) \; \forall \boldsymbol{M} \in \mathbb{C}^{n \times r}$.

$$\mathcal{J}\mathcal{J}^\dagger \mathcal{J}(\boldsymbol{M}) = \mathcal{J}\mathcal{J}^\dagger \left( \sum_{\substack{(i,j):\min\{i,j\} \leq r \\ i \geq j}} \sigma_{ij} \langle \boldsymbol{M}, \boldsymbol{V}_{ij} \rangle \boldsymbol{U}_{ij} \right) = \mathcal{J} \left( \sum_{\substack{(k,l):\min\{k,l\} \leq r \\ k \geq l}} \sigma_{kl}^{-1} \left\langle \sum_{\substack{(i,j):\min\{i,j\} \leq r \\ i \geq j}} \sigma_{ij} \langle \boldsymbol{M}, \boldsymbol{V}_{ij} \rangle \boldsymbol{U}_{ij}, \boldsymbol{U}_{kl} \right\rangle \boldsymbol{V}_{kl} \right)$$

$$= \mathcal{J} \left( \sum_{\substack{(k,l):\min\{k,l\} \leq r \\ k \geq l}} \langle \boldsymbol{M}, \boldsymbol{V}_{kl} \rangle \boldsymbol{V}_{kl} \right) = \sum_{\substack{(m,n):\min\{m,n\} \leq r \\ m \geq n}} \sigma_{mn} \left\langle \sum_{\substack{(k,l):\min\{k,l\} \leq r \\ k \geq l}} \langle \boldsymbol{M}, \boldsymbol{V}_{kl} \rangle \boldsymbol{V}_{kl}, \boldsymbol{V}_{mn} \right\rangle \boldsymbol{U}_{mn}$$

$$= \sum_{\substack{(m,n):\min\{m,n\} \leq r \\ m \geq n}} \sigma_{mn} \langle \boldsymbol{M}, \boldsymbol{V}_{mn} \rangle \boldsymbol{U}_{mn} = \mathcal{J}(\boldsymbol{M}).$$

2. Prove $\mathcal{J}^\dagger \mathcal{J} \mathcal{J}^\dagger(\boldsymbol{Y}) = \mathcal{J}^\dagger(\boldsymbol{Y}) \; \forall \boldsymbol{Y} \in \mathbb{C}_S^{n \times n}$.

$$\mathcal{J}^\dagger \mathcal{J} \mathcal{J}^\dagger(\boldsymbol{Y}) = \mathcal{J}^\dagger \mathcal{J} \left( \sum_{\substack{(i,j):\min\{i,j\} \leq r \\ i \geq j}} \sigma_{ij}^{-1} \langle \boldsymbol{Y}, \boldsymbol{U}_{ij} \rangle \boldsymbol{V}_{ij} \right) = \mathcal{J}^\dagger \left( \sum_{\substack{(k,l):\min\{k,l\} \leq r \\ k \geq l}} \sigma_{kl} \left\langle \sum_{\substack{(i,j):\min\{i,j\} \leq r \\ i \geq j}} \sigma_{ij}^{-1} \langle \boldsymbol{Y}, \boldsymbol{U}_{ij} \rangle \boldsymbol{V}_{ij}, \boldsymbol{V}_{kl} \right\rangle \boldsymbol{U}_{kl} \right)$$

$$= \mathcal{J}^\dagger \left( \sum_{\substack{(k,l):\min\{k,l\}\leq r \\ k \geq l}} \langle \boldsymbol{Y}, \boldsymbol{U}_{kl} \rangle \boldsymbol{U}_{kl} \right) = \sum_{\substack{(m,n):\min\{m,n\}\leq r \\ m \geq n}} \sigma_{mn}^{-1} \left\langle \sum_{\substack{(k,l):\min\{k,l\}\leq r \\ k \geq l}} \langle \boldsymbol{Y}, \boldsymbol{U}_{kl} \rangle \boldsymbol{U}_{kl}, \boldsymbol{U}_{mn} \right\rangle \boldsymbol{V}_{mn}$$

$$= \sum_{\substack{(m,n):\min\{m,n\}\leq r \\ m \geq n}} \sigma_{mn}^{-1} \langle \boldsymbol{Y}, \boldsymbol{U}_{mn} \rangle \boldsymbol{V}_{mn} = \mathcal{J}^\dagger(\boldsymbol{M}).$$

3. Prove $\langle \mathcal{J}^\dagger \mathcal{J}(\boldsymbol{M}), \boldsymbol{Q} \rangle = \langle \boldsymbol{M}, \mathcal{J}^\dagger \mathcal{J}(\boldsymbol{Q}) \rangle \ \forall \boldsymbol{M}, \boldsymbol{Q} \in \mathbb{C}^{n \times r}$.

From the first part of the proof, we know $\mathcal{J}^\dagger \mathcal{J}(\boldsymbol{M}) = \sum_{\substack{(i,j):\min\{i,j\}\leq r \\ i \geq j}} \langle \boldsymbol{M}, \boldsymbol{V}_{ij} \rangle \boldsymbol{V}_{ij}$,

$$\text{LHS} = \left\langle \sum_{\substack{(i,j):\min\{i,j\}\leq r \\ i \geq j}} \langle \boldsymbol{M}, \boldsymbol{V}_{ij} \rangle \boldsymbol{V}_{ij}, \boldsymbol{Q} \right\rangle = \operatorname{tr}\left( \boldsymbol{Q}^{\mathsf{H}} \sum_{\substack{(i,j):\min\{i,j\}\leq r \\ i \geq j}} \langle \boldsymbol{M}, \boldsymbol{V}_{ij} \rangle \boldsymbol{V}_{ij} \right)$$

$$= \sum_{\substack{(i,j):\min\{i,j\}\leq r \\ i \geq j}} \langle \boldsymbol{M}, \boldsymbol{V}_{ij} \rangle \operatorname{tr}(\boldsymbol{Q}^{\mathsf{H}} \boldsymbol{V}_{ij}) = \sum_{\substack{(i,j):\min\{i,j\}\leq r \\ i \geq j}} \operatorname{tr}(\boldsymbol{V}_{ij}^{\mathsf{H}} \boldsymbol{M}) \operatorname{tr}(\boldsymbol{Q}^{\mathsf{H}} \boldsymbol{V}_{ij})$$

$$\text{RHS} = \left\langle \boldsymbol{M}, \sum_{\substack{(i,j):\min\{i,j\}\leq r \\ i \geq j}} \langle \boldsymbol{Q}, \boldsymbol{V}_{ij} \rangle \boldsymbol{V}_{ij} \right\rangle = \operatorname{tr}\left\{ \left( \sum_{\substack{(i,j):\min\{i,j\}\leq r \\ i \geq j}} \langle \boldsymbol{Q}, \boldsymbol{V}_{ij} \rangle \boldsymbol{V}_{ij} \right)^{\mathsf{H}} \boldsymbol{M} \right\}$$

$$= \sum_{\substack{(i,j):\min\{i,j\}\leq r \\ i \geq j}} \overline{\langle \boldsymbol{Q}, \boldsymbol{V}_{ij} \rangle} \operatorname{tr}(\boldsymbol{V}_{ij}^{\mathsf{H}} \boldsymbol{M}) = \sum_{\substack{(i,j):\min\{i,j\}\leq r \\ i \geq j}} \operatorname{tr}(\boldsymbol{V}_{ij}^{\top} \bar{\boldsymbol{Q}}) \operatorname{tr}(\boldsymbol{V}_{ij}^{\mathsf{H}} \boldsymbol{M})$$

$$= \sum_{\substack{(i,j):\min\{i,j\}\leq r \\ i \geq j}} \operatorname{tr}(\boldsymbol{Q}^{\mathsf{H}} \boldsymbol{V}_{ij}) \operatorname{tr}(\boldsymbol{V}_{ij}^{\mathsf{H}} \boldsymbol{M}) = \text{LHS}.$$

Therefore, we proved $(\mathcal{J}^\dagger \mathcal{J})^* = \mathcal{J}^\dagger \mathcal{J}$.

4. Prove $\langle \mathcal{J} \mathcal{J}^\dagger(\boldsymbol{Y}), \boldsymbol{W} \rangle = \langle \boldsymbol{Y}, \mathcal{J} \mathcal{J}^\dagger(\boldsymbol{W}) \rangle \ \forall \boldsymbol{Y}, \boldsymbol{W} \in \mathbb{C}_S^{n \times n}$.

From the second part of the proof, we know $\mathcal{J} \mathcal{J}^\dagger(\boldsymbol{Y}) = \sum_{\substack{(i,j):\min\{i,j\}\leq r \\ i \geq j}} \langle \boldsymbol{Y}, \boldsymbol{U}_{ij} \rangle \boldsymbol{U}_{ij}$,

$$\text{LHS} = \left\langle \sum_{\substack{(i,j):\min\{i,j\}\leq r \\ i \geq j}} \langle \boldsymbol{Y}, \boldsymbol{U}_{ij} \rangle \boldsymbol{U}_{ij}, \boldsymbol{W} \right\rangle = \operatorname{tr}\left( \boldsymbol{W}^{\mathsf{H}} \sum_{\substack{(i,j):\min\{i,j\}\leq r \\ i \geq j}} \langle \boldsymbol{Y}, \boldsymbol{U}_{ij} \rangle \boldsymbol{U}_{ij} \right)$$

$$= \sum_{\substack{(i,j):\min\{i,j\}\leq r \\ i \geq j}} \langle \boldsymbol{Y}, \boldsymbol{U}_{ij} \rangle \operatorname{tr}(\boldsymbol{W}^{\mathsf{H}} \boldsymbol{U}_{ij}) = \sum_{\substack{(i,j):\min\{i,j\}\leq r \\ i \geq j}} \operatorname{tr}(\boldsymbol{U}_{ij}^{\mathsf{H}} \boldsymbol{Y}) \operatorname{tr}(\boldsymbol{W}^{\mathsf{H}} \boldsymbol{U}_{ij})$$

$$\text{RHS} = \left\langle \boldsymbol{Y}, \sum_{\substack{(i,j):\min\{i,j\}\leq r \\ i \geq j}} \langle \boldsymbol{W}, \boldsymbol{U}_{ij} \rangle \boldsymbol{U}_{ij} \right\rangle = \operatorname{tr}\left\{ \left( \sum_{\substack{(i,j):\min\{i,j\}\leq r \\ i \geq j}} \langle \boldsymbol{W}, \boldsymbol{U}_{ij} \rangle \boldsymbol{U}_{ij} \right)^{\mathsf{H}} \boldsymbol{Y} \right\}$$

$$= \sum_{\substack{(i,j):\min\{i,j\}\leq r \\ i \geq j}} \overline{\langle \boldsymbol{W}, \boldsymbol{U}_{ij} \rangle} \operatorname{tr}(\boldsymbol{U}_{ij}^{\mathsf{H}} \boldsymbol{Y}) = \sum_{\substack{(i,j):\min\{i,j\}\leq r \\ i \geq j}} \operatorname{tr}(\overline{\boldsymbol{U}_{ij}^{\mathsf{H}} \boldsymbol{W}}) \operatorname{tr}(\boldsymbol{U}_{ij}^{\mathsf{H}} \boldsymbol{Y})$$

$$= \sum_{\substack{(i,j):\min\{i,j\}\leq r \\ i \geq j}} \operatorname{tr}(\boldsymbol{U}_{ij}^{\top} \overline{\boldsymbol{W}}) \operatorname{tr}(\boldsymbol{U}_{ij}^{\mathsf{H}} \boldsymbol{Y}) = \sum_{\substack{(i,j):\min\{i,j\}\leq r \\ i \geq j}} \operatorname{tr}(\boldsymbol{W}^{\mathsf{H}} \boldsymbol{U}_{ij}) \operatorname{tr}(\boldsymbol{U}_{ij}^{\mathsf{H}} \boldsymbol{Y}).$$

Therefore, we have proved LHS = RHS, that is, $(\mathcal{J} \mathcal{J}^\dagger)^* = \mathcal{J} \mathcal{J}^\dagger$.

We finally prove (26) is a pseudoinverse.

Equations (27) and (28) can be calculated using the expressions in (24) and (25).  □

### A.4. Proof of Corollary 2.1

*Proof.* From Theorem 2.2 equation (27), we have

$$\mathcal{J}^*\mathcal{J}(\boldsymbol{M}) = \sum_{\substack{(i,j):\min\{i,j\}\leq r \\ i \geq j}} \sigma_{ij}^2 \langle \boldsymbol{M}, \boldsymbol{V}_{ij} \rangle \, \boldsymbol{V}_{ij},$$

then we have

$$(\mathcal{J}^*\mathcal{J})^\dagger(\boldsymbol{M}) = \sum_{\substack{(i,j):\min\{i,j\}\leq r \\ i \geq j}} \sigma_{ij}^{-2} \langle \boldsymbol{M}, \boldsymbol{V}_{ij} \rangle \, \boldsymbol{V}_{ij}.$$

Then by checking 4 conditions in Definition A.1, it's easy to prove this is a valid pseudoinverse. Then, we calculate the following,

$$(\mathcal{J}^*\mathcal{J})^\dagger\mathcal{J}^*(\boldsymbol{Y}) = (\mathcal{J}^*\mathcal{J})^\dagger \left( \sum_{\substack{(k,l):\min\{k,l\}\leq r \\ k \geq l}} \sigma_{kl} \langle \boldsymbol{Y}, \boldsymbol{U}_{kl} \rangle \, \boldsymbol{V}_{kl} \right) = \sum_{\substack{(i,j):\min\{i,j\}\leq r \\ i \geq j}} \sigma_{ij}^{-2} \left\langle \sum_{\substack{(k,l):\min\{k,l\}\leq r \\ k \geq l}} \sigma_{kl} \langle \boldsymbol{Y}, \boldsymbol{U}_{kl} \rangle \, \boldsymbol{V}_{kl}, \boldsymbol{V}_{ij} \right\rangle \boldsymbol{V}_{ij}$$

$$= \sum_{\substack{(i,j):\min\{i,j\}\leq r \\ i \geq j}} \sigma_{ij}^{-2} \langle \sigma_{ij} \langle \boldsymbol{Y}, \boldsymbol{U}_{ij} \rangle \, \boldsymbol{V}_{ij}, \boldsymbol{V}_{ij} \rangle \, \boldsymbol{V}_{ij} = \sum_{\substack{(i,j):\min\{i,j\}\leq r \\ i \geq j}} \sigma_{ij}^{-1} \langle \boldsymbol{Y}, \boldsymbol{U}_{ij} \rangle \, \boldsymbol{V}_{ij} = \mathcal{J}^\dagger(\boldsymbol{Y}).$$

□

### A.5. Proof of Theorem 2.3

*Proof.* We firstly expand the term,

$$\frac{1}{2(\sigma_i^2 + \sigma_j^2)^2} \left( \sigma_j \boldsymbol{u}_i^\mathsf{H} \boldsymbol{M} \bar{\boldsymbol{v}}_j + \sigma_i \boldsymbol{u}_j^\mathsf{H} \boldsymbol{M} \bar{\boldsymbol{v}}_i \right) \left( \sigma_j \boldsymbol{u}_i \boldsymbol{v}_j^\top + \sigma_i \boldsymbol{u}_j \boldsymbol{v}_i^\top \right)$$

$$= \frac{\sigma_j^2}{2 \left( \sigma_i^2 + \sigma_j^2 \right)^2} \boldsymbol{u}_i \boldsymbol{u}_i^\mathsf{H} \boldsymbol{M} \bar{\boldsymbol{v}}_j \boldsymbol{v}_j^\top + \frac{\sigma_i^2}{2 \left( \sigma_i^2 + \sigma_j^2 \right)^2} \boldsymbol{u}_j \boldsymbol{u}_j^\mathsf{H} \boldsymbol{M} \bar{\boldsymbol{v}}_i \boldsymbol{v}_i^\top$$

$$+ \frac{\sigma_i \sigma_j}{2 \left( \sigma_i^2 + \sigma_j^2 \right)^2} \boldsymbol{u}_i \boldsymbol{u}_j^\mathsf{H} \boldsymbol{M} \bar{\boldsymbol{v}}_i \boldsymbol{v}_j^\top + \frac{\sigma_i \sigma_j}{2 \left( \sigma_i^2 + \sigma_j^2 \right)^2} \boldsymbol{u}_j \boldsymbol{u}_i^\mathsf{H} \boldsymbol{M} \bar{\boldsymbol{v}}_j \boldsymbol{v}_i^\top$$

$$:= A + B + C + D.$$

Since $\boldsymbol{M}$ is uniformly distributed in $\boldsymbol{u}_i$ and $\boldsymbol{v}_j$, for $i \neq j$,

$$\boldsymbol{u}_i \boldsymbol{u}_i^\mathsf{H} \left( \boldsymbol{u}_i \boldsymbol{u}_j^\mathsf{H} \boldsymbol{M} \bar{\boldsymbol{v}}_i \boldsymbol{v}_j^\top \right) \bar{\boldsymbol{v}}_j \boldsymbol{v}_j^\top = \boldsymbol{u}_i \boldsymbol{u}_j^\mathsf{H} \boldsymbol{M} \bar{\boldsymbol{v}}_i \boldsymbol{v}_j^\top,$$

$$\boldsymbol{u}_j \boldsymbol{u}_j^\mathsf{H} \left( \boldsymbol{u}_i \boldsymbol{u}_j^\mathsf{H} \boldsymbol{M} \bar{\boldsymbol{v}}_i \boldsymbol{v}_j^\top \right) \bar{\boldsymbol{v}}_i \boldsymbol{v}_i^\top = \boldsymbol{0}.$$

So the term $\boldsymbol{u}_i \boldsymbol{u}_j^\mathsf{H} \boldsymbol{M} \bar{\boldsymbol{v}}_i \boldsymbol{v}_j^\top$ can be totally projected onto $\boldsymbol{u}_i \boldsymbol{u}_i^\mathsf{H}(\cdot) \bar{\boldsymbol{v}}_j \boldsymbol{v}_j^\top$ space.
Similarly, $\boldsymbol{u}_j \boldsymbol{u}_i^\mathsf{H} \boldsymbol{M} \bar{\boldsymbol{v}}_j \boldsymbol{v}_i^\top$ can be totally projected onto $\boldsymbol{u}_j \boldsymbol{u}_j^\mathsf{H}(\cdot) \bar{\boldsymbol{v}}_i \boldsymbol{v}_i^\top$ space.

Denote

$$E := \frac{1}{4\sigma_j^2} \boldsymbol{u}_i \boldsymbol{u}_i^\mathsf{H} \boldsymbol{M} \bar{\boldsymbol{v}}_j \boldsymbol{v}_j^\top, \qquad F := \frac{1}{4\sigma_i^2} \boldsymbol{u}_j \boldsymbol{u}_j^\mathsf{H} \boldsymbol{M} \bar{\boldsymbol{v}}_i \boldsymbol{v}_i^\top.$$

And denote the coefficient before each term as $C_i$, where $i \in \{A, B, C, D, E, F\}$. Then,

$$
\begin{aligned}
& \mathbb{E}_{\boldsymbol{M}}[\|A + B + C + D - E - F\|_F^2] \\
& \leq 2\mathbb{E}_{\boldsymbol{M}}[\|A + C - E\|_F^2] + 2\mathbb{E}_{\boldsymbol{M}}[\|B + D - F\|_F^2] \\
& \leq 4\left((C_A - C_E)^2 + C_C^2 + (C_B - C_F)^2 + C_D^2\right)\mathbb{E}_{\boldsymbol{M}}[\|\boldsymbol{M}\|_2^2] \\
& = \left(\frac{1}{4\sigma_i^4} + \frac{1}{4\sigma_j^4} - \frac{1}{(\sigma_i^2 + \sigma_j^2)^2}\right)\mathbb{E}_{\boldsymbol{M}}[\|\boldsymbol{M}\|_2^2] \\
& \leq \frac{1}{2\sigma_{\min}^4(\boldsymbol{Z})}\mathbb{E}_{\boldsymbol{M}}\left[\|\boldsymbol{M}\|_2^2\right],
\end{aligned}
$$

where $\sigma_{\min}(\boldsymbol{Z})$ is the minimum singular value of $\boldsymbol{Z}$; $\boldsymbol{u}_i$ and $\boldsymbol{v}_j$ are singular vectors of $\boldsymbol{Z}$.

$\square$

Combined with the case of $i = j$, we finally obtain,

$$
\begin{aligned}
& \sum_{i=1}^r \frac{1}{4\sigma_i^2}(\boldsymbol{u}_i^{\mathsf{H}}\boldsymbol{M}\bar{\boldsymbol{v}}_i)\boldsymbol{u}_i\boldsymbol{v}_i^{\top} + \sum_{\substack{i,j=1 \\ i>j}}^r \frac{(\sigma_j\boldsymbol{u}_i^{\mathsf{H}}\boldsymbol{M}\bar{\boldsymbol{v}}_j + \sigma_i\boldsymbol{u}_j^{\mathsf{H}}\boldsymbol{M}\bar{\boldsymbol{v}}_i)}{2(\sigma_i^2 + \sigma_j^2)^2}\left(\sigma_j\boldsymbol{u}_i\boldsymbol{v}_j^{\top} + \sigma_i\boldsymbol{u}_j\boldsymbol{v}_i^{\top}\right) \\
& \approx \sum_{i=1}^r \frac{1}{4\sigma_i^2}(\boldsymbol{u}_i^{\mathsf{H}}\boldsymbol{M}\bar{\boldsymbol{v}}_i)\boldsymbol{u}_i\boldsymbol{v}_i^{\top} + \sum_{\substack{i,j=1 \\ i>j}}^r \left\{\frac{1}{4\sigma_j^2}\boldsymbol{u}_i\boldsymbol{u}_i^{\mathsf{H}}\boldsymbol{M}\bar{\boldsymbol{v}}_j\boldsymbol{v}_j^{\top} + \frac{1}{4\sigma_i^2}\boldsymbol{u}_j\boldsymbol{u}_j^{\mathsf{H}}\boldsymbol{M}\bar{\boldsymbol{v}}_i\boldsymbol{v}_i^{\top}\right\} \\
& = \frac{1}{4}\left(\sum_{i=1}^r \boldsymbol{u}_i\boldsymbol{u}_i^{\mathsf{H}}\right)\boldsymbol{M}\left(\sum_{j=1}^r \sigma_j^{-2}\bar{\boldsymbol{v}}_j\boldsymbol{v}_j^{\top}\right) \\
& = \frac{1}{4}\boldsymbol{Z}(\boldsymbol{Z}^{\mathsf{H}}\boldsymbol{Z})^{-1}\boldsymbol{Z}^{\mathsf{H}}\boldsymbol{M}(\overline{\boldsymbol{Z}^{\mathsf{H}}\boldsymbol{Z}})^{-1}.
\end{aligned}
$$

### A.6. Proof of Theorem 2.4

*Proof.* Preconditioners $\mathcal{P}_E$ and $\mathcal{P}_A$ induce the same orthogonal projection onto the tangent space. We have shown in (11) that $\mathcal{J}\mathcal{P}_E\mathcal{J}^* = \mathcal{J}\mathcal{J}^{\dagger}$. We now focus on the proof that $\mathcal{J}\mathcal{P}_E\mathcal{J}^* = \mathcal{J}\mathcal{P}_A\mathcal{J}^*$. For any $\boldsymbol{W}$ in $\mathbb{C}_S^{n\times n}$, one has

$$
\begin{aligned}
& \mathcal{J}\mathcal{P}_E\mathcal{J}^*(\boldsymbol{W}) = \mathcal{J}(\mathcal{J}^*\mathcal{J})^{\dagger}\mathcal{J}^*(\boldsymbol{W}) = \sum_{\substack{(i,j):\min\{i,j\}\leq r \\ i\geq j}} \langle\boldsymbol{W}, \boldsymbol{U}_{ij}\rangle\boldsymbol{U}_{ij} \\
& = \frac{1}{2}\sum_{\substack{(i,j):\min\{i,j\}\leq r \\ i>j}} \langle\boldsymbol{W}, \boldsymbol{u}_i\boldsymbol{u}_j^{\top} + \boldsymbol{u}_j\boldsymbol{u}_i^{\top}\rangle(\boldsymbol{u}_i\boldsymbol{u}_j^{\top} + \boldsymbol{u}_j\boldsymbol{u}_i^{\top}) + \sum_{i=j=1}^r \langle\boldsymbol{W}, \boldsymbol{u}_i\boldsymbol{u}_i^{\top}\rangle\boldsymbol{u}_i\boldsymbol{u}_i^{\top} \\
& = \sum_{\substack{(i,j):\min\{i,j\}\leq r \\ i>j}} \left\{\boldsymbol{u}_i\boldsymbol{u}_i^{\mathsf{H}}\boldsymbol{W}\bar{\boldsymbol{u}}_j\boldsymbol{u}_j^{\top} + \boldsymbol{u}_j\boldsymbol{u}_j^{\mathsf{H}}\boldsymbol{W}\bar{\boldsymbol{u}}_i\boldsymbol{u}_i^{\top}\right\} + \sum_{i=j=1}^r \langle\boldsymbol{W}, \boldsymbol{u}_i\boldsymbol{u}_i^{\top}\rangle\boldsymbol{u}_i\boldsymbol{u}_i^{\top} \\
& = \sum_{\substack{(i,j):\min\{i,j\}\leq r \\ i>j}} \boldsymbol{u}_i\boldsymbol{u}_i^{\mathsf{H}}\boldsymbol{W}\bar{\boldsymbol{u}}_j\boldsymbol{u}_j^{\top} + \sum_{\substack{(i,j):\min\{i,j\}\leq r \\ i<j}} \boldsymbol{u}_i\boldsymbol{u}_i^{\mathsf{H}}\boldsymbol{W}\bar{\boldsymbol{u}}_j\boldsymbol{u}_j^{\top} + \sum_{i=j=1}^r \boldsymbol{u}_i\boldsymbol{u}_i^{\mathsf{H}}\boldsymbol{W}\bar{\boldsymbol{u}}_i\boldsymbol{u}_i^{\top} \\
& = \sum_{(i,j):\min\{i,j\}\leq r} \boldsymbol{u}_i\boldsymbol{u}_i^{\mathsf{H}}\boldsymbol{W}\bar{\boldsymbol{u}}_j\boldsymbol{u}_j^{\top} = \boldsymbol{U}\boldsymbol{U}^{\mathsf{H}}\boldsymbol{W} + \boldsymbol{W}\bar{\boldsymbol{U}}\bar{\boldsymbol{U}}^{\mathsf{H}} - \boldsymbol{U}\boldsymbol{U}^{\mathsf{H}}\boldsymbol{W}\bar{\boldsymbol{U}}\bar{\boldsymbol{U}}^{\mathsf{H}}.
\end{aligned}
$$

On the other hand, we have,

$$\mathcal{J}\mathcal{P}_A\mathcal{J}^*(\boldsymbol{W})$$

$$= \left(\boldsymbol{I}_n - \frac{1}{2}\boldsymbol{Z}(\boldsymbol{Z}^\mathsf{H}\boldsymbol{Z})^{-1}\boldsymbol{Z}^\mathsf{H}\right)\boldsymbol{W}\bar{\boldsymbol{Z}}(\overline{\boldsymbol{Z}^\mathsf{H}\boldsymbol{Z}})^{-1}\boldsymbol{Z}^\top + \boldsymbol{Z}\left(\left(\boldsymbol{I}_n - \frac{1}{2}\boldsymbol{Z}(\boldsymbol{Z}^\mathsf{H}\boldsymbol{Z})^{-1}\boldsymbol{Z}^\mathsf{H}\right)\boldsymbol{W}\bar{\boldsymbol{Z}}(\overline{\boldsymbol{Z}^\mathsf{H}\boldsymbol{Z}})^{-1}\right)^\top$$

$$= \left(\boldsymbol{I}_n - \frac{1}{2}\boldsymbol{U}\boldsymbol{U}^\mathsf{H}\right)\boldsymbol{W}\bar{\boldsymbol{U}}\bar{\boldsymbol{U}}^\mathsf{H} + \boldsymbol{U}\boldsymbol{U}^\mathsf{H}\boldsymbol{W}\left(\boldsymbol{I}_n - \frac{1}{2}\bar{\boldsymbol{U}}\bar{\boldsymbol{U}}^\mathsf{H}\right)$$

$$= \boldsymbol{U}\boldsymbol{U}^\mathsf{H}\boldsymbol{W} + \boldsymbol{W}\bar{\boldsymbol{U}}\bar{\boldsymbol{U}}^\mathsf{H} - \boldsymbol{U}\boldsymbol{U}^\mathsf{H}\boldsymbol{W}\bar{\boldsymbol{U}}\bar{\boldsymbol{U}}^\mathsf{H},$$

completing the proof. □

### A.7. Proof of Theorem A.2

Although the preconditioners are different, they induce the same orthogonal projection onto the tangent space. This equivalence is a direct consequence of the first part of the theorem below.

**Theorem A.2.** *Under the same setting as in Theorem 2.1 for $i \geq j$, we have the following results:*

- *For $i < j$, $\exists\, \widetilde{\boldsymbol{V}}_{ij} = \sqrt{\frac{\sigma_i^2}{\sigma_i^2+\sigma_j^2}}\boldsymbol{u}_i\boldsymbol{v}_j^\top - \sqrt{\frac{\sigma_j^2}{\sigma_i^2+\sigma_j^2}}\boldsymbol{u}_j\boldsymbol{v}_i^\top$ such that $\langle \boldsymbol{V}_{ij}, \widetilde{\boldsymbol{V}}_{ij}\rangle = 0$, and $\mathcal{J}(\widetilde{\boldsymbol{V}}_{ij}) = \boldsymbol{0}$.*

- *$\{\boldsymbol{U}_{ij}\}_{\substack{i,j=1\\i>j}}^n \oplus \{\boldsymbol{U}_{ii}\}$ and $\{\boldsymbol{V}_{ij}\}_{\substack{i,j=1\\i>j}}^n \oplus \{\boldsymbol{V}_{ii}\} \oplus \{\widetilde{\boldsymbol{V}}_{ij}\}_{\substack{i,j=1\\i<j}}^n$ form an orthonormal basis for $\mathbb{C}_S^{n\times n}$ and $\mathbb{C}^{n\times r}$, respectively.*

In particular, the fully approximated preconditioner can be written as the originally defined $\mathcal{P}_A$ plus a perturbation $\Delta$, where $\Delta$ lies in the span of the orthonormal set $\{\widetilde{\boldsymbol{V}}_{ij}\}$. Since $\mathcal{J}(\widetilde{\boldsymbol{V}}_{ij}) = \boldsymbol{0}$ (i.e., they belong to the kernel of the Jacobi), adding $\Delta$ does not affect the resulting tangent-space projection. Accordingly, it suffices to use the reduced approximate preconditioner $\mathcal{P}_A$ directly.

We now provide the proof of Theorem A.2

*Proof.* For the first statement, it's easy to check the orthogonality.
For $i < j$,

$$\mathcal{J}(\widetilde{\boldsymbol{V}}_{ij}) = \mathcal{J}\left(\sqrt{\frac{\sigma_i^2}{\sigma_i^2+\sigma_j^2}}\boldsymbol{u}_i\boldsymbol{v}_j^\top - \sqrt{\frac{\sigma_j^2}{\sigma_i^2+\sigma_j^2}}\boldsymbol{u}_j\boldsymbol{v}_i^\top\right)$$

$$= \sum_{k=1}^r \sigma_k\boldsymbol{u}_k\boldsymbol{v}_k^\mathsf{H}\left(\sqrt{\frac{\sigma_i^2}{\sigma_i^2+\sigma_j^2}}\boldsymbol{v}_j\boldsymbol{u}_i^\top - \sqrt{\frac{\sigma_j^2}{\sigma_i^2+\sigma_j^2}}\boldsymbol{v}_i\boldsymbol{u}_j^\top\right) + \left(\sqrt{\frac{\sigma_i^2}{\sigma_i^2+\sigma_j^2}}\boldsymbol{u}_i\boldsymbol{v}_j^\top - \sqrt{\frac{\sigma_j^2}{\sigma_i^2+\sigma_j^2}}\boldsymbol{u}_j\boldsymbol{v}_i^\top\right)\sum_{k=1}^r \sigma_k\bar{\boldsymbol{v}}_k\boldsymbol{u}_k^\top$$

$$= \sqrt{\frac{\sigma_i^2}{\sigma_i^2+\sigma_j^2}}\cdot\sigma_j\boldsymbol{u}_j\boldsymbol{u}_i^\top - \sqrt{\frac{\sigma_j^2}{\sigma_i^2+\sigma_j^2}}\cdot\sigma_i\boldsymbol{u}_i\boldsymbol{u}_j^\top + \sqrt{\frac{\sigma_i^2}{\sigma_i^2+\sigma_j^2}}\cdot\sigma_j\boldsymbol{u}_i\boldsymbol{u}_j^\top - \sqrt{\frac{\sigma_j^2}{\sigma_i^2+\sigma_j^2}}\cdot\sigma_i\boldsymbol{u}_j\boldsymbol{u}_i^\top$$

$$= \boldsymbol{0}.$$

To prove the second statement, verifying orthogonality and unit norm is straightforward; therefore, it remains only to establish completeness. Specifically, we must prove

$$\sum_{(i,j):i\geq j}\langle \boldsymbol{Y}, \boldsymbol{U}_{ij}\rangle\boldsymbol{U}_{ij} = \boldsymbol{Y}, \qquad \forall \boldsymbol{Y} \in \mathbb{C}_S^{n\times n}.$$

$$\sum_{(i,j):\min\{i,j\}\leq r}\langle \boldsymbol{M}, \boldsymbol{V}_{ij}\rangle\boldsymbol{V}_{ij} = \boldsymbol{M}, \qquad \forall \boldsymbol{M} \in \mathbb{C}^{n\times r}.$$

For the first equation,

$$\sum_{(i,j):i\geq j}\langle \boldsymbol{Y}, \boldsymbol{U}_{ij}\rangle\boldsymbol{U}_{ij}$$

$$= \sum_{(i,j):i>j} \frac{1}{2} \left\langle Y, u_i u_j^\top + u_j u_i^\top \right\rangle \left( u_i u_j^\top + u_j u_i^\top \right) + \sum_{i=j \leq r} \left\langle Y, u_i u_i^\top \right\rangle u_i u_i^\top$$

$$= \sum_{(i,j):i>j} \frac{1}{2} \left\langle Y, u_i u_j^\top \right\rangle u_i u_j^\top + \sum_{(i,j):i>j} \frac{1}{2} \left\langle Y, u_j u_i^\top \right\rangle u_j u_i^\top + \sum_{i=j \leq r} \left\langle Y, u_i u_i^\top \right\rangle u_i u_i^\top$$

$$+ \sum_{(i,j):i>j} \frac{1}{2} \left\langle Y, u_i u_j^\top \right\rangle u_j u_i^\top + \sum_{(i,j):i>j} \frac{1}{2} \left\langle Y, u_j u_i^\top \right\rangle u_i u_j^\top$$

$$= \sum_{(i,j):i>j} \frac{1}{2} \left\langle Y, u_i u_j^\top \right\rangle u_i u_j^\top + \sum_{(i,j):i>j} \frac{1}{2} \left\langle Y, u_j u_i^\top \right\rangle u_j u_i^\top + \sum_{i=j \leq r} \left\langle Y, u_i u_i^\top \right\rangle u_i u_i^\top$$

$$+ \sum_{(i,j):i>j} \frac{1}{2} \left\langle Y, u_j u_i^\top \right\rangle u_j u_i^\top + \sum_{(i,j):i>j} \frac{1}{2} \left\langle Y, u_i u_j^\top \right\rangle u_i u_j^\top$$

$$= \sum_{i,j=1}^n \frac{1}{2} \left\langle Y, u_i u_j^\top \right\rangle u_i u_j^\top + \sum_{i,j=1}^n \frac{1}{2} \left\langle Y, u_j u_i^\top \right\rangle u_j u_i^\top = Y,$$

where the third equality follows the symmetry of $Y$ and the property of trace, $\left\langle Y, u_i u_j^\top \right\rangle = \text{tr}\left( \bar{u}_j u_i^{\mathsf{H}} Y \right) = \text{tr}\left( Y^\top \bar{u}_i u_j^{\mathsf{H}} \right) = \left\langle Y^\top, u_j u_i^\top \right\rangle = \left\langle Y, u_j u_i^\top \right\rangle$.

For the second equation,

$$\sum_{(i,j):\min\{i,j\} \leq r} \langle M, V_{ij} \rangle V_{ij}$$

$$= \sum_{\substack{(i,j):\min\{i,j\} \leq r \\ i>j}} \langle M, V_{ij} \rangle V_{ij} + \sum_{(i,j):i=j \leq r} \langle M, V_{ii} \rangle V_{ii} + \sum_{\substack{(i,j):\min\{i,j\} \leq r \\ i<j}} \langle M, \widetilde{V}_{ij} \rangle \widetilde{V}_{ij}$$

$$= \sum_{\substack{(i,j):\min\{i,j\} \leq r \\ i>j}} \left\langle M, \sqrt{\frac{\sigma_j^2}{\sigma_i^2+\sigma_j^2}} u_i v_j^\top + \sqrt{\frac{\sigma_i^2}{\sigma_i^2+\sigma_j^2}} u_j v_i^\top \right\rangle \cdot \left( \sqrt{\frac{\sigma_j^2}{\sigma_i^2+\sigma_j^2}} u_i v_j^\top + \sqrt{\frac{\sigma_i^2}{\sigma_i^2+\sigma_j^2}} u_j v_i^\top \right) + \sum_{i=j \leq r} \left\langle M, u_i v_i^\top \right\rangle u_i v_i^\top$$

$$+ \sum_{\substack{(i,j):\min\{i,j\} \leq r \\ i<j}} \left\langle M, \sqrt{\frac{\sigma_i^2}{\sigma_i^2+\sigma_j^2}} u_i v_j^\top - \sqrt{\frac{\sigma_j^2}{\sigma_i^2+\sigma_j^2}} u_j v_i^\top \right\rangle \cdot \left( \sqrt{\frac{\sigma_i^2}{\sigma_i^2+\sigma_j^2}} u_i v_j^\top - \sqrt{\frac{\sigma_j^2}{\sigma_i^2+\sigma_j^2}} u_j v_i^\top \right)$$

$$= \sum_{\substack{(i,j):\min\{i,j\} \leq r \\ i>j}} \frac{\sigma_j^2}{\sigma_i^2+\sigma_j^2} \left\langle M, u_i v_j^\top \right\rangle u_i v_j^\top + \sum_{\substack{(i,j):\min\{i,j\} \leq r \\ i>j}} \frac{\sigma_i^2}{\sigma_i^2+\sigma_j^2} \left\langle M, u_j v_i^\top \right\rangle u_j v_i^\top + \sum_{i=j \leq r} \left\langle M, u_i v_i^\top \right\rangle u_i v_i^\top$$

$$+ \sum_{\substack{(i,j):\min\{i,j\} \leq r \\ i<j}} \frac{\sigma_i^2}{\sigma_i^2+\sigma_j^2} \left\langle M, u_i v_j^\top \right\rangle u_i v_j^\top + \sum_{\substack{(i,j):\min\{i,j\} \leq r \\ i<j}} \frac{\sigma_j^2}{\sigma_i^2+\sigma_j^2} \left\langle M, u_j v_i^\top \right\rangle u_j v_i^\top$$

$$+ \sum_{\substack{(i,j):\min\{i,j\} \leq r \\ i>j}} \frac{\sigma_i \sigma_j}{\sigma_i^2+\sigma_j^2} \left\langle M, u_i v_j^\top \right\rangle u_j v_i^\top + \sum_{\substack{(i,j):\min\{i,j\} \leq r \\ i>j}} \frac{\sigma_i \sigma_j}{\sigma_i^2+\sigma_j^2} \left\langle M, u_j v_i^\top \right\rangle u_i v_j^\top$$

$$- \sum_{\substack{(i,j):\min\{i,j\} \leq r \\ i<j}} \frac{\sigma_i \sigma_j}{\sigma_i^2+\sigma_j^2} \left\langle M, u_j v_i^\top \right\rangle u_i v_j^\top - \sum_{\substack{(i,j):\min\{i,j\} \leq r \\ i<j}} \frac{\sigma_i \sigma_j}{\sigma_i^2+\sigma_j^2} \left\langle M, u_i v_j^\top \right\rangle u_j v_i^\top$$

$$= \sum_{\substack{(i,j):\min\{i,j\} \leq r \\ i>j}} \frac{\sigma_j^2}{\sigma_i^2+\sigma_j^2} \left\langle M, u_i v_j^\top \right\rangle u_i v_j^\top + \sum_{\substack{(i,j):\min\{i,j\} \leq r \\ i<j}} \frac{\sigma_j^2}{\sigma_i^2+\sigma_j^2} \left\langle M, u_j v_i^\top \right\rangle u_j v_i^\top + \sum_{i=j \leq r} \left\langle M, u_i v_i^\top \right\rangle u_i v_i^\top$$

$$+ \sum_{\substack{(i,j):\min\{i,j\} \leq r \\ i<j}} \frac{\sigma_i^2}{\sigma_i^2+\sigma_j^2} \left\langle M, u_i v_j^\top \right\rangle u_i v_j^\top + \sum_{\substack{(i,j):\min\{i,j\} \leq r \\ i>j}} \frac{\sigma_i^2}{\sigma_i^2+\sigma_j^2} \left\langle M, u_j v_i^\top \right\rangle u_j v_i^\top$$

$$= \sum_{\substack{(i,j):\min\{i,j\} \leq r \\ i>j}} \left\langle M, u_i v_j^\top \right\rangle u_i v_j^\top + \sum_{i=j \leq r} \left\langle M, u_i v_i^\top \right\rangle u_i v_i^\top + \sum_{\substack{(i,j):\min\{i,j\} \leq r \\ i<j}} \left\langle M, u_i v_j^\top \right\rangle u_i v_j^\top$$

$$= \sum_{j=1}^{r} \sum_{i=1}^{n} \langle M, u_i v_j^\top \rangle u_i v_j^\top = \sum_{j=1}^{r} \sum_{i=1}^{n} u_i u_i^{\mathsf{H}} M \bar{v}_j v_j^\top = \left( \sum_{i=1}^{n} u_i u_i^{\mathsf{H}} \right) M \left( \sum_{j=1}^{r} \bar{v}_j v_j^\top \right) = M.$$

$\square$

## B. Proofs of Results in Section 3

As a preliminary step, we outline the essential definitions and properties of the operators $\mathcal{H}$ and $\mathcal{G}$.

The adjoint operator of $\mathcal{H}$ is denoted by $\mathcal{H}^*$, which maps a matrix $Z \in \mathbb{C}^{n \times n}$ to a vector with size $n_s$-by-1, that is,

$$\mathcal{H}^*(Z) = \left\{ \sum_{i+j=a} Z_{i,j} \right\}_{a=0}^{n_s-1}.$$

An orthonormal basis of Hankel matrices with size $n$-by-$n$ is

$$\left\{ H_a := \frac{1}{\sqrt{w_a}} \mathcal{H} e_a \,\middle|\, w_a = \# \left\{ (i,j) \mid i+j = a, \ i \in [n], \ j \in [n] \right\} \right\}_{a=0}^{n_s-1},$$

where $e_a$ is the $a$-th canonical basis of $\mathbb{C}^{n_s}$.

Recall $\mathcal{G} = \mathcal{H}\mathcal{D}^{-1}$, its adjoint is $\mathcal{G}^* = \mathcal{D}^{-1}\mathcal{H}^*$. One can readily verify $\mathcal{G}^*\mathcal{G} = \mathcal{I}, \|\mathcal{G}\| \leq 1$, and $\|\mathcal{G}^*\| \leq 1$. Furthermore, $\mathcal{G}z$ and $\mathcal{G}^*Z$ can be expressed in terms of basis $\{H_a\}$,

$$\mathcal{G}z = \sum_{a=0}^{n_s-1} z_a H_a, \ \forall \, z \in \mathbb{C}^{n_s}$$

$$\mathcal{G}^*Z = \{\langle Z, H_a \rangle\}_{a=0}^{n_s-1}, \ \forall \, Z \in \mathbb{C}^{n \times n}.$$

### B.1. Technical Lemmas

**Lemma B.1.** For matrix $Z \in \mathbb{C}^{n \times r}$ with full column rank, its pseudo-inverse is $Z^\dagger = (Z^{\mathsf{H}} Z)^{-1} Z^{\mathsf{H}}$. Then we have $(ZZ^\top)^\dagger = (Z^\top)^\dagger Z^\dagger$.

*Proof of Lemma B.1.* For full column rank $Z$, we have $Z^\dagger Z = I_r$, and $Z^\top(Z^\top)^\dagger = I_r$. Define $X = ZZ^\top$, $Y = (Z^\top)^\dagger Z^\dagger$, we need to prove $X^\dagger = Y$. It's sufficient to check the following four conditions,
1. $XYX = ZZ^\top(Z^\top)^\dagger Z^\dagger ZZ^\top = ZZ^\dagger ZZ^\top = ZZ^\top = X$.
2. $YXY = (Z^\top)^\dagger Z^\dagger ZZ^\top(Z^\top)^\dagger Z^\dagger = (Z^\top)^\dagger Z^\dagger ZZ^\dagger = (Z^\top)^\dagger Z^\dagger = Y$.
3. $(XY)^* = (ZZ^\top(Z^\top)^\dagger Z^\dagger)^* = (ZZ^\dagger)^* = ZZ^\dagger = ZZ^\top(Z^\top)^\dagger Z^\dagger = XY$.
4. $(YX)^* = ((Z^\top)^\dagger Z^\dagger ZZ^\top)^* = ((Z^\top)^\dagger Z^\top)^* = (Z^\top)^\dagger Z^\top = YX$. $\square$

**Lemma B.2.** Let $\mathcal{G}y = \mathcal{H}x = \tilde{U}\Sigma\tilde{V}^{\mathsf{H}} = U\Sigma U^\top = EDE^\top \in \mathbb{C}^{n \times n}$. Assume $\mathcal{H}x$ is $\mu_0$-incoherent, and define $c_s = \frac{n_s}{n} \leq 2$. ($n_s$ denotes the signal size, where $n_s + 1 = 2n$). Then

$$\left\| U^{(i,:)} \right\|^2 \leq \frac{2\mu_0 r}{n_s} \quad \text{and} \quad \|\mathcal{P}_U(H_a)\|_F^2 \leq \frac{2\mu_0 r}{n_s}.$$

($\tilde{U}\Sigma\tilde{V}^{\mathsf{H}}$ is SVD of $\mathcal{G}y$, which can be expressed in the form of Takagi factorization $U\Sigma U^\top$.)

**Lemma B.3.** [(Recht, 2011),Proposition 3.3] Under the sampling with replacement model, the maximum number of repetitions of any entry in $\Omega$ is less than $8\log(n_s)$ with probability at least $1 - n_s^{-2}$ provided $n_s \geq 9$.

**Lemma B.4.** [(Chen & Chi, 2014), Lemma 3] Let $U \in \mathbb{C}^{n \times r}$ be an orthogonal matrix which satisfies

$$\|\mathcal{P}_U(H_a)\|_F^2 \leq \frac{2\mu_0 r}{n_s}.$$

Let $\mathbb{T}$ be the subspace defined as $\{\boldsymbol{U}\boldsymbol{M}^\top + \boldsymbol{M}\boldsymbol{U}^\top \,|\, \forall \boldsymbol{M} \in \mathbb{C}^{n\times r}\}$. Then

$$\left\|\mathcal{P}_\mathbb{T}\mathcal{G}\mathcal{G}^*\mathcal{P}_\mathbb{T} - p^{-1}\mathcal{P}_\mathbb{T}\mathcal{G}\mathcal{P}_\Omega\mathcal{G}^*\mathcal{P}_\mathbb{T}\right\| \leq \sqrt{\frac{64\mu_0 r \log(n_s)}{m}}$$

holds with probability at least $1 - n_s^{-2}$ provided that

$$m \geq 64\mu_0 r \log(n_s).$$

**Lemma B.5.** [(Wei et al., 2020), Lemma 4.1] Let $\boldsymbol{X}_t = \boldsymbol{Z}_t\boldsymbol{Z}_t^\top = (\boldsymbol{U}_t\boldsymbol{\Sigma}_t\boldsymbol{V}_t^{\mathsf{H}})(\boldsymbol{U}_t\boldsymbol{\Sigma}_t\boldsymbol{V}_t^{\mathsf{H}})^\top$ be another rank-$r$ matrix and $\mathbb{T}_t$ be the tangent space of the rank-$r$ matrix manifold at $\boldsymbol{X}_t$ as defined in Lemma B.4. Then

$$\left\|(\mathcal{I} - \mathcal{P}_{\mathbb{T}_t})(\boldsymbol{X}_t - \mathcal{G}\boldsymbol{y})\right\|_F \leq \frac{\|\boldsymbol{X}_t - \mathcal{G}\boldsymbol{y}\|_F^2}{\sigma_{\min}(\mathcal{G}\boldsymbol{y})}, \quad \left\|\mathcal{P}_{\mathbb{T}_t} - \mathcal{P}_\mathbb{T}\right\| \leq \frac{2\|\boldsymbol{X}_t - \mathcal{G}\boldsymbol{y}\|_F}{\sigma_{\min}(\mathcal{G}\boldsymbol{y})}.$$

**Lemma B.6.** [(Tropp, 2012), Theorem 1.6] Consider a finite sequence $\{\boldsymbol{Z}_k\}$ of independent, random matrices with dimensions $d_1 \times d_2$. Assume that each random matrix satisfies

$$\mathbb{E}(\boldsymbol{Z}_k) = 0 \quad \text{and} \quad \|\boldsymbol{Z}_k\| \leq R \quad \text{almost surely.}$$

Define

$$\sigma^2 := \max\left\{\left\|\sum_k \mathbb{E}\left(\boldsymbol{Z}_k\boldsymbol{Z}_k^{\mathsf{H}}\right)\right\|, \left\|\sum_k \mathbb{E}\left(\boldsymbol{Z}_k^{\mathsf{H}}\boldsymbol{Z}_k\right)\right\|\right\}.$$

Then for all $t \geq 0$,

$$\mathbb{P}\left\{\left\|\sum_k \boldsymbol{Z}_k\right\| \geq t\right\} \leq (d_1 + d_2)\exp\left(\frac{-t^2/2}{\sigma^2 + Rt/3}\right).$$

## B.2. Proof of Theorem 3.1

In this section, we present detailed proofs of our first main theorem.

### B.2.1. INITIALIZATION VIA TAKAGI FACTORIZATION

In Theorem 3.1, the initialization in Algorithm 1 is $\boldsymbol{Z}_0 = \boldsymbol{U}_0\boldsymbol{\Sigma}_0^{1/2}$, if we map this to the matrix space, we obtain the initialization $\boldsymbol{X}_0 = \boldsymbol{Z}_0\boldsymbol{Z}_0^\top = \boldsymbol{U}_0\boldsymbol{\Sigma}_0\boldsymbol{U}_0^\top = \mathcal{T}_r\left(p^{-1}\mathcal{H}\mathcal{P}_\Omega(\boldsymbol{y})\right)$. So to prove the theorem, we begin by stating a lemma which shows that, under an incoherence condition and for sufficiently large $m$, $\boldsymbol{X}_0$ provides a reliable approximation of $\mathcal{H}\boldsymbol{x}$ with high probability.

**Lemma B.7.** [(Cai et al., 2019), Lemma 2] Assume $\mathcal{H}\boldsymbol{x} = \mathcal{G}\boldsymbol{y}$ is $\mu_0$-incoherent. Then there exists a universal constant $C > 0$ such that

$$\|\boldsymbol{X}_0 - \mathcal{H}\boldsymbol{x}\| \leq C_1\sqrt{\frac{2\mu_0 r \log(n_s)}{m}}\|\mathcal{H}\boldsymbol{x}\|$$

with probability at least $1 - n_s^{-2}$.

A detailed proof is provided in (Cai et al., 2019); the only distinction is that we consider the Hankel matrix to be complex symmetric, implying that $c_s \leq 2$.

### B.2.2. LOCAL CONVERGENCE AND PROOF OF THEOREM 3.1

In this section, we analyze the convergence of Algorithm 1 in matrix space, which is Algorithm 2. Equation (17) has the following compact expression,

$$\boldsymbol{y}_{t+1} = \mathcal{G}^*(\boldsymbol{X}_{t+1}) = \mathcal{G}^*(\mathcal{R}(\boldsymbol{L}_t)), \tag{29}$$

where

$$\boldsymbol{L}_t := \boldsymbol{X}_t - \alpha\mathcal{P}_{\mathbb{T}_t}(\nabla F(\boldsymbol{X}_t)) = \boldsymbol{X}_t - \alpha\mathcal{P}_{\mathbb{T}_t}\left[\frac{1}{2p}\mathcal{G}\mathcal{P}_\Omega\left(\mathcal{G}^*(\boldsymbol{X}_t) - \boldsymbol{y}\right) + \frac{1}{2}\left(\mathcal{I} - \mathcal{G}\mathcal{G}^*\right)(\boldsymbol{X}_t)\right].$$

We begin by establishing several key lemmas necessary for the proof of the theorem.

**Theorem B.1.** *For the iteration ([17](#)) in matrix space, we have the following bound,*

$$\|\boldsymbol{X}_{t+1} - \boldsymbol{L}_t\|_F \leq \frac{3}{4\sigma_r(\boldsymbol{X}_t)}\|\boldsymbol{L}_t - \boldsymbol{X}_t\|_F^2. \tag{30}$$

*Proof of Theorem B.1.* Recall $\boldsymbol{X}_{t+1}$ is updated as following,

$$\boldsymbol{X}_{t+1} = \mathcal{R}\left(\boldsymbol{L}_t\right) = \mathcal{R}\left(\boldsymbol{X}_t - \alpha\mathcal{P}_{\mathbb{T}_t}(\nabla F(\boldsymbol{X}_t))\right),$$

then we have

$$
\begin{aligned}
&\|\boldsymbol{L}_t - \boldsymbol{X}_t\|_F^2 \\
&= \|\alpha\mathcal{P}_{\mathbb{T}_t}(\nabla F(\boldsymbol{X}_t))\|_F^2 \\
&= \alpha^2\|\boldsymbol{U}_t\boldsymbol{U}_t^{\mathsf{H}}\nabla F(\boldsymbol{X}_t) + \nabla F(\boldsymbol{X}_t)\bar{\boldsymbol{U}}_t\bar{\boldsymbol{U}}_t^{\mathsf{H}} - \boldsymbol{U}_t\boldsymbol{U}_t^{\mathsf{H}}\nabla F(\boldsymbol{X}_t)\bar{\boldsymbol{U}}_t\bar{\boldsymbol{U}}_t^{\mathsf{H}}\|_F^2 \\
&= \alpha^2\|\boldsymbol{U}_t\boldsymbol{U}_t^{\mathsf{H}}\nabla F(\boldsymbol{X}_t)(\boldsymbol{I}_n - \bar{\boldsymbol{U}}_t\bar{\boldsymbol{U}}_t^{\mathsf{H}}) + (\boldsymbol{I}_n - \boldsymbol{U}_t\boldsymbol{U}_t^{\mathsf{H}})\nabla F(\boldsymbol{X}_t)\bar{\boldsymbol{U}}_t\bar{\boldsymbol{U}}_t^{\mathsf{H}} + \boldsymbol{U}_t\boldsymbol{U}_t^{\mathsf{H}}\nabla F(\boldsymbol{X}_t)\bar{\boldsymbol{U}}_t\bar{\boldsymbol{U}}_t^{\mathsf{H}}\|_F^2 \\
&= \alpha^2\left\{\|\boldsymbol{U}_t\boldsymbol{U}_t^{\mathsf{H}}\nabla F(\boldsymbol{X}_t)(\boldsymbol{I}_n - \bar{\boldsymbol{U}}_t\bar{\boldsymbol{U}}_t^{\mathsf{H}})\|_F^2 + \|(\boldsymbol{I}_n - \boldsymbol{U}_t\boldsymbol{U}_t^{\mathsf{H}})\nabla F(\boldsymbol{X}_t)\bar{\boldsymbol{U}}_t\bar{\boldsymbol{U}}_t^{\mathsf{H}}\|_F^2 + \|\boldsymbol{U}_t\boldsymbol{U}_t^{\mathsf{H}}\nabla F(\boldsymbol{X}_t)\bar{\boldsymbol{U}}_t\bar{\boldsymbol{U}}_t^{\mathsf{H}}\|_F^2\right.\\
&\quad + 2\Re\left\{\langle\boldsymbol{U}\boldsymbol{U}^{\mathsf{H}}\nabla F(\boldsymbol{X}_t)(\boldsymbol{I}_n - \bar{\boldsymbol{U}}_t\bar{\boldsymbol{U}}_t^{\mathsf{H}}), (\boldsymbol{I}_n - \boldsymbol{U}_t\boldsymbol{U}_t^{\mathsf{H}})\nabla F(\boldsymbol{X}_t)\bar{\boldsymbol{U}}_t\bar{\boldsymbol{U}}_t^{\mathsf{H}}\rangle\right\}\\
&\quad + 2\Re\left\{\langle\boldsymbol{U}_t\boldsymbol{U}_t^{\mathsf{H}}\nabla F(\boldsymbol{X}_t)(\boldsymbol{I}_n - \bar{\boldsymbol{U}}_t\bar{\boldsymbol{U}}_t^{\mathsf{H}}), \boldsymbol{U}_t\boldsymbol{U}_t^{\mathsf{H}}\nabla F(\boldsymbol{X}_t)\bar{\boldsymbol{U}}_t\bar{\boldsymbol{U}}_t^{\mathsf{H}}\rangle\right\}\\
&\quad + \left.2\Re\left\{\langle(\boldsymbol{I}_n - \boldsymbol{U}_t\boldsymbol{U}_t^{\mathsf{H}})\nabla F(\boldsymbol{X}_t)\bar{\boldsymbol{U}}_t\bar{\boldsymbol{U}}_t^{\mathsf{H}}, \boldsymbol{U}_t\boldsymbol{U}_t^{\mathsf{H}}\nabla F(\boldsymbol{X}_t)\bar{\boldsymbol{U}}_t\bar{\boldsymbol{U}}_t^{\mathsf{H}}\rangle\right\}\right\}\\
&= \alpha^2\left\{\|\boldsymbol{U}_t\boldsymbol{U}_t^{\mathsf{H}}\nabla F(\boldsymbol{X}_t)(\boldsymbol{I}_n - \bar{\boldsymbol{U}}_t\bar{\boldsymbol{U}}_t^{\mathsf{H}})\|_F^2 + \|(\boldsymbol{I}_n - \boldsymbol{U}_t\boldsymbol{U}_t^{\mathsf{H}})\nabla F(\boldsymbol{X}_t)\bar{\boldsymbol{U}}_t\bar{\boldsymbol{U}}_t^{\mathsf{H}}\|_F^2 + \|\boldsymbol{U}_t\boldsymbol{U}_t^{\mathsf{H}}\nabla F(\boldsymbol{X}_t)\bar{\boldsymbol{U}}_t\bar{\boldsymbol{U}}_t^{\mathsf{H}}\|_F^2\right\},
\end{aligned}
$$

where $\Re(\boldsymbol{Y})$ denotes the real part of the complex matrix $\boldsymbol{Y}$. In the last equation, we use the orthogonality property, so the three real trace parts are zero. The next is to bound the second order error part,

$$
\begin{aligned}
&\boldsymbol{X}_{t+1} - \boldsymbol{L}_t \\
&= \alpha^2\left(\boldsymbol{I}_n - \frac{1}{2}\boldsymbol{Z}_t(\boldsymbol{Z}_t^{\mathsf{H}}\boldsymbol{Z}_t)^{-1}\boldsymbol{Z}_t^{\mathsf{H}}\right)\nabla F(\boldsymbol{X}_t)\bar{\boldsymbol{Z}}_t(\overline{\boldsymbol{Z}_t^{\mathsf{H}}\boldsymbol{Z}_t})^{-1}(\boldsymbol{Z}_t^{\mathsf{H}}\boldsymbol{Z}_t)^{-1}\boldsymbol{Z}^{\mathsf{H}}\nabla F(\boldsymbol{X}_t)\left(\boldsymbol{I}_n - \frac{1}{2}\bar{\boldsymbol{Z}}_t(\overline{\boldsymbol{Z}_t^{\mathsf{H}}\boldsymbol{Z}_t})^{-1}\bar{\boldsymbol{Z}}_t^{\mathsf{H}}\right) \\
&= \alpha^2\left(\boldsymbol{I}_n - \frac{1}{2}\boldsymbol{U}_t\boldsymbol{U}_t^{\mathsf{H}}\right)\nabla F(\boldsymbol{X}_t)(\boldsymbol{Z}_t^{\top})^{\dagger}\boldsymbol{Z}_t^{\dagger}\nabla F(\boldsymbol{X}_t)\left(\boldsymbol{I}_n - \frac{1}{2}\overline{\boldsymbol{U}_t\boldsymbol{U}_t^{\mathsf{H}}}\right) \\
&= \alpha^2\left(\boldsymbol{I}_n - \boldsymbol{U}_t\boldsymbol{U}_t^{\mathsf{H}}\right)\nabla F(\boldsymbol{X}_t)(\boldsymbol{Z}_t\boldsymbol{Z}_t^{\top})^{\dagger}\nabla F(\boldsymbol{X}_t)\left(\boldsymbol{I}_n - \overline{\boldsymbol{U}_t\boldsymbol{U}_t^{\mathsf{H}}}\right) + \frac{\alpha^2}{2}\left(\boldsymbol{I}_n - \boldsymbol{U}_t\boldsymbol{U}_t^{\mathsf{H}}\right)\nabla F(\boldsymbol{X}_t)(\boldsymbol{Z}_t\boldsymbol{Z}_t^{\top})^{\dagger}\nabla F(\boldsymbol{X}_t)\overline{\boldsymbol{U}_t\boldsymbol{U}_t^{\mathsf{H}}} \\
&\quad + \frac{\alpha^2}{2}\boldsymbol{U}_t\boldsymbol{U}_t^{\mathsf{H}}\nabla F(\boldsymbol{X}_t)(\boldsymbol{Z}_t\boldsymbol{Z}_t^{\top})^{\dagger}\nabla F(\boldsymbol{X}_t)\left(\boldsymbol{I}_n - \overline{\boldsymbol{U}_t\boldsymbol{U}_t^{\mathsf{H}}}\right) + \frac{\alpha^2}{4}\boldsymbol{U}_t\boldsymbol{U}_t^{\mathsf{H}}\nabla F(\boldsymbol{X}_t)(\boldsymbol{Z}_t\boldsymbol{Z}_t^{\top})^{\dagger}\nabla F(\boldsymbol{X}_t)\overline{\boldsymbol{U}_t\boldsymbol{U}_t^{\mathsf{H}}}.
\end{aligned}
$$

The second equation utilizes the equality expression: $\boldsymbol{Z}^{\dagger} = (\boldsymbol{Z}^{\mathsf{H}}\boldsymbol{Z})^{-1}\boldsymbol{Z}^{\mathsf{H}}$, and $\bar{\boldsymbol{Z}}_t(\overline{\boldsymbol{Z}_t^{\mathsf{H}}\boldsymbol{Z}_t})^{-1} = \overline{[(\boldsymbol{Z}^{\mathsf{H}}\boldsymbol{Z})^{-1}\boldsymbol{Z}^{\mathsf{H}}]}^{\mathsf{H}} = (\boldsymbol{Z}^{\top})^{\dagger}$. The third equation uses the result from lemma B.1.

Then, we bound the difference via Frobenius norm,

$$
\begin{aligned}
&\|\boldsymbol{X}_{t+1} - \boldsymbol{L}_t\|_F \\
&\leq \alpha^2\left\|\left(\boldsymbol{I}_n - \boldsymbol{U}_t\boldsymbol{U}_t^{\mathsf{H}}\right)\nabla F(\boldsymbol{X}_t)(\boldsymbol{Z}_t\boldsymbol{Z}_t^{\top})^{\dagger}\nabla F(\boldsymbol{X}_t)\left(\boldsymbol{I}_n - \overline{\boldsymbol{U}_t\boldsymbol{U}_t^{\mathsf{H}}}\right)\right\|_F + \frac{\alpha^2}{2}\left\|\left(\boldsymbol{I}_n - \boldsymbol{U}_t\boldsymbol{U}_t^{\mathsf{H}}\right)\nabla F(\boldsymbol{X}_t)(\boldsymbol{Z}_t\boldsymbol{Z}_t^{\top})^{\dagger}\nabla F(\boldsymbol{X}_t)\overline{\boldsymbol{U}_t\boldsymbol{U}_t^{\mathsf{H}}}\right\|_F \\
&\quad + \frac{\alpha^2}{2}\left\|\boldsymbol{U}_t\boldsymbol{U}_t^{\mathsf{H}}\nabla F(\boldsymbol{X}_t)(\boldsymbol{Z}_t\boldsymbol{Z}_t^{\top})^{\dagger}\nabla F(\boldsymbol{X}_t)\left(\boldsymbol{I}_n - \overline{\boldsymbol{U}_t\boldsymbol{U}_t^{\mathsf{H}}}\right)\right\|_F + \frac{\alpha^2}{4}\left\|\boldsymbol{U}_t\boldsymbol{U}_t^{\mathsf{H}}\nabla F(\boldsymbol{X}_t)(\boldsymbol{Z}_t\boldsymbol{Z}_t^{\top})^{\dagger}\nabla F(\boldsymbol{X}_t)\overline{\boldsymbol{U}_t\boldsymbol{U}_t^{\mathsf{H}}}\right\|_F \\
&= \alpha^2\left\|\left(\boldsymbol{I}_n - \boldsymbol{U}_t\boldsymbol{U}_t^{\mathsf{H}}\right)\nabla F(\boldsymbol{X}_t)\overline{\boldsymbol{U}_t\boldsymbol{U}_t^{\mathsf{H}}}(\boldsymbol{Z}_t\boldsymbol{Z}_t^{\top})^{\dagger}\boldsymbol{U}_t\boldsymbol{U}_t^{\mathsf{H}}\nabla F(\boldsymbol{X}_t)\left(\boldsymbol{I}_n - \overline{\boldsymbol{U}_t\boldsymbol{U}_t^{\mathsf{H}}}\right)\right\|_F \\
&\quad + \frac{\alpha^2}{2}\left\|\left(\boldsymbol{I}_n - \boldsymbol{U}_t\boldsymbol{U}_t^{\mathsf{H}}\right)\nabla F(\boldsymbol{X}_t)\overline{\boldsymbol{U}_t\boldsymbol{U}_t^{\mathsf{H}}}(\boldsymbol{Z}_t\boldsymbol{Z}_t^{\top})^{\dagger}\boldsymbol{U}_t\boldsymbol{U}_t^{\mathsf{H}}\nabla F(\boldsymbol{X}_t)\overline{\boldsymbol{U}_t\boldsymbol{U}_t^{\mathsf{H}}}\right\|_F \\
&\quad + \frac{\alpha^2}{2}\left\|\boldsymbol{U}_t\boldsymbol{U}_t^{\mathsf{H}}\nabla F(\boldsymbol{X}_t)\overline{\boldsymbol{U}_t\boldsymbol{U}_t^{\mathsf{H}}}(\boldsymbol{Z}_t\boldsymbol{Z}_t^{\top})^{\dagger}\boldsymbol{U}_t\boldsymbol{U}_t^{\mathsf{H}}\nabla F(\boldsymbol{X}_t)\left(\boldsymbol{I}_n - \overline{\boldsymbol{U}_t\boldsymbol{U}_t^{\mathsf{H}}}\right)\right\|_F
\end{aligned}
$$

$$+ \frac{\alpha^2}{4} \left\| \boldsymbol{U}_t \boldsymbol{U}_t^\mathsf{H} \nabla F(\boldsymbol{X}_t) \overline{\boldsymbol{U}_t \boldsymbol{U}_t^\mathsf{H}} (\boldsymbol{Z}_t \boldsymbol{Z}_t^\top)^\dagger \boldsymbol{U}_t \boldsymbol{U}_t^\mathsf{H} \nabla F(\boldsymbol{X}_t) \overline{\boldsymbol{U}_t \boldsymbol{U}_t^\mathsf{H}} \right\|_F$$

$$\leq \alpha^2 \left\| \left( \boldsymbol{I}_n - \boldsymbol{U}_t \boldsymbol{U}_t^\mathsf{H} \right) \nabla F(\boldsymbol{X}_t) \overline{\boldsymbol{U}_t \boldsymbol{U}_t^\mathsf{H}} \right\|_F \left\| (\boldsymbol{Z}_t \boldsymbol{Z}_t^\top)^\dagger \right\|_2 \left\| \boldsymbol{U}_t \boldsymbol{U}_t^\mathsf{H} \nabla F(\boldsymbol{X}_t) \left( \boldsymbol{I}_n - \overline{\boldsymbol{U}_t \boldsymbol{U}_t^\mathsf{H}} \right) \right\|_F$$

$$+ \frac{\alpha^2}{2} \left\| \left( \boldsymbol{I}_n - \boldsymbol{U}_t \boldsymbol{U}_t^\mathsf{H} \right) \nabla F(\boldsymbol{X}_t) \overline{\boldsymbol{U}_t \boldsymbol{U}_t^\mathsf{H}} \right\|_F \left\| (\boldsymbol{Z}_t \boldsymbol{Z}_t^\top)^\dagger \right\|_2 \left\| \boldsymbol{U}_t \boldsymbol{U}_t^\mathsf{H} \nabla F(\boldsymbol{X}_t) \overline{\boldsymbol{U}_t \boldsymbol{U}_t^\mathsf{H}} \right\|_F$$

$$+ \frac{\alpha^2}{2} \left\| \boldsymbol{U}_t \boldsymbol{U}_t^\mathsf{H} \nabla F(\boldsymbol{X}_t) \overline{\boldsymbol{U}_t \boldsymbol{U}_t^\mathsf{H}} \right\|_F \left\| (\boldsymbol{Z}_t \boldsymbol{Z}_t^\top)^\dagger \right\|_2 \left\| \boldsymbol{U}_t \boldsymbol{U}_t^\mathsf{H} \nabla F(\boldsymbol{X}_t) \left( \boldsymbol{I}_n - \overline{\boldsymbol{U}_t \boldsymbol{U}_t^\mathsf{H}} \right) \right\|_F$$

$$+ \frac{\alpha^2}{4} \left\| \boldsymbol{U}_t \boldsymbol{U}_t^\mathsf{H} \nabla F(\boldsymbol{X}_t) \overline{\boldsymbol{U}_t \boldsymbol{U}_t^\mathsf{H}} \right\|_F \left\| (\boldsymbol{Z}_t \boldsymbol{Z}_t^\top)^\dagger \right\|_2 \left\| \boldsymbol{U}_t \boldsymbol{U}_t^\mathsf{H} \nabla F(\boldsymbol{X}_t) \overline{\boldsymbol{U}_t \boldsymbol{U}_t^\mathsf{H}} \right\|_F$$

$$\leq \frac{\alpha^2}{2\sigma_r(\boldsymbol{X}_t)} \left( \left\| \left( \boldsymbol{I}_n - \boldsymbol{U}_t \boldsymbol{U}_t^\mathsf{H} \right) \nabla F(\boldsymbol{X}_t) \overline{\boldsymbol{U}_t \boldsymbol{U}_t^\mathsf{H}} \right\|_F^2 + \left\| \boldsymbol{U}_t \boldsymbol{U}_t^\mathsf{H} \nabla F(\boldsymbol{X}_t) \left( \boldsymbol{I}_n - \overline{\boldsymbol{U}_t \boldsymbol{U}_t^\mathsf{H}} \right) \right\|_F^2 \right)$$

$$+ \frac{\alpha^2}{4\sigma_r(\boldsymbol{X}_t)} \left( \left\| \left( \boldsymbol{I}_n - \boldsymbol{U}_t \boldsymbol{U}_t^\mathsf{H} \right) \nabla F(\boldsymbol{X}_t) \overline{\boldsymbol{U}_t \boldsymbol{U}_t^\mathsf{H}} \right\|_F^2 + \left\| \boldsymbol{U}_t \boldsymbol{U}_t^\mathsf{H} \nabla F(\boldsymbol{X}_t) \overline{\boldsymbol{U}_t \boldsymbol{U}_t^\mathsf{H}} \right\|_F^2 \right)$$

$$+ \frac{\alpha^2}{4\sigma_r(\boldsymbol{X}_t)} \left( \left\| \boldsymbol{U}_t \boldsymbol{U}_t^\mathsf{H} \nabla F(\boldsymbol{X}_t) \overline{\boldsymbol{U}_t \boldsymbol{U}_t^\mathsf{H}} \right\|_F^2 + \left\| \boldsymbol{U}_t \boldsymbol{U}_t^\mathsf{H} \nabla F(\boldsymbol{X}_t) \left( \boldsymbol{I}_n - \overline{\boldsymbol{U}_t \boldsymbol{U}_t^\mathsf{H}} \right) \right\|_F^2 \right) + \frac{\alpha^2}{4\sigma_r(\boldsymbol{X}_t)} \left\| \boldsymbol{U}_t \boldsymbol{U}_t^\mathsf{H} \nabla F(\boldsymbol{X}_t) \overline{\boldsymbol{U}_t \boldsymbol{U}_t^\mathsf{H}} \right\|_F^2$$

$$= \frac{\alpha^2}{\sigma_r(\boldsymbol{X}_t)} \left\{ \frac{3}{4} \left\| \left( \boldsymbol{I}_n - \boldsymbol{U}_t \boldsymbol{U}_t^\mathsf{H} \right) \nabla F(\boldsymbol{X}_t) \overline{\boldsymbol{U}_t \boldsymbol{U}_t^\mathsf{H}} \right\|_F^2 + \frac{3}{4} \left\| \boldsymbol{U}_t \boldsymbol{U}_t^\mathsf{H} \nabla F(\boldsymbol{X}_t) \left( \boldsymbol{I}_n - \overline{\boldsymbol{U}_t \boldsymbol{U}_t^\mathsf{H}} \right) \right\|_F^2 + \frac{3}{4} \left\| \boldsymbol{U}_t \boldsymbol{U}_t^\mathsf{H} \nabla F(\boldsymbol{X}_t) \overline{\boldsymbol{U}_t \boldsymbol{U}_t^\mathsf{H}} \right\|_F^2 \right\}$$

$$= \frac{3}{4\sigma_r(\boldsymbol{X}_t)} \| \boldsymbol{L}_t - \boldsymbol{X}_t \|_F^2.$$

The first inequality applies the triangle inequality. The first equality utilizes the following properties,

$$\boldsymbol{Z}^\dagger \boldsymbol{U} \boldsymbol{U}^\mathsf{H} = \boldsymbol{Z}^\dagger \boldsymbol{Z} \boldsymbol{Z}^\dagger = \boldsymbol{Z}^\dagger,$$

$$\overline{\boldsymbol{U} \boldsymbol{U}^\mathsf{H}} (\boldsymbol{Z}^\top)^\dagger = \overline{\boldsymbol{Z}(\boldsymbol{Z}^\mathsf{H} \boldsymbol{Z})^{-1} \boldsymbol{Z}^\mathsf{H}} [\boldsymbol{Z}(\boldsymbol{Z}^\mathsf{H} \boldsymbol{Z})^{-1} \boldsymbol{Z}^\mathsf{H}]^\top = (\boldsymbol{Z}^\dagger)^\top = (\boldsymbol{Z}^\top)^\dagger,$$

and then

$$(\boldsymbol{Z} \boldsymbol{Z}^\top)^\dagger = (\boldsymbol{Z}^\top)^\dagger \boldsymbol{Z}^\dagger = \overline{\boldsymbol{U} \boldsymbol{U}^\mathsf{H}} (\boldsymbol{Z} \boldsymbol{Z}^\top)^\dagger \boldsymbol{U} \boldsymbol{U}^\mathsf{H}.$$

The third inequality uses the AMGM inequality. $\qquad\square$

**Corollary B.1.** If $\|\mathcal{G}\boldsymbol{y} - \boldsymbol{X}_t\| \leq q\sigma_r(\mathcal{G}\boldsymbol{y})$ for some given $0 < q < 1$, we further extend the bound in Theorem B.1 to the following,

$$\|\boldsymbol{X}_{t+1} - \boldsymbol{L}_t\|_F \leq \frac{3}{4(1-q)\sigma_r(\mathcal{G}\boldsymbol{y})} \|\boldsymbol{L}_t - \boldsymbol{X}_t\|_F^2. \tag{31}$$

*Proof of Corollary B.1.* For $\boldsymbol{A}, \boldsymbol{B} \in \mathbb{C}^{m \times n}$. One basic inequality for singular values is $\sigma_{i+j-1}(\boldsymbol{A}+\boldsymbol{B}) \leq \sigma_i(\boldsymbol{A}) + \sigma_j(\boldsymbol{B})$ if $1 \leq i, j \leq q$ and $i + j \leq q + 1$. In our case, $m = n$, $q = r$, substitute $\boldsymbol{B} = \mathcal{G}\boldsymbol{y} - \boldsymbol{X}_t$, $\boldsymbol{A} = \boldsymbol{X}_t$, then $\boldsymbol{A} + \boldsymbol{B} = \mathcal{G}\boldsymbol{y}$. Consider $i = r$ and $j = 1$, we obtain,

$$\sigma_r(\mathcal{G}\boldsymbol{y}) \leq \sigma_r(\boldsymbol{X}_t) + \sigma_1(\mathcal{G}\boldsymbol{y} - \boldsymbol{X}_t). \tag{32}$$

Under the assumption $\|\mathcal{G}\boldsymbol{y} - \boldsymbol{X}_t\| \leq q\sigma_r(\mathcal{G}\boldsymbol{y})$ for $0 < q < 1$, we have,

$$\sigma_r(\boldsymbol{X}_t) \geq \sigma_r(\mathcal{G}\boldsymbol{y}) - \|\mathcal{G}\boldsymbol{y} - \boldsymbol{X}_t\| \geq (1-q)\sigma_r(\mathcal{G}\boldsymbol{y}) > 0. \tag{33}$$

Then substituting the above bound into the bound in Theorem B.1, we obtain

$$\|\boldsymbol{X}_{t+1} - \boldsymbol{L}_t\|_F \leq \frac{3}{4\sigma_r(\boldsymbol{X}_t)} \|\boldsymbol{L}_t - \boldsymbol{X}_t\|_F^2 \leq \frac{3}{4(1-q)\sigma_r(\mathcal{G}\boldsymbol{y})} \|\boldsymbol{L}_t - \boldsymbol{X}_t\|_F^2.$$

$\qquad\square$

**Remark B.1.** Throughout the whole paper, when proving the convergence of the algorithm, we set $q = 0.1$.

Theorem B.2, stated below, serves as a key ingredient in the proof of Theorem 3.1.

**Theorem B.2.** *Assume* $0 < \epsilon_0 < 0.0352(< \frac{1}{28})$,

$$\frac{0.4\epsilon_0 + 0.5}{0.5 - 4\epsilon_0 - \frac{3\epsilon_0}{1+\epsilon_0}} < \alpha < \frac{1.5 - 0.4\epsilon_0}{\frac{3\epsilon_0}{1+\epsilon_0} + 4\epsilon_0 + 0.5}, \tag{34}$$

*and the following conditions*

$$\|\mathcal{P}_\Omega\| \le 8\log(n_s), \tag{35}$$

$$\left\|\mathcal{P}_\mathbb{T}\mathcal{G}\mathcal{G}^*\mathcal{P}_\mathbb{T} - p^{-1}\mathcal{P}_\mathbb{T}\mathcal{G}\mathcal{P}_\Omega\mathcal{G}^*\mathcal{P}_\mathbb{T}\right\| \le \epsilon_0, \tag{36}$$

$$\frac{\|X_0 - \mathcal{G}\boldsymbol{y}\|_F}{\sigma_{\min}(\mathcal{G}\boldsymbol{y})} \le \frac{p^{\frac{1}{2}}\epsilon_0}{\sqrt{8\log(n_s)(1+\epsilon_0)}} \tag{37}$$

*are satisfied.* $p = \frac{m}{n_s}$ *refers to the oversampling ratio,* $\alpha$ *is the step size. Then the iterate* $\boldsymbol{y}_t$ *in (29) satisfies* $\|\boldsymbol{y}_t - \boldsymbol{y}\| \le \nu^t\|X_0 - \mathcal{G}\boldsymbol{y}\|_F$ *with* $\nu < 1$.

**Lemma B.8.** *Suppose (35) and (36) hold and*

$$\frac{\|X_t - \mathcal{G}\boldsymbol{y}\|_F}{\sigma_{\min}(\mathcal{G}\boldsymbol{y})} \le \frac{p^{\frac{1}{2}}\epsilon_0}{\sqrt{8\log(n_s)(1+\epsilon_0)}}. \tag{38}$$

*Then we have*

$$\|\mathcal{P}_\Omega\mathcal{G}^*\mathcal{P}_{\mathbb{T}_t}\| \le \left(\frac{2\epsilon_0}{1+\epsilon_0} + 1\right)\sqrt{8\log(n_s)(1+\epsilon_0)p} \tag{39}$$

*and*

$$\left\|\mathcal{P}_{\mathbb{T}_t}\mathcal{G}\mathcal{G}^*\mathcal{P}_{\mathbb{T}_t} - p^{-1}\mathcal{P}_{\mathbb{T}_t}\mathcal{G}\mathcal{P}_\Omega\mathcal{G}^*\mathcal{P}_{\mathbb{T}_t}\right\| \le 6.6\epsilon_0 + \frac{4\epsilon_0}{1+\epsilon_0}. \tag{40}$$

*Proof of Lemma B.8.* Since $\|\mathcal{P}_\mathbb{T}\mathcal{G}\mathcal{P}_\Omega\| = \|(\mathcal{P}_\mathbb{T}\mathcal{G}\mathcal{P}_\Omega)^*\| = \|\mathcal{P}_\Omega\mathcal{G}^*\mathcal{P}_\mathbb{T}\|$, for any $\boldsymbol{W} \in \mathbb{C}^{n\times n}$,

$$\begin{aligned}
\|\mathcal{P}_\Omega\mathcal{G}^*\mathcal{P}_\mathbb{T}(\boldsymbol{W})\|^2 &= \langle \mathcal{P}_\Omega\mathcal{G}^*\mathcal{P}_\mathbb{T}(\boldsymbol{W}), \mathcal{P}_\Omega\mathcal{G}^*\mathcal{P}_\mathbb{T}(\boldsymbol{W})\rangle \\
&\le 8\log(n_s)\langle \mathcal{G}^*\mathcal{P}_\mathbb{T}(\boldsymbol{W}), \mathcal{P}_\Omega\mathcal{G}^*\mathcal{P}_\mathbb{T}(\boldsymbol{W})\rangle \\
&= 8\log(n_s)\langle \boldsymbol{W}, \mathcal{P}_\mathbb{T}\mathcal{G}\mathcal{P}_\Omega\mathcal{G}^*\mathcal{P}_\mathbb{T}(\boldsymbol{W})\rangle \\
&\le 8\log(n_s)\|\boldsymbol{W}\|_F^2\|\mathcal{P}_\mathbb{T}\mathcal{G}\mathcal{P}_\Omega\mathcal{G}^*\mathcal{P}_\mathbb{T}\| \\
&\le 8\log(n_s)(1+\epsilon_0)p\|\boldsymbol{W}\|_F^2,
\end{aligned}$$

where the first inequality follows (35), and the second inequality utilizes (36), $p^{-1}\|\mathcal{P}_\mathbb{T}\mathcal{G}\mathcal{P}_\Omega\mathcal{G}^*\mathcal{P}_\mathbb{T}\|_2 \le \epsilon_0 + \|\mathcal{P}_\mathbb{T}\mathcal{G}\mathcal{G}^*\mathcal{P}_\mathbb{T}\|_2 \le \epsilon_0 + 1$. Then we have, $\|\mathcal{P}_\mathbb{T}\mathcal{G}\mathcal{P}_\Omega\| = \|\mathcal{P}_\Omega\mathcal{G}^*\mathcal{P}_\mathbb{T}\| \le \sqrt{8\log(n_s)(1+\epsilon_0)p}$ and,

$$\begin{aligned}
\|\mathcal{P}_\Omega\mathcal{G}^*\mathcal{P}_{\mathbb{T}_t}\| &\le \|\mathcal{P}_\Omega\mathcal{G}^*(\mathcal{P}_\mathbb{T} - \mathcal{P}_{\mathbb{T}_t})\| + \|\mathcal{P}_\Omega\mathcal{G}^*\mathcal{P}_\mathbb{T}\| \\
&\le 8\log(n_s)\|\mathcal{P}_\mathbb{T} - \mathcal{P}_{\mathbb{T}_t}\| + \|\mathcal{P}_\Omega\mathcal{G}^*\mathcal{P}_\mathbb{T}\| \\
&\le 8\log(n_s)\frac{2\|X_t - \mathcal{G}\boldsymbol{y}\|_F}{\sigma_{\min}(\mathcal{G}\boldsymbol{y})} + \sqrt{8\log(n_s)(1+\epsilon_0)p} \\
&\le 8\log(n_s)\frac{2p^{\frac{1}{2}}\epsilon_0}{\sqrt{8\log(n_s)(1+\epsilon_0)}} + \sqrt{8\log(n_s)(1+\epsilon_0)p} \\
&= 2\sqrt{8\log(n_s)p} \cdot \frac{\epsilon_0}{\sqrt{1+\epsilon_0}} + \sqrt{8\log(n_s)(1+\epsilon_0)p} \\
&= \left(\frac{2\epsilon_0}{1+\epsilon_0} + 1\right)\sqrt{8\log(n_s)(1+\epsilon_0)p},
\end{aligned}$$

where the second inequality follows from (35), the third inequality follows Lemma B.5, and the forth inequality follows from (38). So finally we have,

$$
\begin{aligned}
&\left\| \mathcal{P}_{\mathbb{T}_t} \mathcal{G} \mathcal{G}^* \mathcal{P}_{\mathbb{T}_t} - p^{-1} \mathcal{P}_{\mathbb{T}_t} \mathcal{G} \mathcal{P}_\Omega \mathcal{G}^* \mathcal{P}_{\mathbb{T}_t} \right\| \\
&\leq \left\| \mathcal{P}_{\mathbb{T}} \mathcal{G} \mathcal{G}^* \mathcal{P}_{\mathbb{T}} - p^{-1} \mathcal{P}_{\mathbb{T}} \mathcal{G} \mathcal{P}_\Omega \mathcal{G}^* \mathcal{P}_{\mathbb{T}} \right\| + \left\| (\mathcal{P}_{\mathbb{T}} - \mathcal{P}_{\mathbb{T}_t}) \mathcal{G} \mathcal{G}^* \mathcal{P}_{\mathbb{T}_t} \right\| + \left\| \mathcal{P}_{\mathbb{T}} \mathcal{G} \mathcal{G}^* (\mathcal{P}_{\mathbb{T}} - \mathcal{P}_{\mathbb{T}_t}) \right\| \\
&\quad + \left\| p^{-1} (\mathcal{P}_{\mathbb{T}} - \mathcal{P}_{\mathbb{T}_t}) \mathcal{G} \mathcal{P}_\Omega \mathcal{G}^* \mathcal{P}_{\mathbb{T}_t} \right\| + \left\| p^{-1} \mathcal{P}_{\mathbb{T}} \mathcal{G} \mathcal{P}_\Omega \mathcal{G}^* (\mathcal{P}_{\mathbb{T}} - \mathcal{P}_{\mathbb{T}_t}) \right\| \\
&\leq \epsilon_0 + \frac{4 \| \boldsymbol{X}_t - \mathcal{G} \boldsymbol{y} \|_F}{\sigma_{\min}(\mathcal{G} \boldsymbol{y})} + p^{-1} \cdot \frac{2 \| \boldsymbol{X}_t - \mathcal{G} \boldsymbol{y} \|_F}{\sigma_{\min}(\mathcal{G} \boldsymbol{y})} \cdot \left( \| \mathcal{P}_\Omega \mathcal{G}^* \mathcal{P}_{\mathbb{T}_t} \| + \| \mathcal{P}_{\mathbb{T}} \mathcal{G} \mathcal{P}_\Omega \| \right) \\
&\leq \epsilon_0 + \frac{4 p^{\frac{1}{2}} \epsilon_0}{\sqrt{8 \log(n_s)(1 + \epsilon_0)}} + p^{-1} \cdot \frac{2 p^{\frac{1}{2}} \epsilon_0 \left( \frac{2\epsilon_0}{1 + \epsilon_0} + 2 \right)}{\sqrt{8 \log(n_s)(1 + \epsilon_0)}} \cdot \sqrt{8 \log(n_s)(1 + \epsilon_0) p} \\
&\leq \epsilon_0 + \frac{4 \epsilon_0}{2.5} + 2 \epsilon_0 \left( \frac{2 \epsilon_0}{1 + \epsilon_0} + 2 \right) \\
&\leq 6.6 \epsilon_0 + \frac{4 \epsilon_0}{1 + \epsilon_0},
\end{aligned}
$$

where the forth inequality is deducted from: $2.5 p^{\frac{1}{2}} \leq 2.5 \leq \sqrt{8 \log(3)} \leq \sqrt{8 \log(n_s)(1 + \epsilon_0)}$ for $p = \frac{m}{n_s} \leq 1$, $0 < \epsilon_0 < 1$, and $n_s \geq 3$. The last inequality follows $\frac{\epsilon_0^2}{1 + \epsilon_0} \leq \frac{\epsilon_0}{1 + \epsilon_0}$ when $0 < \epsilon_0 < 1$.

$\square$

**Theorem B.3.** *Assume $0 < \epsilon_0 < \frac{1}{28}$, and conditions (34) (38) are satisfied. $p = \frac{m}{n_s}$ refers to the oversampling ratio, $\alpha$ is the step size. Then*

$$
\| \boldsymbol{L}_t - \mathcal{G} \boldsymbol{y} \|_F \leq c \| \boldsymbol{X}_t - \mathcal{G} \boldsymbol{y} \|_F, \tag{41}
$$

*where $c < 0.5$ is a constant.*

*Proof of Theorem B.3.*

$$
\begin{aligned}
\| \boldsymbol{L}_t - \mathcal{G} \boldsymbol{y} \|_F &= \left\| \boldsymbol{X}_t - \alpha \mathcal{P}_{\mathbb{T}_t} \left[ \frac{1}{2p} \mathcal{G} \mathcal{P}_\Omega \left( \mathcal{G}^*(\boldsymbol{X}_t) - \boldsymbol{y} \right) + \frac{1}{2} \left( \mathcal{I} - \mathcal{G} \mathcal{G}^* \right) (\boldsymbol{X}_t) \right] - \mathcal{G} \boldsymbol{y} \right\|_F \\
&\leq \| (\mathcal{I} - \mathcal{P}_{\mathbb{T}_t})(\boldsymbol{X}_t - \mathcal{G} \boldsymbol{y}) \|_F + \left| 1 - \frac{\alpha}{2} \right| \cdot \| \mathcal{P}_{\mathbb{T}_t}(\boldsymbol{X}_t - \mathcal{G} \boldsymbol{y}) \|_F \\
&\quad + \left| \frac{\alpha}{2} \right| \cdot \| (\mathcal{P}_{\mathbb{T}_t} \mathcal{G} \mathcal{G}^* \mathcal{P}_{\mathbb{T}_t} - p^{-1} \mathcal{P}_{\mathbb{T}_t} \mathcal{G} \mathcal{P}_\Omega \mathcal{G}^* \mathcal{P}_{\mathbb{T}_t})(\boldsymbol{X}_t - \mathcal{G} \boldsymbol{y}) \|_F \\
&\quad + \left| \frac{\alpha}{2} \right| \cdot \| \mathcal{P}_{\mathbb{T}_t} \mathcal{G} \mathcal{G}^* (\mathcal{I} - \mathcal{P}_{\mathbb{T}_t})(\boldsymbol{X}_t - \mathcal{G} \boldsymbol{y}) \|_F \\
&\quad + \left| \frac{\alpha}{2} \right| \cdot p^{-1} \cdot \| \mathcal{P}_{\mathbb{T}_t} \mathcal{G} \mathcal{P}_\Omega \mathcal{G}^* (\mathcal{I} - \mathcal{P}_{\mathbb{T}_t})(\boldsymbol{X}_t - \mathcal{G} \boldsymbol{y}) \|_F \\
&:= I_1 + I_2 + I_3 + I_4 + I_5,
\end{aligned}
$$

where the inequality utilizes $(\mathcal{I} - \mathcal{P}_{\mathbb{T}_t}) \boldsymbol{X}_t = \boldsymbol{0}$, and $\mathcal{G}^* \mathcal{G} = \mathcal{I}$. Then we analyze the upper bounds for each $I_i$ separately. Assume (38) holds, then apply Lemma B.5, we obtain

$$
I_1 \leq \frac{\| \boldsymbol{X}_t - \mathcal{G} \boldsymbol{y} \|_F^2}{\sigma_{\min}(\mathcal{G} \boldsymbol{y})} \leq \frac{p^{\frac{1}{2}} \epsilon_0}{\sqrt{8 \log(n_s)(1 + \epsilon_0)}} \| \boldsymbol{X}_t - \mathcal{G} \boldsymbol{y} \|_F,
$$

$$
I_2 \leq \left| 1 - \frac{\alpha}{2} \right| \cdot \| \boldsymbol{X}_t - \mathcal{G} \boldsymbol{y} \|_F,
$$

$$
I_4 \leq \left| \frac{\alpha}{2} \right| \cdot \| (\mathcal{I} - \mathcal{P}_{\mathbb{T}_t})(\boldsymbol{X}_t - \mathcal{G} \boldsymbol{y}) \|_F \leq \left| \frac{\alpha}{2} \right| \cdot \frac{p^{\frac{1}{2}} \epsilon_0}{\sqrt{8 \log(n_s)(1 + \epsilon_0)}} \| \boldsymbol{X}_t - \mathcal{G} \boldsymbol{y} \|_F.
$$

Applying Lemma B.8 and $\| \mathcal{G}^* \|_2 \leq 1$, we have,

$$
\begin{aligned}
I_3 &\leq \left| \frac{\alpha}{2} \right| \cdot \| \mathcal{P}_{\mathbb{T}_t} \mathcal{G} \mathcal{G}^* \mathcal{P}_{\mathbb{T}_t} - p^{-1} \mathcal{P}_{\mathbb{T}_t} \mathcal{G} \mathcal{P}_\Omega \mathcal{G}^* \mathcal{P}_{\mathbb{T}_t} \|_2 \| \boldsymbol{X}_t - \mathcal{G} \boldsymbol{y} \|_F \\
&\leq \left| \frac{\alpha}{2} \right| \cdot \left( 6.6 \epsilon_0 + \frac{4 \epsilon_0}{1 + \epsilon_0} \right) \| \boldsymbol{X}_t - \mathcal{G} \boldsymbol{y} \|_F,
\end{aligned}
$$

and

$$I_5 \leq \left|\frac{\alpha}{2}\right| \cdot p^{-1} \cdot \|\mathcal{P}_{\mathbb{T}_t}\mathcal{G}\mathcal{P}_\Omega\|_2 \|(\mathcal{I} - \mathcal{P}_{\mathbb{T}_t})(\boldsymbol{X}_t - \mathcal{G}\boldsymbol{y})\|_F$$

$$\leq \left|\frac{\alpha}{2}\right| \cdot p^{-1} \cdot \left(\frac{2\epsilon_0}{1 + \epsilon_0} + 1\right) \sqrt{8\log(n_s)(1 + \epsilon_0)p} \cdot \frac{p^{\frac{1}{2}}\epsilon_0}{\sqrt{8\log(n_s)(1 + \epsilon_0)}} \|\boldsymbol{X}_t - \mathcal{G}\boldsymbol{y}\|_F$$

$$= \left|\frac{\alpha}{2}\right| \cdot \left(\frac{2\epsilon_0}{1 + \epsilon_0} + 1\right) \epsilon_0 \|\boldsymbol{X}_t - \mathcal{G}\boldsymbol{y}\|_F$$

$$\leq \left|\frac{\alpha}{2}\right| \cdot \left(\frac{2\epsilon_0}{1 + \epsilon_0} + \epsilon_0\right) \|\boldsymbol{X}_t - \mathcal{G}\boldsymbol{y}\|_F.$$

Then combine all of above, we have

$$\|\boldsymbol{L}_t - \mathcal{G}\boldsymbol{y}\|_F \leq I_1 + I_2 + I_3 + I_4 + I_5$$

$$\leq \frac{p^{\frac{1}{2}}\epsilon_0}{\sqrt{8\log(n_s)(1 + \epsilon_0)}} \|\boldsymbol{X}_t - \mathcal{G}\boldsymbol{y}\|_F + \left|1 - \frac{\alpha}{2}\right| \|\boldsymbol{X}_t - \mathcal{G}\boldsymbol{y}\|_F + \left|\frac{\alpha}{2}\right| \cdot \left(6.6\epsilon_0 + \frac{4\epsilon_0}{1 + \epsilon_0}\right) \|(\boldsymbol{X}_t - \mathcal{G}\boldsymbol{y})\|_F$$

$$+ \left|\frac{\alpha}{2}\right| \cdot \frac{p^{\frac{1}{2}}\epsilon_0}{\sqrt{8\log(n_s)(1 + \epsilon_0)}} \|\boldsymbol{X}_t - \mathcal{G}\boldsymbol{y}\|_F + \left|\frac{\alpha}{2}\right| \cdot \left(\frac{2\epsilon_0}{1 + \epsilon_0} + \epsilon_0\right) \|\boldsymbol{X}_t - \mathcal{G}\boldsymbol{y}\|_F$$

$$\leq \left\{\frac{\epsilon_0}{2.5} + \left|1 - \frac{\alpha}{2}\right| + \left|\frac{\alpha}{2}\right| \cdot \left(6.6\epsilon_0 + \frac{4\epsilon_0}{1 + \epsilon_0}\right) + \left|\frac{\alpha}{2}\right| \cdot \frac{\epsilon_0}{2.5} \left|\frac{\alpha}{2}\right| \cdot \left(\frac{2\epsilon_0}{1 + \epsilon_0} + \epsilon_0\right)\right\} \|\boldsymbol{X}_t - \mathcal{G}\boldsymbol{y}\|_F$$

$$= \left(\frac{\epsilon_0}{2.5} + \left|1 - \frac{\alpha}{2}\right| + 3\alpha\frac{\epsilon_0}{1 + \epsilon_0} + 4\alpha\epsilon_0\right) \|\boldsymbol{X}_t - \mathcal{G}\boldsymbol{y}\|_F$$

$$< 0.5\|\boldsymbol{X}_t - \mathcal{G}\boldsymbol{y}\|_F,$$

where the last inequality is true for $\epsilon_0 \in (0, 0.0352)$ and

$$\alpha \in \left(\frac{0.4\epsilon_0 + 0.5}{0.5 - 4\epsilon_0 - \frac{3\epsilon_0}{1+\epsilon_0}}, \frac{1.5 - 0.4\epsilon_0}{\frac{3\epsilon_0}{1+\epsilon_0} + 4\epsilon_0 + 0.5}\right).$$

For simplicity, given the range of $\epsilon_0$, the range of $\alpha$ is $(1, 3)$. $\qquad\square$

*Proof of Theorem B.2.* Note that the iteration $\boldsymbol{X}_{t+1} = \mathcal{R}(\boldsymbol{L}_t)$, where

$$\boldsymbol{L}_t := \boldsymbol{X}_t - \alpha\mathcal{P}_{\mathbb{T}_t}(\nabla F(\boldsymbol{X}_t)) = \boldsymbol{X}_t - \alpha\mathcal{P}_{\mathbb{T}_t}\left[\frac{1}{2p}\mathcal{G}\mathcal{P}_\Omega\left(\mathcal{G}^*(\boldsymbol{X}_t) - \boldsymbol{y}\right) + \frac{1}{2}\left(\mathcal{I} - \mathcal{G}\mathcal{G}^*\right)(\boldsymbol{X}_t)\right].$$

Besides, from Corollary B.1 with $q = 0.1$ and Theorem B.3, we have,

$$\|\boldsymbol{X}_{t+1} - \boldsymbol{L}_t\|_F \leq c'\|\boldsymbol{L}_t - \boldsymbol{X}_t\|_F^2,$$

where $c' = \frac{3}{4(1-q)\sigma_{\min}(\mathcal{G}\boldsymbol{y})}$, and

$$\|\boldsymbol{L}_t - \mathcal{G}\boldsymbol{y}\|_F \leq c\|\boldsymbol{X}_t - \mathcal{G}\boldsymbol{y}\|_F,$$

where $c < 0.5$.

So we have

$$
\begin{aligned}
\|\boldsymbol{X}_{t+1} - \mathcal{G}\boldsymbol{y}\|_F &\leq \|\boldsymbol{X}_{t+1} - \boldsymbol{L}_t\|_F + \|\boldsymbol{L}_t - \mathcal{G}\boldsymbol{y}\|_F \\
&\leq c'\|\boldsymbol{L}_t - \boldsymbol{X}_t\|_F^2 + c\|\boldsymbol{X}_t - \mathcal{G}\boldsymbol{y}\|_F \\
&\leq c' \left(2\|\boldsymbol{L}_t - \mathcal{G}\boldsymbol{y}\|_F^2 + 2\|\boldsymbol{X}_t - \mathcal{G}\boldsymbol{y}\|_F^2\right) + c\|\boldsymbol{X}_t - \mathcal{G}\boldsymbol{y}\|_F \\
&\leq c' \left(2c^2\|\boldsymbol{X}_t - \mathcal{G}\boldsymbol{y}\|_F^2 + 2\|\boldsymbol{X}_t - \mathcal{G}\boldsymbol{y}\|_F^2\right) + c\|\boldsymbol{X}_t - \mathcal{G}\boldsymbol{y}\|_F \\
&= \left[2c'(c^2 + 1)\|\boldsymbol{X}_t - \mathcal{G}\boldsymbol{y}\|_F + c\right]\|\boldsymbol{X}_t - \mathcal{G}\boldsymbol{y}\|_F \\
&\leq \left(2 \cdot \frac{3}{4(1-q)\sigma_{\min}(\mathcal{G}\boldsymbol{y})} \cdot 1.25 \cdot \|\boldsymbol{X}_t - \mathcal{G}\boldsymbol{y}\|_F + 0.5\right)\|\boldsymbol{X}_t - \mathcal{G}\boldsymbol{y}\|_F \\
&\leq \left(1.5 \cdot \frac{1}{1-q} \cdot 1.25 \cdot \frac{\epsilon_0}{2.5} + 0.5\right)\|\boldsymbol{X}_t - \mathcal{G}\boldsymbol{y}\|_F \\
&\leq \left(1.5 \cdot \frac{1}{1-q} \cdot 1.25 \cdot \frac{1}{28} \cdot 0.4 + 0.5\right)\|\boldsymbol{X}_t - \mathcal{G}\boldsymbol{y}\|_F \\
&= \nu\|\boldsymbol{X}_t - \mathcal{G}\boldsymbol{y}\|_F,
\end{aligned}
$$

where the sixth inequality follows (38), the seventh inequality follows $\epsilon_0 < 0.0352 < \frac{1}{28}$ and the last inequality is true with $0 < \nu < 1$ with $q = 0.1$.
Therefore, we've proved

$$
\|\boldsymbol{X}_{t+1} - \mathcal{G}\boldsymbol{y}\|_F \leq \nu\|\boldsymbol{X}_t - \mathcal{G}\boldsymbol{y}\|_F,
$$

where $0 < \nu < 1$.
Since (38) holds for $t = 0$ by the assumption of Theorem B.2, and $\|\boldsymbol{X}_t - \mathcal{G}\boldsymbol{y}\|_F$ is a contractive sequence, which concludes (38) is true for all $t \geq 0$. Therefore,

$$
\|\boldsymbol{x}_t - \boldsymbol{x}\| = \|\mathcal{D}^{-1}(\boldsymbol{y}_t - \boldsymbol{y})\| \leq \|\boldsymbol{y}_t - \boldsymbol{y}\| = \|\mathcal{G}^*(\boldsymbol{X}_{t+1} - \mathcal{G}\boldsymbol{y})\| \leq \|\boldsymbol{X}_{t+1} - \mathcal{G}\boldsymbol{y}\|_F \leq \nu^t\|\boldsymbol{X}_0 - \mathcal{G}\boldsymbol{y}\|_F.
$$

$\square$

**Proof of Theorem 3.1:**

*Proof.* Since we have $\|\boldsymbol{x}_t - \boldsymbol{x}\| = \|\mathcal{D}^{-1}(\boldsymbol{y}_t - \boldsymbol{y})\| \leq \|\boldsymbol{y}_t - \boldsymbol{y}\|$, the remaining part is just to verify the validity of the three conditions in Theorem B.2 and condition on $q$ in Corollary B.1. The lemma B.3 guarantees (35) hold with probability at least $1 - n_s^{-2}$. If $m \geq C_2\epsilon_0^{-2}\mu_0 r \log(n_s)$ with a sufficiently large constant $C_2 > 0$, (36) is true with probability at least $1 - n_s^{-2}$ followed by Lemma B.2 and B.4. Inequality (37) holds with probability at least $1 - n_s^{-2}$ if $m \geq C_3(1 + \epsilon_0)^{\frac{1}{2}}\epsilon_0^{-1}\mu_0^{\frac{1}{2}} r\kappa n_s^{\frac{1}{2}} \log(n_s)$ following Lemma B.7 and fact $\|\boldsymbol{X}_0 - \mathcal{G}\boldsymbol{y}\|_F \leq \sqrt{2r}\|\boldsymbol{X}_0 - \mathcal{G}\boldsymbol{y}\|_2$. In addition,

$$
\|\boldsymbol{X}_0 - \mathcal{G}\boldsymbol{y}\| \leq C_1\sqrt{\frac{2\mu_0 r \log(n_s)}{m}}\|\mathcal{G}\boldsymbol{y}\| \leq 0.1\sigma_{\min}(\mathcal{G}\boldsymbol{y})
$$

is true if $m \geq C_4\mu_0 r \log(n_s)\kappa^2$. Finally, taking an upper bound on the number of measurements completes the proof of Theorem 3.1.  $\square$

## B.3. Proof of Theorem 3.2

### B.3.1. RESAMPLING AND TRIMMING

The algorithm is designed around two central principles to ensure robust convergence.

1. Independence via Resampling: To mitigate the statistical dependencies that typically complicate the analysis of non-convex iterations, we utilize a fresh batch of samples for every update step. This independence between the sampling operator and the current iterate allows for the application of standard concentration inequalities, yielding sharper estimates of the sample complexity.

2. Incoherence via Trimming: To prevent the iterates from becoming spiky or aligned with the canonical basiswhich would degrade the effectiveness of samplingwe apply a trimming step. This operation projects the current estimate onto the set of $\mu_0$-incoherent matrices, actively maintaining the necessary structural properties for recovery.

---

**Algorithm 3** Initialization via Resampled JHGD and Trimming in Factor Space

---

1: **Preprocessing:**

    1. Make each dimension of the signal odd by zero-padding.

    2. Partition $\Omega$ into disjoint sets $\Omega_0, \cdots, \Omega_T$ of equal size $\widehat{m}$, let $\widehat{p} = \frac{\widehat{m}}{n_s}$.

2: **Set:** $\widetilde{X}_0 = \mathcal{T}_r\left(\widehat{p}^{-1}\mathcal{G}\mathcal{P}_{\Omega_0}(\boldsymbol{y})\right) = \widetilde{U}_0\widetilde{\Sigma}_0\widetilde{U}_0^\top$, and $\widetilde{Z}_0 = \widetilde{U}_0\widetilde{\Sigma}_0^{1/2}$.

3: **for** $t = 0, 1, \cdots, K$ **do**

    1. $\widetilde{Z}_t = Q_t R_t$.

    2. $R_t R_t^\top = W_t \widetilde{\Sigma}_t W_t^\top$.

    3. $\widehat{Z}_t = \mathcal{P}_\mathcal{C}(Q_t W_t)\widetilde{\Sigma}_t^{\frac{1}{2}}$.

    4. Calculate the preconditioned gradient $W_t$:

$$\widehat{W}_t = \left(I_n - \frac{1}{2}\widehat{Z}_t(\widehat{Z}_t^\mathsf{H}\widehat{Z}_t)^{-1}\widehat{Z}_t^\mathsf{H}\right) \cdot$$
$$\left[\frac{1}{\widehat{p}}\left(\mathcal{G}\mathcal{P}_\Omega\left(\mathcal{G}^*\left(\widehat{Z}_t\widehat{Z}_t^\top\right) - \boldsymbol{y}\right)\right)\overline{\widehat{Z}_t} + (\mathcal{I} - \mathcal{G}\mathcal{G}^*)\left(\widehat{Z}_t\widehat{Z}_t^T\right)\overline{\widehat{Z}_t}\right](\overline{\widehat{Z}_t^\mathsf{H}\widehat{Z}_t})^{-1}.$$

    5. $\widetilde{Z}_{t+1} = \widehat{Z}_t - \widehat{W}_t$.

4: **end for**

---

When $\mathcal{G}\boldsymbol{y} = \mathcal{H}\boldsymbol{x}$ is $\mu_0$-incoherent, $\mathcal{C}$ is a convex set defined as,

$$\mathcal{C} = \left\{\boldsymbol{Y} \in \mathbb{C}^{n \times r} \;\middle|\; \|\boldsymbol{Y}\|_{2,\infty} \leq \sqrt{\frac{2\mu_0 r}{n_s}}\right\}. \tag{42}$$

Then the projection $\mathcal{P}_\mathcal{C}(\cdot)$ of a given matrix $\boldsymbol{Y}$ onto the set $\mathcal{C}$ is calculated by row-wise trimming,

$$[\mathcal{P}_\mathcal{C}(\boldsymbol{Y})]^{(i,:)} = \begin{cases} \boldsymbol{Y}^{(i,:)} & \text{if } \left\|\boldsymbol{Y}^{(i,:)}\right\|_2 \leq \sqrt{\frac{2\mu_0 r}{n_s}}, \\ \frac{\boldsymbol{Y}^{(i,:)}}{\left\|\boldsymbol{Y}^{(i,:)}\right\|_2}\sqrt{\frac{2\mu_0 r}{n_s}} & \text{otherwise.} \end{cases}$$

For simplification of the notation, let $\widetilde{U}_t := Q_t W_t$. By applying the generator to Algorithm 3, we can obtain the corresponding algorithm in matrix space as following.

---

**Algorithm 4** Initialization via Resampled JHGD and Trimming in Matrix Space

---

1: **Preprocessing:**
     1. Make each dimension of the signal odd by zero-padding.
     2. Partition $\Omega$ into disjoint sets $\Omega_0, \cdots, \Omega_T$ of equal size $\widehat{m}$, let $\widehat{p} = \frac{\widehat{m}}{n_s}$.

2: **Set:** $\widetilde{\boldsymbol{X}}_0 = \mathcal{T}_r\left(\widehat{p}^{-1}\mathcal{G}\mathcal{P}_{\Omega_0}(\boldsymbol{y})\right) = \widetilde{\boldsymbol{U}}_0\widetilde{\boldsymbol{\Sigma}}_0\widetilde{\boldsymbol{U}}_0^\top$

3: **for** $t = 0, 1, \cdots, K$ **do**
     1. $\widehat{\boldsymbol{X}}_t = \mathcal{P}_{\mathcal{C}}(\widetilde{\boldsymbol{U}}_t)\widetilde{\boldsymbol{\Sigma}}_t\mathcal{P}_{\mathcal{C}}(\widetilde{\boldsymbol{U}}_t)^\top := \text{Trim}_{\mu_0}(\widetilde{\boldsymbol{X}}_t)$

     2. Update $\widehat{\boldsymbol{X}}_t$ in the negative direction of its Riemannian gradient:

$$\widehat{\boldsymbol{L}}_t = \widehat{\boldsymbol{X}}_t - \mathcal{P}_{\widehat{\mathbb{T}}_t}\left[\frac{1}{\widehat{p}}\mathcal{G}\mathcal{P}_{\Omega_{t+1}}\left(\mathcal{G}^*(\widehat{\boldsymbol{X}}_t) - \boldsymbol{y}\right) + (\mathcal{I} - \mathcal{G}\mathcal{G}^*)(\widehat{\boldsymbol{X}}_t)\right].$$

     3. Project updated $\boldsymbol{L}_t$ back to $\mathcal{M}_r$:

$$\widetilde{\boldsymbol{X}}_{t+1} = \mathcal{R}\left(\widehat{\boldsymbol{L}}_t\right).$$

4: **end for**

---

---

**Algorithm 5** $\text{Trim}_{\mu_0}$

---

  **Input:** $\widetilde{\boldsymbol{X}}_t = \widetilde{\boldsymbol{U}}_t\widetilde{\boldsymbol{\Sigma}}_t\widetilde{\boldsymbol{U}}_t^\top$
  **Output:** $\widehat{\boldsymbol{X}}_t = \widehat{\boldsymbol{A}}_t\widetilde{\boldsymbol{\Sigma}}_t\widehat{\boldsymbol{A}}_t^\top$, where

$$\widehat{\boldsymbol{A}}_t^{(i,:)} = \frac{\boldsymbol{U}_t^{(i,:)}}{\|\boldsymbol{U}_t^{(i,:)}\|}\min\left\{\|\boldsymbol{U}_t^{(i,:)}\|, \sqrt{\frac{2\mu_0 r}{n_s}}\right\}.$$

---

### B.3.2. INITIALIZATION VIA RESAMPLED JHGD AND PROOF OF THEOREM 3.2

The subsequent lemma characterizes the accuracy of the initial estimate produced by Algorithm 4.

**Lemma B.9.** Assume $\mathcal{H}\boldsymbol{x}$ is $\mu_0$-incoherent, then with probability at least $1 - (2K+1)n_s^{-2}$, the output of Algorithm 4 satisfies

$$\left\|\widetilde{\boldsymbol{X}}_K - \mathcal{H}\boldsymbol{x}\right\|_F \leq \left(\frac{12}{13}\right)^K \frac{\sigma_{\min}(\mathcal{H}\boldsymbol{x})}{256\kappa^2}$$

provided $\widehat{m} \geq C_5\mu_0\kappa^6 r^2\log(n_s)$ for some universal numerical constant $C_5 > 0$.

To facilitate the proof of this lemma, we first present several auxiliary results.

**Lemma B.10.** [(Wei et al., 2020), Lemma 4.10] Let $\widetilde{\boldsymbol{X}}_t = \mathcal{Q}(\widetilde{\boldsymbol{Z}}_t) = \widetilde{\boldsymbol{Z}}_t\widetilde{\boldsymbol{Z}}_t^\top = \widetilde{\boldsymbol{U}}_t\widetilde{\boldsymbol{\Sigma}}_t\widetilde{\boldsymbol{U}}_t^\top$ and $\mathcal{G}\boldsymbol{y} = \boldsymbol{U}\boldsymbol{\Sigma}\boldsymbol{U}^\top$ be two complex rank $r$ matrices, which satisfies the following property,

$$\left\|\widetilde{\boldsymbol{X}}_t - \mathcal{G}\boldsymbol{y}\right\|_F \leq \frac{\sigma_{\min}(\mathcal{G}\boldsymbol{y})}{10\sqrt{2}}.$$

Assume $\mathcal{G}\boldsymbol{y}$ is $\mu_0$-incoherent, then $\left\|\boldsymbol{U}^{(i,:)}\right\|^2 \leq \frac{2\mu_0 r}{n_s}$, then $\widehat{\boldsymbol{X}}_t = \text{Trim}_{\mu_0}(\widetilde{\boldsymbol{X}}_t) = \widehat{\boldsymbol{A}}_t\widetilde{\boldsymbol{\Sigma}}_t\widehat{\boldsymbol{A}}_t^\top = \widehat{\boldsymbol{U}}_t\widehat{\boldsymbol{\Sigma}}_t\widehat{\boldsymbol{U}}_t^\top$ in Algorithm 5 satisfies,

$$\left\|\widehat{\boldsymbol{X}}_t - \mathcal{G}\boldsymbol{y}\right\|_F \leq 8\kappa\left\|\widetilde{\boldsymbol{X}}_t - \mathcal{G}\boldsymbol{y}\right\|_F \quad\text{and}\quad \left\|\widehat{\boldsymbol{U}}^{(i,:)}\right\|^2 \leq \frac{200\mu_0 r}{81n_s},$$

where $\kappa$ refers to the condition number of $\mathcal{G}\boldsymbol{y}$.

**Lemma B.11.** [(Cai et al., 2019), Lemma 9] Assume there exists a numerical constant $\mu$ such that

$$\left\|\mathcal{P}_{\widehat{\boldsymbol{U}}_t}\boldsymbol{H}_a\right\|_F^2 \leq \frac{2\mu r}{n_s}$$

and

$$\|\mathcal{P}_{\boldsymbol{U}}\boldsymbol{H}_a\|_F^2 \leq \frac{2\mu r}{n_s},$$

for all $0 \leq a \leq n_s - 1$. Let $\Omega_{t+1} = \{a_k \mid k = 1, \cdots, \widehat{m}\}$ be a set of indices sampled independently and uniformly with replacement. If $\mathcal{P}_{\Omega_{l+1}}$ is independent of $\boldsymbol{U}$, and $\widehat{\boldsymbol{U}}_t$, then with probability at least $1 - n_s^{-2}$

$$\left\|\mathcal{P}_{\widehat{\mathbb{T}}_t}\mathcal{G}\left(\mathcal{I} - \widehat{p}^{-1}\mathcal{P}_{\Omega_{t+1}}\right)\mathcal{G}^*\left(\mathcal{P}_{\boldsymbol{U}} - \mathcal{P}_{\widehat{\boldsymbol{U}}_t}\right)\right\| \leq \sqrt{\frac{320\mu r \log(n_s)}{\widehat{m}}},$$

provided

$$\widehat{m} \geq \frac{125}{9}\mu r \log(n_s).$$

*Proof of Lemma B.9.* First assume that

$$\left\|\widetilde{\boldsymbol{X}}_t - \mathcal{G}\boldsymbol{y}\right\|_F \leq \frac{\sigma_{\min}(\mathcal{G}\boldsymbol{y})}{256\kappa^2}, \tag{43}$$

which is smaller than $\frac{\sigma_{\min}(\mathcal{G}\boldsymbol{y})}{10\sqrt{2}}$, then we apply the Lemma B.10 to obtain the following,

$$\left\|\widehat{\boldsymbol{X}}_t - \mathcal{G}\boldsymbol{y}\right\|_F \leq 8\kappa\left\|\widetilde{\boldsymbol{X}}_t - \mathcal{G}\boldsymbol{y}\right\|_F \quad \text{and} \quad \left\|\widehat{\boldsymbol{U}}^{(i,:)}\right\|^2 \leq \frac{200\mu_0 r}{81n_s}. \tag{44}$$

Direct calculus then gives,

$$\left\|\mathcal{P}_{\widehat{\boldsymbol{U}}_t}\boldsymbol{H}_a\right\|_F^2 \leq \frac{200\mu_0 r}{81n_s}. \tag{45}$$

Recall that

$$\widetilde{\boldsymbol{y}}_{t+1} = \mathcal{G}^*\left(\widetilde{\boldsymbol{X}}_{t+1}\right) = \mathcal{G}^*\left(\mathcal{R}\left(\widehat{\boldsymbol{L}}_t\right)\right), \tag{46}$$

where $\widehat{\boldsymbol{L}}_t := \widehat{\boldsymbol{X}}_t - \mathcal{P}_{\widehat{\mathbb{T}}_t}\left[\frac{1}{\widehat{p}}\mathcal{G}\mathcal{P}_{\Omega_{t+1}}\left(\mathcal{G}^*\left(\widehat{\boldsymbol{X}}_t\right) - \boldsymbol{y}\right) + (\mathcal{I} - \mathcal{G}\mathcal{G}^*)\left(\widehat{\boldsymbol{X}}_t\right)\right]$. Then follow a similar proof strategy as in Theorem B.2, we firstly prove the following,

$$\left\|\widetilde{\boldsymbol{X}}_{t+1} - \widehat{\boldsymbol{L}}_t\right\|_F \leq c'\left\|\widehat{\boldsymbol{L}}_t - \widehat{\boldsymbol{X}}_t\right\|_F^2, \tag{47}$$

$$\left\|\widehat{\boldsymbol{L}}_t - \mathcal{G}\boldsymbol{y}\right\|_F \leq c\left\|\widetilde{\boldsymbol{X}}_t - \mathcal{G}\boldsymbol{y}\right\|_F, \tag{48}$$

for some numerical constants $c < 1$ and $c'$, then prove

$$\left\|\widetilde{\boldsymbol{X}}_{t+1} - \mathcal{G}\boldsymbol{y}\right\|_F \leq \widetilde{\nu}\left\|\widetilde{\boldsymbol{X}}_t - \mathcal{G}\boldsymbol{y}\right\|_F \tag{49}$$

for some $\widetilde{\nu} < 1$.

To estimate the approximation error at the $(t+1)$-th iteration in (48), we do the following decomposition,

$$
\begin{aligned}
&\left\|\widehat{\boldsymbol{L}}_t - \mathcal{G}\boldsymbol{y}\right\|_F \\
&= \left\|\widehat{\boldsymbol{X}}_t - \mathcal{P}_{\widehat{\mathbb{T}}_t}\left[\frac{1}{\widehat{p}}\mathcal{G}\mathcal{P}_{\Omega_{t+1}}\left(\mathcal{G}^*\left(\widehat{\boldsymbol{X}}_t\right) - \boldsymbol{y}\right) + (\mathcal{I} - \mathcal{G}\mathcal{G}^*)\left(\widehat{\boldsymbol{X}}_t\right)\right] - \mathcal{G}\boldsymbol{y}\right\|_F \\
&= \left\|\widehat{\boldsymbol{X}}_t - \mathcal{G}\boldsymbol{y} - \mathcal{P}_{\widehat{\mathbb{T}}_t}\left[\widehat{p}^{-1}\mathcal{G}\mathcal{P}_{\Omega_{t+1}}\left(\mathcal{G}^*\left(\widehat{\boldsymbol{X}}_t\right) - \boldsymbol{y}\right)\right] - \mathcal{P}_{\widehat{\mathbb{T}}_t}\left[(\mathcal{I} - \mathcal{G}\mathcal{G}^*)\left(\widehat{\boldsymbol{X}}_t\right)\right]\right\|_F \\
&= \left\|\widehat{\boldsymbol{X}}_t - \mathcal{G}\boldsymbol{y} - \mathcal{P}_{\widehat{\mathbb{T}}_t}\left[\widehat{p}^{-1}\mathcal{G}\mathcal{P}_{\Omega_{t+1}}\left(\mathcal{G}^*\left(\widehat{\boldsymbol{X}}_t\right) - \boldsymbol{y}\right)\right] - \mathcal{P}_{\widehat{\mathbb{T}}_t}\left[(\mathcal{I} - \mathcal{G}\mathcal{G}^*)\left(\widehat{\boldsymbol{X}}_t - \mathcal{G}\boldsymbol{y}\right)\right]\right\|_F \\
&\leq \left\|\left(\mathcal{I} - \mathcal{P}_{\widehat{\mathbb{T}}_t}\right)\left(\widehat{\boldsymbol{X}}_t - \mathcal{G}\boldsymbol{y}\right)\right\|_F \\
&\quad + \left\|\left(\mathcal{P}_{\widehat{\mathbb{T}}_t}\mathcal{G}\mathcal{G}^*\mathcal{P}_{\widehat{\mathbb{T}}_t} - \widehat{p}^{-1}\mathcal{P}_{\widehat{\mathbb{T}}_t}\mathcal{G}\mathcal{P}_{\Omega_{t+1}}\mathcal{G}^*\mathcal{P}_{\widehat{\mathbb{T}}_t}\right)\left(\widehat{\boldsymbol{X}}_t - \mathcal{G}\boldsymbol{y}\right)\right\|_F \\
&\quad + \left\|\mathcal{P}_{\widehat{\mathbb{T}}_t}\mathcal{G}\left(\mathcal{I} - \widehat{p}^{-1}\mathcal{P}_{\Omega_{t+1}}\right)\mathcal{G}^*\left(\mathcal{I} - \mathcal{P}_{\widehat{\mathbb{T}}_t}\right)\left(\widehat{\boldsymbol{X}}_t - \mathcal{G}\boldsymbol{y}\right)\right\|_F \\
&:= I_1 + I_2 + I_3,
\end{aligned}
\tag{50}
$$

where the third equality utilizes the property $\mathcal{G}^*\mathcal{G} = \mathcal{I}$, thus $(\mathcal{I} - \mathcal{G}\mathcal{G}^*)(\mathcal{G}\boldsymbol{y}) = \boldsymbol{0}$.

For $I_1$,

$$I_1 = \left\| \left(\mathcal{I} - \mathcal{P}_{\widehat{\mathbb{T}}_t}\right)\left(\widehat{\boldsymbol{X}}_t - \mathcal{G}\boldsymbol{y}\right)\right\|_F \leq \frac{\left\|\widehat{\boldsymbol{X}}_t - \mathcal{G}\boldsymbol{y}\right\|_F^2}{\sigma_{\min}(\mathcal{G}\boldsymbol{y})} \leq \frac{1}{4}\left\|\widetilde{\boldsymbol{X}}_t - \mathcal{G}\boldsymbol{y}\right\|_F,$$

which applies Lemma B.10 and assumption (43).

Applying Lemma B.4, Lemma B.10 and equation (45), we obtain an upper bound for $I_2$,

$$\begin{aligned}
I_2 &= \left\|\left(\mathcal{P}_{\widehat{\mathbb{T}}_t}\mathcal{G}\mathcal{G}^*\mathcal{P}_{\widehat{\mathbb{T}}_t} - \widehat{p}^{-1}\mathcal{P}_{\widehat{\mathbb{T}}_t}\mathcal{G}\mathcal{P}_{\Omega_{t+1}}\mathcal{G}^*\mathcal{P}_{\widehat{\mathbb{T}}_t}\right)\left(\widehat{\boldsymbol{X}}_t - \mathcal{G}\boldsymbol{y}\right)\right\|_F \\
&\leq \sqrt{\frac{6400\mu_0 r\log(n_s)}{81\widehat{m}}}\left\|\widehat{\boldsymbol{X}}_t - \mathcal{G}\boldsymbol{y}\right\|_F \\
&\leq 8\kappa\sqrt{\frac{6400\mu_0 r\log(n_s)}{81\widehat{m}}}\left\|\widetilde{\boldsymbol{X}}_t - \mathcal{G}\boldsymbol{y}\right\|_F
\end{aligned}$$

with probability at least $1 - n_s^{-2}$. To bound $I_3$, since

$$\begin{aligned}
&\left(\mathcal{I} - \mathcal{P}_{\widehat{\mathbb{T}}_t}\right)\left(\widehat{\boldsymbol{X}}_t - \mathcal{G}\boldsymbol{y}\right) \\
&= \left(\mathcal{I} - \mathcal{P}_{\widehat{\mathbb{T}}_t}\right)(-\mathcal{G}\boldsymbol{y}) \\
&= -\boldsymbol{U}\boldsymbol{U}^{\mathsf{H}}\mathcal{G}\boldsymbol{y} + \widehat{\boldsymbol{U}}_t\widehat{\boldsymbol{U}}_t^{\mathsf{H}}\mathcal{G}\boldsymbol{y} + \boldsymbol{U}\boldsymbol{U}^{\mathsf{H}}\mathcal{G}\boldsymbol{y}\bar{\widehat{\boldsymbol{U}}}_t\bar{\widehat{\boldsymbol{U}}}_t^{\mathsf{H}} - \widehat{\boldsymbol{U}}_t\widehat{\boldsymbol{U}}_t^{\mathsf{H}}\mathcal{G}\boldsymbol{y}\bar{\widehat{\boldsymbol{U}}}_t\bar{\widehat{\boldsymbol{U}}}_t^{\mathsf{H}} \\
&= -\left(\boldsymbol{U}\boldsymbol{U}^{\mathsf{H}} - \widehat{\boldsymbol{U}}_t\widehat{\boldsymbol{U}}_t^{\mathsf{H}}\right)\mathcal{G}\boldsymbol{y}\left(\boldsymbol{I} - \bar{\widehat{\boldsymbol{U}}}_t\bar{\widehat{\boldsymbol{U}}}_t^{\mathsf{H}}\right) \\
&= \left(\boldsymbol{U}\boldsymbol{U}^{\mathsf{H}} - \widehat{\boldsymbol{U}}_t\widehat{\boldsymbol{U}}_t^{\mathsf{H}}\right)\left(\widehat{\boldsymbol{X}}_t - \mathcal{G}\boldsymbol{y}\right)\left(\boldsymbol{I} - \bar{\widehat{\boldsymbol{U}}}_t\bar{\widehat{\boldsymbol{U}}}_t^{\mathsf{H}}\right) \\
&= \left(\mathcal{P}_{\boldsymbol{U}} - \mathcal{P}_{\widehat{\boldsymbol{U}}_t}\right)\left(\mathcal{I} - \mathcal{P}_{\bar{\widehat{\boldsymbol{U}}}_t}\right)\left(\widehat{\boldsymbol{X}}_t - \mathcal{G}\boldsymbol{y}\right),
\end{aligned}$$

thus we have,

$$\begin{aligned}
I_3 &= \left\|\mathcal{P}_{\widehat{\mathbb{T}}_t}\mathcal{G}\left(\mathcal{I} - \widehat{p}^{-1}\mathcal{P}_{\Omega_{t+1}}\right)\mathcal{G}^*\left(\mathcal{I} - \mathcal{P}_{\widehat{\mathbb{T}}_t}\right)\left(\widehat{\boldsymbol{X}}_t - \mathcal{G}\boldsymbol{y}\right)\right\|_F \\
&= \left\|\mathcal{P}_{\widehat{\mathbb{T}}_t}\mathcal{G}\left(\mathcal{I} - \widehat{p}^{-1}\mathcal{P}_{\Omega_{t+1}}\right)\mathcal{G}^*\left(\mathcal{P}_{\boldsymbol{U}} - \mathcal{P}_{\widehat{\boldsymbol{U}}_t}\right)\left(\mathcal{I} - \mathcal{P}_{\bar{\widehat{\boldsymbol{U}}}_t}\right)\left(\widehat{\boldsymbol{X}}_t - \mathcal{G}\boldsymbol{y}\right)\right\|_F \\
&\leq \left\|\mathcal{P}_{\widehat{\mathbb{T}}_t}\mathcal{G}\left(\mathcal{I} - \widehat{p}^{-1}\mathcal{P}_{\Omega_{t+1}}\right)\mathcal{G}^*\left(\mathcal{P}_{\boldsymbol{U}} - \mathcal{P}_{\widehat{\boldsymbol{U}}_t}\right)\right\|\left\|\widehat{\boldsymbol{X}}_t - \mathcal{G}\boldsymbol{y}\right\|_F \\
&\leq \sqrt{\frac{32000\mu_0 r\log(n_s)}{81\widehat{m}}}\left\|\widehat{\boldsymbol{X}}_t - \mathcal{G}\boldsymbol{y}\right\|_F \\
&\leq 8\kappa\sqrt{\frac{32000\mu_0 r\log(n_s)}{81\widehat{m}}}\left\|\widetilde{\boldsymbol{X}}_t - \mathcal{G}\boldsymbol{y}\right\|_F,
\end{aligned}$$

with probability at least $1 - n_s^{-2}$. The second inequality follows Lemma B.11, and the last inequality follows (44) and (45).

Then combine results from $I_1$ to $I_3$, we obtain,

$$\left\|\widehat{\boldsymbol{L}}_t - \mathcal{G}\boldsymbol{y}\right\|_F \leq \left(\frac{1}{4} + 231\kappa\sqrt{\frac{\mu_0 r\log(n_s)}{\widehat{m}}}\right)\left\|\widetilde{\boldsymbol{X}}_t - \mathcal{G}\boldsymbol{y}\right\|_F \leq \frac{1}{2}\left\|\widetilde{\boldsymbol{X}}_t - \mathcal{G}\boldsymbol{y}\right\|_F$$

with probability at least $1 - 2n_s^{-2}$, provided $\widehat{m} \geq C_5\mu_0\kappa^2 r\log(n_s)$ for some sufficiently large universal constant $C_5$. So we've proved (48).

To prove (47), we apply Corollary B.1 with $q = 0.1$, if $\left\|\mathcal{G}\boldsymbol{y} - \widehat{\boldsymbol{X}}_t\right\|_2 \leq 0.1\sigma_{\min}(\mathcal{G}\boldsymbol{y})$ we have the following,

$$\left\|\widetilde{\boldsymbol{X}}_{t+1} - \widehat{\boldsymbol{L}}_t\right\|_F \leq \frac{5}{6\sigma_{\min}(\mathcal{G}\boldsymbol{y})}\left\|\widehat{\boldsymbol{L}}_t - \widehat{\boldsymbol{X}}_t\right\|_F^2. \tag{51}$$

Notice the assumption $\left\|\widehat{\boldsymbol{X}}_t - \mathcal{G}\boldsymbol{y}\right\|_2 \leq 0.1\sigma_r(\mathcal{G}\boldsymbol{y})$ can be satisfied if $\widehat{m} \geq C_6\kappa^4r^2$, since

$$\left\|\widehat{\boldsymbol{X}}_t - \mathcal{G}\boldsymbol{y}\right\|_2 \leq \left\|\widehat{\boldsymbol{X}}_t - \mathcal{G}\boldsymbol{y}\right\|_F \leq 8\kappa\sqrt{2r}\left\|\widetilde{\boldsymbol{X}}_t - \mathcal{G}\boldsymbol{y}\right\|_2 \leq 0.1\sigma_{\min}(\mathcal{G}\boldsymbol{y})$$

following Lemma B.7.

We finally need to prove the following,

$$
\begin{aligned}
&\left\|\widetilde{\boldsymbol{X}}_{t+1} - \mathcal{G}\boldsymbol{y}\right\|_F \\
\leq{}& \left\|\widetilde{\boldsymbol{X}}_{t+1} - \widehat{\boldsymbol{L}}_t\right\|_F + \left\|\widehat{\boldsymbol{L}}_t - \mathcal{G}\boldsymbol{y}\right\|_F \\
\leq{}& c'\left\|\widehat{\boldsymbol{L}}_t - \widehat{\boldsymbol{X}}_t\right\|_F^2 + c\left\|\widetilde{\boldsymbol{X}}_t - \mathcal{G}\boldsymbol{y}\right\|_F \\
\leq{}& c'\left(2\left\|\widehat{\boldsymbol{L}}_t - \mathcal{G}\boldsymbol{y}\right\|_F^2 + 2\left\|\widehat{\boldsymbol{X}}_t - \mathcal{G}\boldsymbol{y}\right\|_F^2\right) + c\left\|\widetilde{\boldsymbol{X}}_t - \mathcal{G}\boldsymbol{y}\right\|_F \\
\leq{}& c'\left(2c^2\left\|\widetilde{\boldsymbol{X}}_t - \mathcal{G}\boldsymbol{y}\right\|_F^2 + 2(8\kappa)^2\left\|\widetilde{\boldsymbol{X}}_t - \mathcal{G}\boldsymbol{y}\right\|_F^2\right) + c\left\|\widetilde{\boldsymbol{X}}_t - \mathcal{G}\boldsymbol{y}\right\|_F \\
={}& \left[2c'(c^2 + 64\kappa^2)\left\|\widetilde{\boldsymbol{X}}_t - \mathcal{G}\boldsymbol{y}\right\|_F + c\right]\left\|\widetilde{\boldsymbol{X}}_t - \mathcal{G}\boldsymbol{y}\right\|_F \\
\leq{}& \left(2 \cdot \frac{5}{6\sigma_{\min}(\mathcal{G}\boldsymbol{y})} \cdot (0.5^2 + 64\kappa^2) \cdot \left\|\widetilde{\boldsymbol{X}}_t - \mathcal{G}\boldsymbol{y}\right\|_F + 0.5\right)\left\|\widetilde{\boldsymbol{X}}_t - \mathcal{G}\boldsymbol{y}\right\|_F \\
\leq{}& \left(\frac{10}{6} \cdot (0.5^2 + 64\kappa^2) \cdot \frac{1}{256\kappa^2} + 0.5\right)\left\|\widetilde{\boldsymbol{X}}_t - \mathcal{G}\boldsymbol{y}\right\|_F \\
\leq{}& \left(\frac{5}{3} \cdot \left(\frac{0.25}{256} + \frac{1}{4}\right) + 0.5\right)\left\|\widetilde{\boldsymbol{X}}_t - \mathcal{G}\boldsymbol{y}\right\|_F \\
\leq{}& \frac{12}{13}\left\|\widetilde{\boldsymbol{X}}_t - \mathcal{G}\boldsymbol{y}\right\|_F.
\end{aligned}
$$

Clearly, (43) is also true for the $(t+1)$-th iteration.

Since $\widetilde{\boldsymbol{X}}_0 = \mathcal{T}_r\left(\widehat{p}^{-1}\mathcal{G}\mathcal{P}_{\Omega_0}(\boldsymbol{y})\right)$, (43) holds for $t=0$ with probability at least $1 - n_s^{-2}$ provided,

$$\widehat{m} \geq C\mu_0\kappa^6r^2\log(n_s)$$

for some numerical constant $C > 0$. Then take an upper bound on the numeber of measurements, we have,

$$\left\|\widetilde{\boldsymbol{X}}_K - \mathcal{H}\boldsymbol{x}\right\|_F \leq \left(\frac{12}{13}\right)^K \frac{\sigma_{\min}(\mathcal{H}\boldsymbol{x})}{256\kappa^2}$$

with probability at least $1 - (2K+1)n_s^{-2}$ provided $\widehat{m} \geq C\mu_0\kappa^6r^2\log(n_s)$, which completes the proof. $\qquad\square$

**Proof of Theorem 3.2:**

*Proof.* If we take $K = \left\lceil 13\log\left(\frac{\sqrt{n_s\log(n_s)}}{91\epsilon_0}\right)\right\rceil$, then the third condition (37) in Theorem B.2 holds with probability at least $1 - (2K+1)n_s^{-2}$. Combining Lemma B.3 and Lemma B.4, the theorem can be proved. $\qquad\square$

## C. Empirical Phase Transition

We evaluate the empirical phase-transition behavior of five recovery algorithms: JHGD, PGD (Cai et al., 2018), SHGD (Li et al., 2024), FIHT (Cai et al., 2019), and SGD (Tong et al., 2021). All experiments use signal length $n_s = 127$. The sampling ratio is $p = m/n_s$ and takes 18 equally spaced values in $[0.15, 0.95]$. For each pair $(n_s, m)$, we vary the sparsity level from $r = 1$ to $r = 60$ in increments of one. All methods terminate when $\|\boldsymbol{x}_{t+1} - \boldsymbol{x}_t\|_2/\|\boldsymbol{x}_t\|_2 \leq 10^{-7}$. A trial is declared successful if the relative reconstruction error satisfies

$$\|\boldsymbol{x}_{\text{rec}} - \boldsymbol{x}\|_2/\|\boldsymbol{x}\|_2 \leq 10^{-4},$$

where $\boldsymbol{x}$ and $\boldsymbol{x}_{\text{rec}}$ denote the ground-truth and reconstructed signals, respectively. For PGD, SHGD, SGD, and JHGD, step sizes are selected via backtracking line search. For each triple $(n_s, m, r)$, we run 50 independent Monte Carlo trials. We consider two undamped signal models: (i) *nonseparated* frequencies $\{f_k\}_{k=1}^r$, and (ii) *separated* frequencies with minimum separation at least $\frac{1.5}{n_s}$.

### C.1. Nonseparated Frequencies

Figure 4 reports the success rates of the five algorithms (PGD, SHGD, FIHT, SGD, and JHGD) for undamped signals with nonseparated frequencies. White indicates perfect recovery on all 50 randomly generated test signals, whereas dark black indicates that the algorithm fails on all 50 trials. We observe that when both the sampling ratio and sparsity are large ($p > 0.5$ and $r > 30$), FIHT, SGD, and JHGD achieve higher success rates than PGD and SHGD. When $p \leq 0.5$, PGD, SHGD, SGD, and JHGD attain higher success rates than FIHT. Moreover, JHGD exhibits a larger success region than SGD, indicating improved reconstruction performance. For example, at $p = 0.65$ and $r = 30$, the success rate of SGD is below 20%, whereas JHGD exceeds 50%; FIHT achieves roughly 70% success, while both SHGD and PGD have a success rate of 0%.

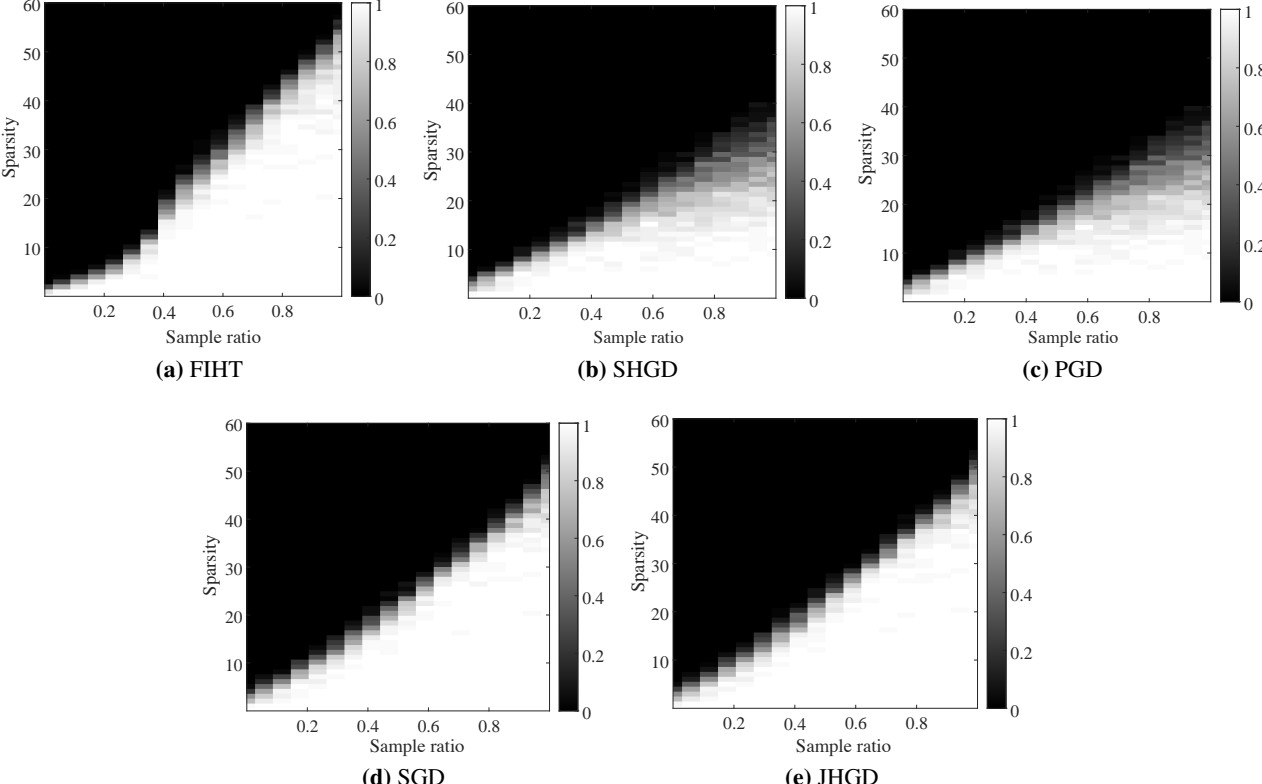

**Figure 4.** Results of success rates for low-rank Hankel matrix completion on simulated data with varying sparsity $r$ and varying sample ratio $p = m/n_s$ when the signal has no damping factor and the frequencies are not separated.

## C.2. Separated Frequencies

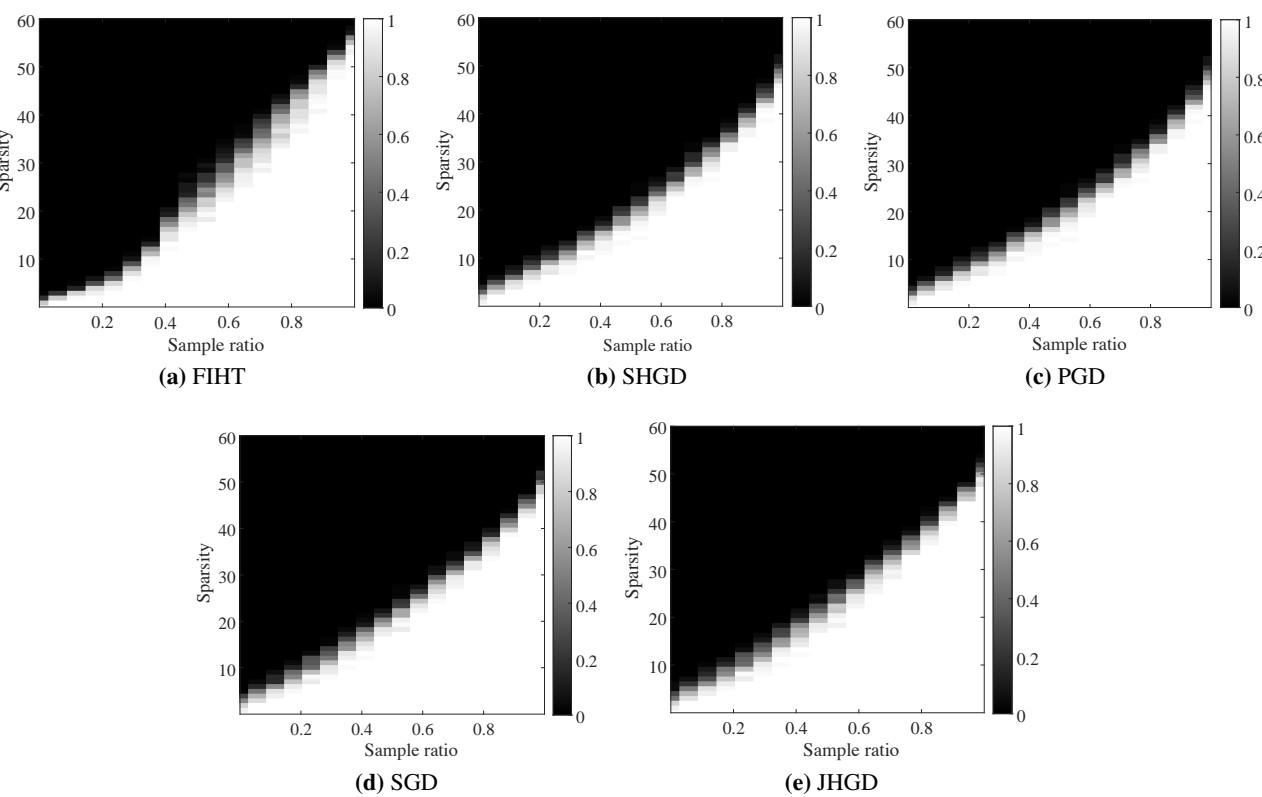

**Figure 5.** Results of success rates for low-rank Hankel matrix completion on simulated data with varying sparsity $r$ and varying sample ratio $p = m/n_s$ when the signal has no damping factor and the frequencies are generated with a minimum separation of $1.5/n_s$.

Figure 5 shows the results of the recovery ability of the five methods when the signals are undamped and the frequencies are generated with a minimum separation of $\frac{1.5}{n_s}$. We also conduct 50 random Monte Carlo trails through distinct $(m, r)$. All algorithms show significantly higher success rates compared to those presented in Figure 4. Taking SHGD and PGD as an example, the performance sensitivity of SHGD and PGD to frequency separation becomes evident when examining reconstruction success rates: while achieving $100\%$ recovery for separated frequencies at $p = 0.9$ and $r = 40$, the algorithm fails completely for non-separated cases under the same setting (see Figure 4). Compared to SHGD and PGD, the proposed JHGD algorithm demonstrates robust recovery performance, maintaining consistently high success rates regardless of frequency separation conditions. When the sampling ratio is 0.65 and the sparsity is 32 in Figure 5, the success rate of JHGD is greater than $70\%$ and the success rate of FIHT is roughly $85\%$, while SGD has a success rate less than $10\%$, both PGD and SHGD fail completely.

To draw a conclusion, regardless of whether the frequencies are separated, the proposed JHGD demonstrates robust performance across all sampling ratios. When the sampling ratio and sparsity are low, FIHT exhibits a lower success rate compared to the other four methods, while JHGD outperforms all alternatives. However, for higher sampling ratios $(p > 0.5)$ and sparsity levels $(r > 25)$, FIHT achieves the highest success rate, followed closely by JHGD, with SGD ranking third. Both FIHT and JHGD consistently surpass PGD and SHGD in performance under these conditions. These findings suggest that while FIHT excels in high-sampling/high-sparsity conditions, JHGD maintains consistently strong performance across all settings of $(m, r)$.

