# OpenReview forum: "Fast Spectrally Sparse Signal Reconstruction via Jacobi-Preconditioned Gradient Descent"
_ICML.cc/2026/Conference — ICML 2026 spotlight_

### Official Review · Reviewer_xaEt · 2026-02-22

**Soundness:** 3
**Presentation:** 3
**Significance:** 3
**Originality:** 2
**Overall Recommendation:** 4
**Confidence:** 3

**Summary:**

This paper studies spectrally sparse signal reconstruction via low-rank Hankel matrix completion. The authors propose a Jacobi-preconditioned gradient descent method derived from the Jacobian of the generator, which maps factor-space iterates to complex-symmetric matrix space. The central idea is to construct a preconditioner based on the pseudoinverse of the Jacobi operator, leading to an orthogonal projection onto the tangent space of the rank-r manifold. The paper further establishes an equivalence between factor-space preconditioned gradient descent and Riemannian gradient descent in matrix space. An efficient approximation of the exact Jacobi preconditioner is derived, yielding a practical algorithm with per-iteration complexity comparable to existing first-order methods. Theoretical results provide linear convergence guarantees under incoherence and sampling assumptions, and consistent runtime improvements over several baselines.

**Compliance With Llm Reviewing Policy:**

Affirmed.

**Final Justification:**

Thank the authors for the detailed rebuttal. Overall, I find this to be a solid and well-executed paper. Although the work is developed within an established framework, it nevertheless provides meaningful insights into this class of problems.

**Key Questions For Authors:**

1. Are the assumptions in Theorem 2.3 valid in practice? How sensitive is JHGD to violations of the assumptions in Theorem 2.3? For example, if singular values are highly skewed during early iterations, does the approximate preconditioner still behave like the exact orthogonal projector?

2. Resampled JHGD achieves a theoretically superior sample complexity. However, its practical relevance is questionable. Firstly, the algorithm requires a fresh set of samples at each iteration, which is often infeasible in real-world applications where the measurement set is fixed. Second, the overhead introduced by resampling and trimming may offset the computational gains offered by the algorithm. Eventually, the experimental section does not include separate tests evaluating the performance of resampled JHGD, leaving its practical effectiveness unclear.

3. The paper claims that the preconditioning framework extends to other structured and general low-rank matrix recovery problems. Can the authors provide a concrete example demonstrating this extension?

**Limitations:**

At the current stage, limitations are mainly around the validation of assumptions of Theorem 2.3 and Theorem 3.2, and application to real-world problems.

**Strengths And Weaknesses:**

Strengths:

1. Conceptual unification of factor and manifold viewpoints: A major strength of the paper is the introduction of the generator-based framework that explicitly connects factor-space and matrix-space optimization. By deriving the Jacobi operator, its adjoint, and pseudoinverse, the authors provide a clean explanation of how preconditioned gradient descent in factor space corresponds to Riemannian gradient descent in matrix space.

2. Principled preconditioner construction: The proposed preconditioner is not heuristic but derived from the structure of the Jacobian of the generator. The SVD-based characterization of the Jacobi operator and the resulting pseudoinverse expansion are technically nontrivial. Even though the exact preconditioner is expensive, the derivation also provides a principled basis for approximation.

3. Empirical evaluation is solid: The experimental section is well-structured and compares against multiple strong baselines. Results show consistent improvements in iteration count and runtime, particularly in large-scale settings. The gains appear stable across sparsity levels, sample sizes, and signal dimensions.


Weaknesses:

1. Incremental rather than paradigm-shifting contribution: While the paper is technically sound and presents a clean geometric reformulation of preconditioned low-rank optimization, the contribution remains largely within the established Hankel lifting framework for spectral compressed sensing. The proposed JHGD can be viewed as a principled geometric refinement along the existing line of Hankel-based nonconvex spectral compressed sensing methods. While the Jacobi generator framework provides useful conceptual clarity and improved conditioning, the overall recovery paradigm, assumptions, and sample complexity scaling remain similar to prior work. As such, the contribution is better characterized as a technically elegant acceleration within an established framework rather than a paradigm shift.

2. Approximate preconditioner relies on heuristic assumptions: The approximation in Theorem 2.3 leads to the practical preconditioner P_A  by assuming singular values of Z are of comparable magnitude and certain uniformity properties of M. These assumptions are not rigorously justified. Some important issues remain unclear, such as, whether they hold throughout iterations, how sensitive the method is to violations, whether the approximation degrades in highly ill-conditioned regimes. An experiment verifing empirically whether these assumptions hold in typical signal recovery scenarios and a deeper robustness analysis would strengthen the paper.

3. Condition number dependence in sample complexity: Although the iteration contraction rate is claimed to be independent of the condition number, the sampling complexity in Theorem 3.2 scales as \kappa^6, which is quite strong. This appears to shift condition number dependence from optimization to sample requirements. A clearer discussion reconciling these two aspects would be helpful.

4. Practical scope is somewhat narrow: Experiments focus on synthetic spectrally sparse signals. It would strengthen the paper to include experiments on real-world applications or datasets.

---

> ### Author Rebuttal · Authors · 2026-03-31
>
> >D1. Validity of the assumptions in Theorem 2.3 and robustness of the approximate preconditioner to skewed singular values
>
> **Reply:** We thank the reviewers for these related comments. The assumption in
> Theorem 2.3 that the nonzero singular values of $\mathbf{Z}$ have
> comparable magnitude is a sufficient condition under which the simplified
> preconditioner most closely approximates the exact orthogonal projector,
> rather than a strict requirement for applying JHGD.
>
> When this assumption is violated, the approximation quality degrades
> gracefully. Specifically, a direct calculation shows that the expected
> squared approximation error is bounded by $\frac{1}{2 \sigma_{\min} ^4(\mathbf{Z})}
>   \mathbb{E}_{\mathbf{M}} \bigl[ ||\mathbf{M} ||_2^2\bigr]$,
> which we will include in the revised manuscript. This bound confirms that
> the approximation remains reliable as long as
> $\sigma _{\min}(\mathbf{Z})$ is bounded away from zero.
>
> Empirically, in both general and structured low-rank matrix recovery
> problems, the factor matrix is typically full rank throughout the iterations, so
> $\sigma _{\min}(\mathbf{Z})$ stays well above zero and the simplified
> preconditioner serves as a robust surrogate for the exact projector. If the
> factor matrix becomes numerically rank deficient, this typically indicates
> that the prescribed rank $r$ has been overestimated; reducing $r$ restores
> both the conditioning and the approximation quality.
>
> We will clarify this point and include the above error bound in
> the revised version.
>
> >D2. Resampled JHGD achieves a theoretically superior sample complexity. However, its practical relevance is questionable.
>
> **Reply:** We thank the reviewer for this important comment. We agree that the
> resampled-and-trimmed version of JHGD is not the practically preferred
> implementation; its main role is theoretical, introduced to establish a
> sharper sample complexity guarantee.
>
> Resampling and trimming are standard proof techniques in nonconvex low-rank
> recovery, also used in FIHT [R1] and SHGD [R2]. Resampling avoids
> statistical coupling between the current iterate and the measurements used
> in the next update, yielding sharper concentration bounds. Trimming
> enforces incoherence by projecting each iterate onto the set of
> $\mu_0$-incoherent matrices. Both are introduced primarily for the
> theoretical analysis.
>
> In practice, the measurement set is fixed, so obtaining fresh samples at
> each iteration is unrealistic, and the overhead of resampling and trimming
> may offset the computational advantages of the algorithm. For this reason,
> as in FIHT [R1] and SHGD [R2], we do not provide separate experiments for
> the resampled variant; our experiments focus on the practically relevant
> fixed-data version.
>
> We will revise the manuscript to clarify that the resampled-and-trimmed
> scheme is primarily a theoretical tool for improving sample complexity,
> whereas the fixed-data version of JHGD is the one of practical interest
> and the one evaluated in our experiments.
>
> References:
>
> [R1] Fast and provable algorithms for spectrally sparse signal reconstruction via low-rank Hankel matrix completion. Applied and Computational Harmonic Analysis, 2019.
>
> [R2] Projected gradient descent for spectral compressed sensing via symmetric Hankel factorization. IEEE Transactionson Signal Processing, 2024.
>
> >D3. The paper claims that the preconditioning framework extends to other structured and general low-rank matrix recovery problems. Can the authors provide a concrete example demonstrating this extension?
>
> **Reply:**  We thank the reviewer for this suggestion. As a concrete example, our preconditioning framework extends naturally to low-rank symmetric matrix sensing, a standard problem in general low-rank matrix recovery. Specifically, given measurements
>
> $$\mathbf{y} = \mathcal{A}(\mathbf{X}_\star) \in \mathbb{R}^m,$$
>
> where $\mathbf{X}_\star \in \mathbb{R}^{n \times n}$ is a rank-$r$ positive
> semidefinite matrix and
> $\mathcal{A}(\mathbf{X}) = \{\langle \mathbf{A}_k, \mathbf{X} \rangle\} _{k=1}^m$,
> we factorize $\mathbf{X} = \mathbf{Z}\mathbf{Z}^\top$ and consider
>
> $$\min_{\mathbf{Z} \in \mathbb{R}^{n \times r}} \quad g(\mathbf{Z}) = \frac{1}{2} || \mathcal{A}(\mathbf{Z}\mathbf{Z}^\top) - \mathbf{y} ||_2^2.$$
>
> A corresponding preconditioned gradient update is
>
> $$\mathbf{Z}_{t+1} = \mathbf{Z}_t - \alpha \left( \frac{1}{2}\mathbf{I}_n - \frac{1}{4}\mathbf{Z}_t(\mathbf{Z}_t^\top \mathbf{Z}_t)^{-1}\mathbf{Z}_t^\top \right) \nabla g(\mathbf{Z}_t) (\mathbf{Z}_t^\top \mathbf{Z}_t)^{-1},$$
>
> where the preconditioning scheme is the same as that in our method.
>
> This example illustrates how our preconditioning framework can be extended
> from structured to general low-rank matrix recovery. However, establishing
> convergence guarantees in this broader setting remains nontrivial, since the
> general measurement model changes both the problem structure and the
> associated optimization geometry. We will include this example in the revised manuscript.

---

> > ### Author Rebuttal · Reviewer_xaEt · 2026-04-02
> >
> > Thank the authors for the response. My concern has been largely resolved. However, I will keep my rating unchanged w.r.t. the novelty, as I still view the contribution as incremental rather than paradigm-shifting.

---

> > > ### Author Response · Authors · 2026-04-02
> > >
> > > Thank you for your time. We agree that our method is developed within the
> > > established framework of low-rank Hankel matrix completion, and that its
> > > recovery paradigm, assumptions, and sample complexity scaling are similar to
> > > those of prior work. However, we respectfully emphasize that our contribution
> > > lies in a novel and principled approach to designing computationally efficient
> > > preconditioners with rigorous theoretical guarantees.
> > >
> > > **Existing methods are mainly either factorization-based or manifold-based, each
> > > with clear trade-offs.** Factorization-based methods are computationally
> > > efficient, but their analysis is complicated by the complex-symmetric
> > > factorization ambiguity $\mathbf{X} = \mathbf{Z}\mathbf{Z}^\top = (\mathbf{Z}\mathbf{Q})(\mathbf{Z}\mathbf{Q})^\top$,
> > > which makes it difficult to define meaningful distances in factor space.
> > > Manifold-based methods avoid this issue by working directly in matrix space,
> > > which simplifies the analysis, but incurs higher per-iteration cost and memory
> > > usage.
> > >
> > > **Our main contribution is to unify these two perspectives** through a generator
> > > that maps factor-space iterates to their matrix-space counterparts and
> > > establishes their equivalence under suitable retractions. **This lets the
> > > algorithm operate in factor space for efficiency while enabling convergence
> > > analysis in matrix space, thereby avoiding the factorization ambiguity.**
> > > Moreover, this equivalence leads to a principled preconditioning strategy:
> > > viewed in matrix space, our preconditioner corresponds to an orthogonal
> > > projection of the gradient onto the tangent space, strengthening ScaledGD,
> > > which lacks this projection property. To the best of our knowledge, **both this
> > > factorization-manifold unification and the resulting preconditioning strategy
> > > are new, and they may provide a useful framework for designing efficient
> > > preconditioners for structured and general low-rank matrix recovery.**

---

### Official Review · Reviewer_8pAH · 2026-02-26

**Soundness:** 3
**Presentation:** 4
**Significance:** 3
**Originality:** 4
**Overall Recommendation:** 5
**Confidence:** 4

**Summary:**

This paper addresses the challenge of spectrally sparse signal reconstruction from partial observations by formulating it as a low-rank Hankel matrix completion problem.
To solve it efficiently, the authors propose a novel Jacobi-preconditioned gradient descent (JHGD) method that elegantly unifies factor-space and matrix-space Riemannian manifold optimization, which achieves low-cost, condition-number-independent linear convergence.

**Compliance With Llm Reviewing Policy:**

Affirmed.

**Final Justification:**

This paper is overall a high-quality paper and addresses an important problem in the fields of signal processing and optimization, which benefits the machine learning community. I recommend "accept" considering the complete theory and good-to-excellent reproducibility. I am glad to see more real-world applications regarding this method.

**Key Questions For Authors:**

1. What is the meaning of $\mathcal P$, projection or preconditioner?
2. Can the authors provide more details about implementation?
3. How to find the optimal $r$ empirically or theoretically?

**Limitations:**

The authors have pointed out that the theoretical analysis in the paper is based on the undamped system (assuming $\tau_k=0$), which is not always the case. Moreover, the experiments did not include real-world datasets, while the results in the paper are persuasive to some extent.

**Strengths And Weaknesses:**

Strengths:
- S1. The optimization techniques in the paper (e.g., SVD of Jacobi operator, preconditioning) are sound. Although I have not checked every detail in the appendix, the methodology seems to be correct and elegant.
- S2. The paper is easy to follow. The motivation and the key challenges are stated clearly. Then, the algorithm with a theoretical guarantee is presented.
- S3. The paper focuses on the application of spectrally sparse signal reconstruction, and it further explores the new insights from the optimization perspective. Theorem 2.4 is impressive, which suggests that the proposed approximation does not sacrifice accuracy.
- S4. The paper rediscovered the connection between factor- and matrix-space algorithms, from which it motivates a more efficient solver.

Weakness:
- W1. Some notations are confusing. For example, $\mathcal P$ is an orthogonal projection by definition 2.2, while in (10), it seems to refer to a preconditioner.
- W2. The paper includes many mathmatical deviation. However, it may not be intuitive enough. A figure to illustrate the idea or the algorithms is recommended.
- W3. The rank $r$ is a tunable hyperparameter. But the experiments did not include the related analysis.
- W4. minor point: The code to reproduce the experiments is unavailable.

---

> ### Author Rebuttal · Authors · 2026-03-31
>
> >C1. What is the meaning of $\mathcal{P}$, projection or preconditioner?
>
> **Reply:** We thank the reviewer for raising this point. In our paper, both projections
> and preconditioners are linear operators, but they play different roles and act
> in different spaces. Specifically, $\mathcal{P}$ in Definition~2.2 is a generic
> notation for an \textbf{orthogonal projection} in matrix space. By contrast,
> $\mathcal{P}_E$ in Eq. (10) and $\mathcal{P}_A$ in Eq. (12) denote
> \textbf{preconditioners} acting in the factor space; they are not orthogonal
> projections themselves. The two notions are connected through the Jacobi
> operator $\mathcal{J}$ and its adjoint $\mathcal{J}^*$: although
> $\mathcal{P}_E$ and $\mathcal{P}_A$ are not orthogonal projections, the
> composed operators
>
> $$\mathcal{J}\mathcal{P}_E \mathcal{J}^* = \mathcal{J}\mathcal{P}_A\mathcal{J}^* = \mathcal{J}\mathcal{J}^\dagger$$
>
> are orthogonal projections onto the tangent space $\mathbb{T}_t$. We will
> clarify this distinction in the revised manuscript.
>
> >C2. Can the authors provide more details about implementation?
>
> **Reply:** We thank the reviewer for this suggestion. We will add more implementation
> details in the revised manuscript. All experiments were implemented in
> MATLAB R2021b and run on a computer with an Intel Xeon(R) CPU at 2.60\,GHz
> (32 cores) under an Ubuntu system.
>
> For the spectral initialization, we compute a rank-$r$ truncated spectral
> decomposition of $p^{-1}\mathcal{H}\mathcal{P}_\Omega (\mathbf{y})$. In practice,
> this step is carried out using a Lanczos-type truncated SVD together with
> FFT-based Hankel matrix--vector multiplications, rather than explicitly
> forming the Hankel matrix and applying a dense SVD. This yields an
> initialization cost of $\mathcal{O}(nr\log n + nr^2)$.
>
> >C3. How to find the optimal $r$ empirically or theoretically?
>
> **Reply:** We thank the reviewer for this important question. In our paper, the rank $r$
> is assumed to be known. Theoretically, when the signal is a superposition of
> $r$ complex exponentials, the associated Hankel matrix has rank exactly $r$,
> so $r$ coincides with the number of spectral components.
>
> When $r$ is unknown, choosing the optimal value is in general a nontrivial
> model-order selection problem, especially in the presence of noise. In
> practice, however, a suitable $r$ can often be identified reliably in several
> ways. First, many applications provide prior information on the approximate
> number of spectral components; for example, in spectroscopy one may know the
> approximate number of dominant resonances, while in radar or array processing
> one may know the approximate number of targets or sources. Second, one may
> inspect the singular values of the lifted Hankel matrix and choose $r$ based
> on a spectral gap or standard model-selection criteria such as AIC, MDL, or
> BIC. Third, one may form a coarse interpolation in the time domain, compute
> its Fourier transform, and use the number of dominant peaks as an initial
> estimate of $r$.
>
> In practice, mild overestimation of $r$ is often safer than underestimation.
> If $r$ is chosen too small, some true spectral components are excluded,
> leading to an intrinsic modeling bias. If $r$ is chosen slightly larger than
> the true value, the correct signal subspace is still included, and the extra
> components often correspond to weak or noise directions that can be pruned
> afterward. On the other hand, choosing $r$ excessively large may increase
> computational cost and sensitivity to noise. We will add a brief discussion
> along these lines in the revised manuscript.
>
> >C4. The paper includes many mathmatical deviation. However, it may not be intuitive enough. A figure to illustrate the idea or the algorithms is recommended.
>
> **Reply:** We thank the reviewer for this suggestion. We agree that a figure
> illustrating the key idea behind the preconditioning framework would
> improve readability and make the mathematical derivations more accessible.
> In the revised manuscript, we will enhance Figure 1, which currently
> illustrates the equivalence between factor-space and matrix-space iterates,
> by incorporating additional details that summarize the main steps of JHGD
> as an algorithmic flowchart. This will provide a more intuitive overview of
> our algorithm and analysis, making the method easier to follow at a glance.

---

> > ### Author Rebuttal · Reviewer_8pAH · 2026-04-02
> >
> > Thank you for the detailed response.
> > My concerns were resolved. I believe that this paper can be accepted since its theory is promising and useful in the field of optimization.
> > I tend to raise my score to accept.
> > The reason for the no higher score is that the code for reproducing experimental results is still not available, and reproducibility is one of the evaluation dimensions according to the ICML review instructions. Although the authors provide many implementation details in the paper, it may be uneasy to reproduce the results.

---

> > > ### Author Response · Authors · 2026-04-02
> > >
> > > Thank you for your thoughtful follow-up review. We are glad that our response addressed your concerns, and we appreciate your emphasis on reproducibility. **Since we cannot include an anonymous code link in the response at this stage, we have attached the code below and will provide a GitHub link in the final revision.** We hope this further addresses your remaining concern and possibly supports raising your score.
> > >
> > > To run the code, simply save the code below as a single `.m` file (e.g., `demo_jhgd.m`) and execute it in MATLAB.
> > >
> > > ```
> > > clear; close all; clc;
> > >
> > > %% Parameters
> > > % nd : Signal length
> > > % r  : Number of frequency components
> > > % m  : Number of observed entries
> > > nd = 1999; r = 10; m = 500;
> > > [xs, K, x_star] = generate_signal(m, nd, r, false, false);
> > >
> > > %% Solve via JHGD
> > > opts = struct('True',x_star,'Step',1.8,'Tol',5e-6,'MaxIter',1000);
> > > x_rec = solve_jhgd(xs, K, r, opts);
> > >
> > > %% Main solver: Jacobi-type Hankel Gradient Descent
> > > function x_rec = solve_jhgd(y, K, r, o)
> > >     % Parse options
> > >     tol   = get_opt(o, 'Tol',     1e-6);
> > >     maxIt = get_opt(o, 'MaxIter', 500);
> > >     mu    = get_opt(o, 'Step',    0.5);
> > >     x_true = get_opt(o, 'True',   []);
> > >     track  = ~isempty(x_true);
> > >
> > >     N = length(y);
> > >     rho = length(K) / N;          % sampling rate
> > >     DIV = 1e2;                    % divergence threshold
> > >
> > >     % Sampling mask
> > >     w = zeros(N,1); w(K) = 1;
> > >
> > >     % Zero-pad for square Hankel matrix
> > >     if mod(N,2)==0, d=(N+2)/2; y=[y;0]; w=[w;0];
> > >     else,           d=(N+1)/2; end
> > >     L  = 2*d - 1;
> > >     nf = 2^nextpow2(L);
> > >
> > >     % Anti-diagonal averaging weights
> > >     ad = min((1:L)', min(d, L-(0:L-1)'));
> > >
> > >     % Spectral init: truncated SVD + Takagi factorization
> > >     ys = y / rho;
> > >     [U, S, V] = svds(@(v,t) hankel_mvp(ys,v,t,L,d,d), [d,d], r);
> > >     Z = zeros(d, r);
> > >     for i = 1:r
> > >         Z(:,i) = sqrt(U(:,i)' * conj(V(:,i))) * U(:,i);
> > >     end
> > >     Z = Z * sqrt(S);
> > >
> > >     % Initial Hankel projection
> > >     sig = hankel_proj(Z, L, nf, ad);
> > >     sig_prev = sig;
> > >
> > >     % Print header
> > >     if track, fprintf('\n %4s | %10s | %12s | %12s\n','Iter','Time(s)','Error','RelUpd');
> > >     else,     fprintf('\n %4s | %10s | %12s\n','Iter','Time(s)','RelUpd'); end
> > >     fprintf(repmat('-',1, 48+track*16)); fprintf('\n');
> > >
> > >     % Iterative refinement
> > >     tic;
> > >     for k = 1:maxIt
> > >         % Preconditioned gradient step
> > >         grad = (w.*sig - y)/rho - sig;
> > >         Zinv = Z / (Z'*Z);
> > >         HZ   = hankel_mult(grad, Zinv, L, d);
> > >         P    = (mu/2)*(HZ + Z) - (mu/4)*(Zinv*(Z'*HZ) + Z);
> > >
> > >         % Update factor and project
> > >         Z   = Z - P;
> > >         sig = hankel_proj(Z, L, nf, ad);
> > >
> > >         % Convergence check
> > >         rel = norm(sig - sig_prev) / norm(sig_prev);
> > >         elapsed = toc;
> > >
> > >         % Print progress
> > >         if track
> > >             err = norm(sig(1:N) - x_true) / norm(x_true);
> > >             fprintf(' %4d | %10.4f | %12.4e | %12.4e\n', k, elapsed, err, rel);
> > >         else
> > >             fprintf(' %4d | %10.4f | %12.4e\n', k, elapsed, rel);
> > >         end
> > >         sig_prev = sig;
> > >
> > >         if ~isfinite(rel) || rel < tol || rel > DIV
> > >             if rel < tol,   fprintf('Converged at iteration %d.\n', k);
> > >             else,            fprintf('Stopped (divergence or NaN).\n'); end
> > >             break
> > >         end
> > >     end
> > >     if k == maxIt, fprintf('Reached max iterations (%d).\n', maxIt); end
> > >
> > >     x_rec = sig(1:N);
> > > end
> > >
> > > %% Signal generation
> > > function [xs, K, x_star, freq, amp] = generate_signal(m, nd, r, sep, damp)
> > >     amp = exp(1i*2*pi*rand(r,1)) .* (1 + 10.^(rand(r,1)*10/20));
> > >     xs = zeros(nd,1);
> > >     K = randsample(nd, m);
> > >
> > >     % Frequency placement
> > >     if sep
> > >         d1 = 1.5/nd; E1 = 1 - r*d1;
> > >         assert(E1 >= 0, 'Model order too large for separation condition');
> > >         fs = rand(1, r+1);
> > >         freq = cumsum(E1*fs(1:r)/sum(fs) + d1);
> > >     else
> > >         freq = randperm(nd, r) / nd;
> > >     end
> > >
> > >     % Signal construction (with or without damping)
> > >     if damp
> > >         decay = -1 ./ (32 + 32*rand(r,1));
> > >         x_star = exp((0:nd-1)' * (decay' + 1i*2*pi*freq)) * amp;
> > >     else
> > >         x_star = exp(2*pi*1i*(0:nd-1)' * freq) * amp;
> > >     end
> > >     xs(K) = x_star(K);
> > > end
> > >
> > > %% Helper: retrieve struct field with default
> > > function v = get_opt(o, f, d)
> > >     if isfield(o,f), v = o.(f); else, v = d; end
> > > end
> > >
> > > %% Helper: Hankel projection via anti-diagonal averaging of ZZ^T
> > > function x = hankel_proj(Z, L, nf, w)
> > >     x = sum(ifft(fft(Z,nf).^2), 2);
> > >     x = x(1:L) ./ w;
> > > end
> > >
> > > %% Helper: Hankel(x) * conj(Z) via FFT
> > > function R = hankel_mult(x, Z, L, d)
> > >     nf = 2^nextpow2(L);
> > >     R  = ifft(fft(flip(conj(Z)), nf) .* fft(x, nf));
> > >     R  = R(d:L, :);
> > > end
> > >
> > > %% Helper: matrix-free Hankel-vector product for svds
> > > function z = hankel_mvp(x, v, tflag, L, nr, nc)
> > >     nf = 2^nextpow2(L);
> > >     if tflag(1) == 'n'
> > >         z = ifft(fft(flip(v),nf) .* fft(x,nf));       z = z(nc:L);
> > >     else
> > >         z = ifft(fft(flip(v),nf) .* fft(conj(x),nf)); z = z(nr:L);
> > >     end
> > > end
> > >
> > > ```

---

### Official Review · Reviewer_WAoG · 2026-03-13

**Soundness:** 3
**Presentation:** 3
**Significance:** 2
**Originality:** 3
**Overall Recommendation:** 4
**Confidence:** 2

**Summary:**

This paper introduces Jacobi-Preconditioned Hankel Gradient Descent (JHGD), a non-convex optimization algorithm designed for reconstructing spectrally sparse signals from partial observations. By formulating the task as a low-rank Hankel matrix completion problem, the authors address the convergence bottlenecks common in ill-conditioned signal recovery.

**Compliance With Llm Reviewing Policy:**

Affirmed.

**Final Justification:**

I thank the authors for their detailed responses. The theoretical analysis of the initialization overhead and the clarification on the unique constraints of Hankel matrices effectively address my concerns. The consistency in stopping criteria across methods also ensures a fair comparison. I maintain my positive assessment.

**Key Questions For Authors:**

1.The refinement stage relies on a specific spectral initialization. What is the computational overhead of this initialization compared to the total cost of the iterations across the reported signal lengths?
2.Please clarify the performance of the approximate preconditioner in cases where the signal is ill-conditioned with rapidly decaying singular values.
3.Theorem 3.1 claims a contraction rate independent of the condition number. Does this independence still hold empirically when the frequencies are not well-separated?
4.To fully demonstrate the competitive edge of the proposed algorithm, the authors are encouraged to include comparisons with more recent SOTA methods.
5.Since the authors highlight the potential impact on applications like seismic imaging, it would be a great addition if they could demonstrate the algorithm's performance using a real-world dataset.

**Limitations:**

yes

**Strengths And Weaknesses:**

The strengths can be summarized as below:
1.The manuscript is well-motivated, targeting the computational bottleneck of high-dimensional spectral reconstruction in compressed sensing.
2.The introduction of the Jacobi preconditioner provides a clever and mathematically sound way to balance the scaling of different frequency components, leading to significantly faster convergence compared to standard gradient descent.
3.The proposed JPGD algorithm demonstrates high robustness, maintaining low reconstruction error even in the presence of noise and high undersampling ratios.

weakness:
I carefully address some issues with some comments.
1.The introduction does not sufficiently emphasize why existing non-convex methods fundamentally suffer from ill-conditioning, beyond stating empirical inefficiency.
2.The experimental comparison may not be entirely fair due to different step-size strategies and stopping criteria across methods.
3.The simplification in Theorem 2.3 assumes that singular values are of the same order of magnitude. It remains to be explained whether the performance of this approximate preconditioner is still robust when singular values decay rapidly.
4.The paper compares JHGD against SGD (2021) and older methods like PGD and FIHT. However, in the field of low-rank matrix recovery, there have been many Riemannian preconditioning methods developed in recent years.

---

> ### Author Rebuttal · Authors · 2026-03-31
>
> >B1. What is the computational overhead of this initialization compared to the total cost of the iterations?
>
> **Reply:** We thank the reviewer for this helpful question. The spectral initialization
> has the same order of complexity as a single refinement iteration. The
> initialization computes a rank-$r$ truncated spectral decomposition of
> $p^{-1}\mathcal{H}\mathcal{P}_\Omega(\mathbf{y})$ via a Lanczos-type
> truncated SVD with FFT-based Hankel matrix-vector multiplications, rather than by explicitly forming the Hankel
> matrix and applying a dense SVD. The resulting initialization cost is $\mathcal{O}(nr\log n + nr^2)$.
>
> Each refinement iteration has the same cost. Since the refinement stage
> converges linearly, reaching $\varepsilon$-accuracy requires
> $\mathcal{O}(\log(1/\varepsilon))$ iterations, giving a total refinement
> cost of $\mathcal{O}\bigl((nr\log n + nr^2)\log(1/\varepsilon)\bigr)$.
> The initialization cost is therefore dominated by the overall refinement
> cost. We will add this discussion to the revised manuscript.
>
> >B2. Does Theorem 3.1’s condition-number-independent contraction rate still hold empirically without well-separated frequencies?
>
> **Reply:** We thank the reviewer for this question. The condition-number-independent
> contraction rate and the frequency separation condition address two distinct
> aspects of the problem.
>
> The condition-number-independent contraction rate in Theorem 3.1 is a consequence of the Jacobi preconditioner, which compensates for the ill-conditioning of the factored objective. Frequency separation serves a different purpose: it is a standard sufficient condition ensuring that the underlying low-rank Hankel matrix satisfies incoherence, without which recovery from partial observations becomes intrinsically harder.
>
> In short, preconditioning governs the speed of convergence, whereas frequency separation governs recoverability through incoherence.
>
> >B3. The authors are encouraged to include comparisons with more SOTA methods from low-rank matrix recovery.
>
> **Reply:** We thank the reviewer for raising this point. In our experiments, we
> compare JHGD with all existing state-of-the-art methods specifically
> designed for low-rank Hankel matrix recovery. To the best of our knowledge,
> no more recent algorithms have been proposed for this problem.
>
> We agree that the broader literature has seen recent progress in Riemannian
> preconditioning and related techniques. However, extending these methods to
> the Hankel setting is nontrivial: a Hankel matrix must satisfy
> anti-diagonal equality constraints, reducing the degrees of freedom from
> $O(nr)$ for a general rank-$r$ matrix to only $O(r)$, which fundamentally
> changes both the optimization geometry and the analysis. Adapting recent
> Riemannian methods to this setting is an interesting direction for future
> work but is beyond the scope of this paper.
>
> >B4. It would be a great addition if they demonstrate the performance using a real-world dataset.
>
> **Reply:** We thank the reviewer for this suggestion. We agree that evaluation on
> real-world data would be valuable. The main contribution of this paper is
> methodological and theoretical, and controlled synthetic experiments are
> the most appropriate setting for this purpose, as they provide ground truth
> and allow us to isolate the effects of sampling, noise, and problem size.
> A meaningful real-data study would additionally require domain-specific
> acquisition and evaluation protocols beyond the scope of this work. We will
> clarify this in the revision and present real-world validation as an
> important direction for future work.
>
> >B5. The numerical comparison may not be fair due to different step-size strategies and stopping criteria.
>
> **Reply:** We thank the reviewer for raising this concern. All factor-space methods
> (PGD, SHGD, SGD, and JHGD) use the same stopping criterion
> $||\mathbf{x}_{t+1}-\mathbf{x}_t||/||\mathbf{x}_t || \leq 5\times10^{-6}$,
> under which PGD, SHGD, and SGD attain errors on the order of $10^{-5}$
> while our JHGD attains a slightly smaller error. For FIHT (a matrix-space
> method), we set the tolerance to $5\times10^{-5}$ so that it achieves
> comparable accuracy; this allows FIHT to stop sooner, favoring it in
> runtime comparisons. Stepsizes are fixed for each method across all tested dimensions, sparsity levels, and sampling ratios in Tables 1-3, chosen according to the recommendations in the original references.
>
> In summary, **the stopping criteria are calibrated so that all methods
> achieve comparable reconstruction accuracy, and the stepsizes follow the
> respective authors' guidelines.** We will clarify these details in the
> revised manuscript.
>
> >B6. It remains unclear whether the performance of the approximate preconditioner is robust when singular values decay rapidly.
>
> **Reply:** Please refer to our reply to Reviewer xaEt, Comment D1.

---

> > ### Author Rebuttal · Reviewer_WAoG · 2026-04-07
> >
> > Thank you for your thoughtful response. I have revised my score accordingly.

---

### Official Review · Reviewer_4hab · 2026-03-23

**Soundness:** 3
**Presentation:** 3
**Significance:** 2
**Originality:** 3
**Overall Recommendation:** 5
**Confidence:** 4

**Summary:**

The paper studies spectrally sparse signal reconstruction from partial time-domain samples, building on the well-known low-rank Hankel modelling. It builds on a symmetric factorization of the complex-valued Hankel matrix and proposes a tailored preconditioning approach built into a gradient descent method with the goal to accelerated convergence. To derive this approach, the authors use a Riemannian formulation with a low-rank retraction that is tailored to the problem structure. The paper further presents recovery guarantees under incoherence and sampling assumptions together with numerical experiments comparing the proposed method against several existing nonconvex Hankel reconstruction algorithms, in which the proposed method JHGD shines in terms of convergence speed compared to baseline algorithms.

**Compliance With Llm Reviewing Policy:**

Affirmed.

**Final Justification:**

The authors have addressed the majority of my questions and the stated weaknesses. While I am not able to audit the proof of the stated update of Theorem 2.3, which future readers of the final camera-ready version of the paper should do, the discussion period provides sufficient reason to update my assessment of the paper in a positive direction.

**Key Questions For Authors:**

1. How is (13) possible despite $\mathcal{P}_A$ being an approximation?
2. Please correct the issues in proofs pointed out above.
3. Can you elaborate on how the retraction can work despite it not being well-defined? Are there any numerical issues observed in practice?

**Limitations:**

Yes.

**Strengths And Weaknesses:**

## Strengths

This manuscript contains several strengths, which make it a contender for acceptance at a machine learning venue such as ICML:
 1. The spectrally sparse signal reconstruction problem is a classical one in signal processing with ample applications in time series analysis and machine learning. The authors propose a (to the best of my knowledge,) new algorithm which has favorable properties in terms of convergence speed. The numerical experiments are relative extensive and empirical phase transitions comparing to many relevant algorithms are provided.
 2. The authors, if the proofs are correct, seem to establish a linear convergence rate of the proposed method JHGD under assumptions that a similar to the one of related gradient descent methods for the problem (such as Li et al. 2024), even though, e.g., the dependence of the sample complexity on the dimension and/or the condition number for convergence is somewhat worse in the established theory.
 3. The approach of using a Jacobi operator of the underlying Riemannian structure to construct a preconditioner might be of independent interest for similar problems.

Furthermore, the paper is generally well-written.

## Weaknesses
On the other hand, there are some issues in the theoretical part of the paper that need to be pointed out. While the overall proof strategy follows the one of other gradient descent methods with matrix factorization structure and is sound on a high level, there are issues in the details of some of the proofs:
1. There seems to be an issue with the definition of the retraction $\mathcal{R}(A)$ on page 13: It is claimed that the mapping (21) is well-defined; however, based on other parts of the proofs in the paper, $\mathcal{J}$ is not injective, which means that the matrix $\mathbf{\Delta}$ in $A = X + \mathcal{J}(\mathbf{\Delta})$ is not uniquely determined (differently from what is claimed): In particular, in the proof of Theorem A.2 on page 20, a non-trivial element in the kernel of $\mathcal{J}$ is computed.
2. The approximation notion in Theorem 2.3 and its proof (Appendix A.5) is unclear and not rigorous (no norm bound / probability bound is provided). This theorem is at the core of the motivation of the preconditioning operator $\mathcal{P}(A)$ which drops some terms compared to $\mathcal{P}_E$. A more detailed / rigorous analysis of this would be beneficial to the overall narrative of the paper.
3. The proof of Theorem A.1. seems to be flawed: It is unclear what $\mathbf{V}_{ij}$ is for indices where $j > i$ but $min(i,j) \leq r$$ in the proof of Theorem A.1 in line 814 on page 15: This matrix seems to be not defined in Theorem 2.1. This needs to be clarified.
4. The proof of Theorem 3.2 is very short and not understandable. While the ingredients for the proof seem to be in place, it is unclear how all of them fit together.

Furthermore, it is a limitation that no implementation of the algorithms of presented in the paper
are provided with the submission, making a verification of the experimental results challenging.

Finally, a weakness of the theory is that the faster convergence of JHGD compared to, e.g., SHGD, does not become clear from the convergence results.

### Minor issues

- p. 5, line 239: "busing"
- p. 6, line 317: The initial iterate $\mathbf{X}_0$ should be replaced by $\widetilde{\mathbf{Z}}_K$, otherwise confusion with the definition of line 304 is possible.

---

> ### Author Rebuttal · Authors · 2026-03-31
>
> >A1. There seems to be an issue with the definition of the retraction. Are there any numerical issues?
>
> **Reply:** We thank the reviewer for this insightful comment. We agree that the original definition of $\mathcal{R}$ is not well-defined. This can be fixed by adding a restriction. Writing $\mathbf{A} = \mathbf{X} + \mathcal{J}\mathcal{J} ^\dagger \boldsymbol{\Xi}$, we remove the ambiguity
> by imposing a restriction on $\boldsymbol{\Xi}$ that $(\mathbf{I} - \mathcal{J}\mathcal{J} ^\dagger)\boldsymbol{\Xi} = (\mathbf{I} - \mathcal{J}\mathcal{J}^\dagger)\nabla F(\mathbf{X})$, under which **$\mathcal{J}\mathcal{J}^\dagger\boldsymbol{\Xi}$ is uniquely defined**. Then $\boldsymbol{A}$ can be uniquely parametrized by $\boldsymbol{\Xi}$, thus $\mathcal{R}$ becomes well-defined. The revised definition is
> as follows.
>
> Suppose $\mathbf{X} = \mathcal{Q}(\mathbf{Z}) =
> \mathbf{Z}\mathbf{Z}^\top$. For any $\mathbf{A} \in \mathbf{X} +
> \mathbb{T}_{\mathbf{X}}$ parametrized as $\mathbf{A} = \mathbf{X} +
> \mathcal{J}\mathcal{J}^\dagger \boldsymbol{\Xi}$ for some
> $\boldsymbol{\Xi} \in \mathbb{C} ^{n \times n} _S$ satisfying $(\mathbf{I} - \mathcal{J}\mathcal{J} ^\dagger )\boldsymbol{\Xi} =(\mathbf{I} - \mathcal{J}\mathcal{J} ^\dagger) \nabla F(\mathbf{X})$, we define $\mathcal{R}: \mathbf{X} + \mathbb{T} _{\mathbf{X}} \to \mathcal{M}_r$ by
>
> $$\mathcal{R}(\mathbf{A}) = \mathcal{Q}(\mathbf{Z} + \mathcal{J} ^\dagger \boldsymbol{\Xi}) = (\mathbf{Z} + \mathcal{J} ^\dagger
> \boldsymbol{\Xi})(\mathbf{Z} + \mathcal{J} ^\dagger \boldsymbol{\Xi}) ^\top \in \mathcal{M}_r,$$
>
> which is well-defined by both existence and uniqueness.
>
> This revised definition precisely captures our intended formulation, since the iteration formula is $\mathbf{X}_{t+1} = \mathcal{R}(\mathbf{X}_t + \mathcal{J}\mathcal{J}^\dagger \nabla F(\mathbf{X}_t))$ with $\boldsymbol{\Xi} = \nabla F(\mathbf{X}_t)$ fixed. All proofs remain
> valid, as they already admit this refined version of $\mathcal{R}$.
>
> **The definition of $\mathcal{R}$ does not affect any numerical experiments, since the algorithm is
> implemented entirely in the factor space (Algorithm 1) and $\mathcal{R}$ serves only as an analytical tool.** We will revise the manuscript accordingly.
>
>
> >A2. The approximation notion in Theorem 2.3 and its proof (Appendix A.5) is unclear and not rigorous (no norm bound / probability bound is provided).
>
> **Reply:** We thank the reviewer for this suggestion. In the revision, we strengthen
> Theorem 2.3 by adding an explicit expectation bound:
>
> $$\mathbb{E} _{\mathbf{M}} \left[ || \Delta _{ij} (\mathbf{M}) || _F ^2\right] \le \frac{1}{2\sigma _{\min} ^4}\mathbb{E} _{\mathbf{M}} \left[ ||\mathbf{M}|| _2^2\right],$$
>
> where $\Delta_{i j}$ denotes the $(i, j)$-th component of the difference between $\mathcal{P}_E$ and $\mathcal{P}_A$ for all indices satisfying $j<i\leq r$, and $\sigma _{\min}$ is the smallest singular value of $\mathbf{Z}$.
>
> Please refer to our reply to Reviewer xaEt, Comment D1 for more details about approximation error.
>
> >A3. It is unclear what $\mathbf{V}_{ij}$ is for indices where  $j > i$ but $\min(i,j) \leq r$ in the proof of Theorem A.1
>
> **Reply:** We thank the reviewer for pointing this out. In the proof of Theorem A.1, $\mathbf{M}$ is expanded in the full basis from Theorem A.2, so for $j>i$ the relevant terms are $\tilde{\mathbf{V}} _{ij}$, not $\mathbf{V} _{ij}$. Since $\tilde{\mathbf{V}} _{ij}\in\ker(\mathcal{J})$, these terms vanish under $\mathcal{J}$ and therefore do not enter the singular-pair
> characterization in Theorem 2.1. We will make this explicit in the proof.
>
> >A4. The proof of Theorem 3.2 is short and not understandable.
>
> **Reply:** We agree that the proof of Theorem 3.2 was too compressed and will expand
> it in the revision. The key observation is that $||\mathbf{x}_t - \mathbf{x}|| = ||\mathcal{D}^{-1}(\mathbf{y}_t - \mathbf{y})|| \le||\mathbf{y}_t - \mathbf{y}||$, so it suffices to verify the assumptions of Theorem B.2 and the condition in Corollary B.1.
> The first two assumptions follow exactly as in the proof of Theorem 3.1,
> and the third holds with probability at least $1 - (2K+1)n_s^{-2}$ by
> choosing $K$ as in line 312. Combining these with the condition in
> Corollary B.1 yields the claimed measurement bound and completes the proof.
>
> >A5. How is (13) possible despite $\mathcal{P}_A$ being an approximation?
>
> **Reply:** We thank the reviewer for this question. Both $\mathcal{P}_E \mathcal{J}^* \nabla F(\boldsymbol{X})$ and $\mathcal{P}_A \mathcal{J}^* \nabla F(\boldsymbol{X})$ decompose as $\Delta_1 + \Delta_2$ with $\Delta_1 \in \operatorname{range}(\mathcal{J}^*)$ and $\Delta_2 \in \operatorname{ker}(\mathcal{J})$. Since applying $\mathcal{J}$ eliminates $\Delta_2$, only $\mathcal{J}\Delta_1$ enters the iteration. As shown in Appendix A.6, $\mathcal{P}_E$ and $\mathcal{P}_A$ induce the same $\mathcal{J}\Delta_1$, so they agree on the tangent-space projection relevant to (13) while potentially differing only in the null-space component.

---

> > ### Author Rebuttal · Reviewer_4hab · 2026-04-04
> >
> > I thank the authors for their replies. The rebuttal addresses many of my concerns, however, I have two follow-up questions:
> > 1. Can you provide an exact updated formulation for Theorem 2.3, so that the reviewers can understand your new bound better?
> > 2. Can you outline how the assumption of your convergence theory are better, worse, or in line with the ones of previous works such as Li et al. 2024, in particular, but not necessarily only, in its dependence on the sample complexity?
> > 3. I have trouble understanding the reconstruction error vs. time plots of Figure 2, in particular, the advantage of JHGD over SHGD, FIHT and PGD comes from very long times taken by these algorithms in the very first steps. Can you explain why this not potentially an artifact of a suboptimal implementation? Is there a reason why these algorithms take this additional time in the first iterations?

---

> > > ### Author Response · Authors · 2026-04-05
> > >
> > > Thank you for your insightful follow-up questions. Below, we address each of them in turn.
> > >
> > > >1. Can you provide an exact updated formulation for Theorem 2.3?
> > >
> > > **Reply:** We provide the updated statement of Theorem 2.3 below; its full version was omitted from the initial rebuttal because of the length limit.
> > >
> > > **Theorem 2.3 (Revised).** Assume $\boldsymbol{M}$ is uniformly distributed in $\boldsymbol{u}_i$ and $\boldsymbol{v}_j$. Then we have
> > >
> > > $$
> > > \mathbb{E} _{\boldsymbol{M}}  \left[\left \Vert\frac{\left(\sigma _j \boldsymbol{u} _i^H \boldsymbol{M} \bar{\boldsymbol{v}} _j+ \sigma_i \boldsymbol{u} _j ^H \boldsymbol{M} \bar{\boldsymbol{v}} _i \right)}{2(\sigma _i ^2+\sigma _j ^2) ^2} \left(\sigma _j \boldsymbol{u} _i \boldsymbol{v} _j ^\top + \sigma _i \boldsymbol{u} _j \boldsymbol{v} _i ^\top \right)  -\frac{1}{4\sigma _j ^2} \boldsymbol{u} _i \boldsymbol{u} _i ^H \boldsymbol{M} \bar{\boldsymbol{v}} _j \boldsymbol{v} _j ^\top - \frac{1}{4\sigma _i ^2} \boldsymbol{u} _j \boldsymbol{u} _j ^H \boldsymbol{M} \bar{\boldsymbol{v}} _i \boldsymbol{v} _i ^\top \right\Vert _F ^2  \right] \leq \left(\frac{1}{2 \sigma _{\min} ^4} \right) \mathbb{E} _{\boldsymbol{M}} \left[\left\Vert \boldsymbol{M} \right\Vert _2 ^2 \right],
> > > $$
> > >
> > > where $\sigma _{\min}$ is the minimum singular value of $\boldsymbol{Z}$; $\boldsymbol{u} _i$ and $\boldsymbol{v} _j$ are singular vectors of $\boldsymbol{Z}$.
> > >
> > > Following the reviewer's suggestion, we have added an explicit upper bound in
> > > expectation, which shows that the approximation remains reliable as long as
> > > $\sigma_{\min}(\boldsymbol{Z})$ is bounded away from zero. In practice, the
> > > factor matrix $\boldsymbol{Z}$ typically stays full rank throughout the iterations. If the factor matrix becomes numerically rank deficient, this
> > > usually indicates that the prescribed rank $r$ has been overestimated;
> > > reducing $r$ restores both the conditioning and the approximation quality.
> > >
> > > This revised statement will be incorporated into the final version of the
> > > paper.
> > >
> > > >2. Can you outline how the assumption of your convergence theory are better, worse, or in line with the ones of previous works such as Li et al. 2024, in particular, but not necessarily only, in its dependence on the sample complexity?
> > >
> > > **Reply:** In terms of sample complexity, both JHGD
> > > and FIHT require $\mathcal{O}(r^2 \log^2(n_s))$, slightly larger than the
> > > $\mathcal{O}(r^2 \log(n_s))$ of PGD and SHGD by one logarithmic factor.
> > > This stems from the proof framework: our analysis is carried out in matrix
> > > space, which avoids the factorization ambiguity inherent in complex-symmetric
> > > matrix factorization, whereas factor-space analyses such as SHGD require a more
> > > involved distance construction to handle this ambiguity. We will clarify this comparison in the revised manuscript.
> > >
> > > **Our main contribution is not a tighter sample complexity bound but twofold:
> > > (i) a principled approach to designing efficient preconditioners that yield a
> > > convergence rate independent of the condition number, and (ii) a framework
> > > that connects the factor-space and matrix-space viewpoints via a generator,
> > > thereby combining factor-space efficiency with matrix-space analysis.** Under
> > > this framework, the preconditioner admits a clear geometric interpretation as an
> > > orthogonal projection onto the tangent space.
> > >
> > > >3. I have trouble understanding the reconstruction error vs. time plots of Figure 2, in particular, the advantage of JHGD over SHGD, FIHT and PGD comes from very long times taken by these algorithms in the very first steps.
> > >
> > > **Reply:** We apologize for the lack of clarity. Each point on a curve in Figure 2 does
> > > not correspond to a single iterate. Instead, we define 50 logarithmically
> > > spaced error levels (from $0.5$ to $5\times10^{-7}$) and record the
> > > wall-clock time at which each level is first achieved, averaged over 10
> > > trials. **The large horizontal gaps for SHGD, PGD, and FIHT therefore reflect the
> > > substantial wall-clock time these methods need to reach those error levels,
> > > not a small number of plotted iterates.**
> > >
> > > The reasons for the long wall-clock times differ across methods. **SHGD
> > > and PGD have low per-iteration cost, but they converge slowly and therefore
> > > require many iterations to reach each error level.** This is because they are
> > > first-order methods whose convergence rates depend on the condition number
> > > $\kappa$. **FIHT, by contrast, converges in fewer iterations but incurs a high
> > > per-iteration cost** because it requires an SVD at every iteration. Both cases
> > > result in long wall-clock times.
> > >
> > > Finally, we note that all implementations were provided by the respective
> > > authors of each method, and the experimental settings, including step sizes and the signal generation procedure, follow those in the original papers. We will add a clarifying note in the revised
> > > manuscript on how Figure 2 should be interpreted.

---

### Decision · Program_Chairs · 2026-04-30

**Decision:**

Accept (spotlight)

**Comment:**

This paper proposes a preconditioned Riemannian optimization algorithm for reconstructing spectrally sparse signals. Reviewers note that the problem is well motivated, the method novel, and that the theoretical results are high quality. Authors have addressed during the discussion period all the reviewers' concerns, so all reviewers are recommend acceptance.